# Sample Complexity of Nonparametric Off-Policy Evaluation on Low-Dimensional Manifolds using Deep Networks

**Xiang Ji**[*]  **Minshuo Chen**[*]  **Mengdi Wang**[*]  **Tuo Zhao**[†]

[*]Princeton University  [†] Georgia Institute of Technology

`{xiangj, mc0750, mengdiw}`@princeton.edu  `tourzhao`@gatech.edu

## ABSTRACT

We consider the off-policy evaluation problem of reinforcement learning using deep convolutional neural networks. We analyze the deep fitted Q-evaluation method for estimating the expected cumulative reward of a target policy, when the data are generated from an unknown behavior policy. We show that, by choosing network size appropriately, one can leverage any low-dimensional manifold structure in the Markov decision process and obtain a sample-efficient estimator without suffering from the curse of high data ambient dimensionality. Specifically, we establish a sharp error bound for fitted Q-evaluation, which depends on the intrinsic dimension of the state-action space, the smoothness of Bellman operator, and a function class-restricted $\chi^2$-divergence. It is noteworthy that the restricted $\chi^2$-divergence measures the behavior and target policies' *mismatch in the function space*, which can be small even if the two policies are not close to each other in their tabular forms. We also develop a novel approximation result for convolutional neural networks in Q-function estimation. Numerical experiments are provided to support our theoretical analysis.

## 1 INTRODUCTION

Off-policy Reinforcement Learning (RL) [38, 40] is an important area in decision-making applications, when the data cannot be acquired with arbitrary policies. For example, in clinical decision-making problems, experimenting new treatment policies on patients is risky and may raise ethical concerns. Therefore, we are only allowed to generate data using certain policies (or sampling distributions), which have been approved by medical professionals. These so-called "behavior policies" are unknown but could impact our problem of interest, resulting in distribution shift and insufficient data coverage of the problem space. In general, the goal is to design algorithms that need as little data as possible to attain desired accuracy.

A crucial problem in off-policy RL is policy evaluation. The goal of Off-Policy Evaluation (OPE) is to estimate the value of a new target policy based on experience data generated by existing behavior policies. Due to the mismatch between behavior and target policies, the off-policy setting is entirely different from the on-policy one, in which policy value can be easily estimated via Monte Carlo.

A popular algorithm to solve OPE is the fitted Q-evaluation method (FQE), as an off-policy variant of the fitted Q-iteration [28, 15, 75]. FQE iteratively estimates Q-functions by supervised regression using various function approximation methods, e.g., linear function approximation, and has achieved great empirical success [65, 20, 21], especially in large-scale Markov decision problems. Complementary to the empirical studies, several works theoretically justify the success of FQE. Under linear function approximation, [31] show that FQE is asymptotically efficient, and [15] further provide a minimax optimal non-asymptotic bound, and [47] provide a variance-aware characterization of the distribution shift via a weighted variant of FQE. [75] analyze FQE with realizable, general differentiable function approximation. [37, 64] tackle OPE for even more general function approximation, but they require stronger assumptions such as full data coverage. [16] focus on on-policy estimation and study a kernel least square temporal difference estimator.

Recently, deploying neural networks in FQE has achieved great empirical success, which is largely due to networks' superior flexibility of modeling in high-dimensional complex environments

[65, 20, 21]. Nonetheless, the theory of FQE using deep neural networks has not been fully understood. While there are existing results on FQE with various function approximators [28, 15, 75], many of them are not immediately applicable to neural network approximation. [18] focus on the online policy learning problem and studies DQN with feed-forward ReLU network; a concurrent work [73] studies offline policy learning with realizable, general differentiable function approximation. Notably, a recent study [51] provide an analysis of the estimation error of nonparametric FQE using feed-forward ReLU network, yet this error bound grows quickly when data dimension is high. Moreover, their result requires full data coverage, i.e., every state-action pair has to eventually be visited in the experience data. Precisely, besides universal function approximation, there are other properties that contribute to the success of neural networks in supervised learning, for example, its ability to **adapt to the intrinsic low-dimensional structure of data**. While these properties are actively studied in the deep supervised learning literature, they have not been reflected in RL theory. Hence, it is of interest to examine whether these properties still hold in a problem with sequential nature under standard assumptions and how neural networks can take advantage of such low-dimensional structures in OPE.

**Main results.** This paper establishes sample complexity bounds of deep FQE using convolutional neural networks (CNNs). Different from existing results, our theory exploits the intrinsic geometric structures in the state-action space. This is motivated by the fact that in many practical high-dimensional applications, especially image-based ones [59, 11, 76], the data are actually governed by a much smaller number of intrinsic free parameters [2, 55, 32]. See an example in Figure 1.

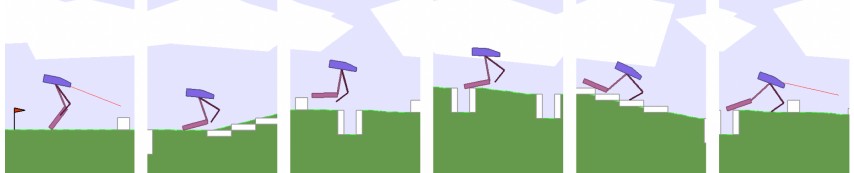

Figure 1: An example of state-action space with low-dimensional structures. The states of OpenAI Gym Bipedal Walker can be visually displayed in high resolution (e.g., $200 \times 300$), while they are internally represented by a 24-tuple [29].

Consequently, we model the state-action space as a $d$-dimensional Riemannian manifold embedded in $\mathbb{R}^D$ with $d \ll D$. Under some standard regularity conditions, we show CNNs can efficiently approximate Q-functions and allow for fast-rate policy value estimation—free of the curse of ambient dimensionality $D$. Moreover, our results do not need strong data coverage assumptions. In particular, we develop a function class-restricted $\chi^2$-divergence to quantify the mismatch between the visitation distributions induced by behavior and target policies. The function class can be viewed as a smoothing factor of the distribution mismatch, since the function class may be insensitive to certain differences in the two distributions. Our approximation theory and mismatch characterization significantly sharpen the dimension dependence of deep FQE. In detail, our theoretical results are summarized as follows:

**(I)** Given a target policy $\pi$, we measure the distribution shift between the experience data distribution $\{q_h^{\text{data}}\}_{h=1}^H$ and the visitation distribution of target policy $\{q_h^\pi\}_{h=1}^H$ by

$$\kappa = \frac{1}{H} \sum_{h=1}^H \sqrt{\chi_{\mathcal{Q}}^2(q_h^\pi, q_h^{\text{data}}) + 1}, \tag{1}$$

where $\chi_{\mathcal{Q}}^2(q_h^\pi, q_h^{\text{data}})$ is the restricted $\chi^2$-divergence between $q_h^\pi$ and $q_h^{\text{data}}$ defined as

$$\chi_{\mathcal{Q}}^2(q_h^\pi, q_h^{\text{data}}) = \sup_{f \in \mathcal{Q}} \frac{\mathbb{E}_{q_h^\pi}[f]^2}{\mathbb{E}_{q_h^{\text{data}}}[f^2]} - 1$$

with $\mathcal{Q}$ being a function space relevant to our algorithm.

**(II)** We prove that the value estimation error of a target policy $\pi$ is

$$\mathbb{E}|v^\pi - \widehat{v}^\pi| = \widetilde{\mathcal{O}}\left(\kappa H^2 K^{-\frac{\alpha}{2\alpha+d}}\right),$$

where $K$ is the effective sample size of experience data sampled by the behavior policy (more details in Section 3), $H$ is the length of the horizon, $\alpha$ is the smoothness parameter of the Bellman operator,

$\kappa$ is defined in (1), and $\widetilde{\mathcal{O}}(\cdot)$ hides some constant depending on state-action space and a polynomial factor in $D$.

We compare our results with several related works in Table 1. Both [15] and [75] consider parametric function approximation to the Q-function: [15] study linear function approximation, and [75] assume third-order differentibility, which is not applicable to neural networks with non-smooth activation. On the other hand, [51] use feed-forward neural networks with ReLU activation to parametrize nonparametric Q-functions, but they do not take any low-dimensional structures of the state-action space into consideration. Therefore, their result suffer from the curse of dimension $D$. Moreover, they characterize the distribution shift with the absolute density ratio between the experience data distribution and the visitation distribution of target policy, which is strictly larger than our characterization in restricted $\chi^2$-divergences. As can be seen from our comparison, our result improves over existing results. Moreover, since CNNs have been widely used in deep RL applications and also retain state-of-the-art performance [48, 22], our consideration of CNNs is a further step of bridging practice and theory.

Table 1: $\kappa_1$ and $\kappa_2$ are measures of distribution shift with respect to their respective regularity spaces; $D_{\text{eff}}$ denotes the effective dimension of the function approximator in [75], which usually suffers from the curse of dimensionality; $\kappa_3$ is the absolute density ratio between the data distribution and target policy's visitation distribution; $\kappa$ is defined in (1) and is no larger than and often substantially smaller than $\kappa_3$. See in-depth discussions in Section 4.

| Work | Regularity | Approximation | Estimation Error |
|---|---|---|---|
| [15] | Linear | None | $\widetilde{\mathcal{O}}\left(H^2\sqrt{\kappa_1 D/K}\right)$ |
| [75] | Third-time differentiable | None | $\widetilde{\mathcal{O}}\left(H^2\sqrt{\kappa_2 D_{\text{eff}}/K}\right)$ |
| [51] | Besov | Feed-Forward ReLU Net | $\widetilde{\mathcal{O}}\left(H^{2-\alpha/(2\alpha+2D)}\kappa_3 K^{-\alpha/(2\alpha+2D)}\right)$ |
| This work | Besov | CNN | $\widetilde{\mathcal{O}}\left(H^2\kappa K^{-\alpha/(2\alpha+d)}\right)$ |

**Additional Related Work**  Besides FQE, there are other types of methods in the OPE literature. One popular type is using importance sampling to reweigh samples by the distribution shift ratio [56], but importance sampling suffers from large variance, which is exponential in the length of the horizon in the worst case. To address this issue, some variants with reduced variance such as marginal importance sampling (MIS) [68] and doubly robust estimation [35, 61] have been developed. For the tabular setting with complete data coverage, [72] show that MIS is an asymptotically efficient OPE estimator, which matches the Cramer-Rao lower bound in [35]. Moreover, a line of work [49, 50, 39, 74] focuses on policy evaluation without function approximation using MIS and linear programming.

**Notation**  For a scalar $a > 0$, $\lceil a \rceil$ denotes the ceiling function, which gives the smallest integer which is no less than $a$; $\lfloor a \rfloor$ denotes the floor function, which gives the largest integer which is no larger than $a$. For any scalars $a$ and $b$, $a \vee b$ denotes $\max(a, b)$ and $a \wedge b$ denotes $\min(a, b)$. For a vector or a matrix, $\|\cdot\|_0$ denotes the number of nonzero entries and $\|\cdot\|_\infty$ denotes the maximum magnitude of entries. Given a function $f : \mathbb{R}^D \to \mathbb{R}$ and a multi-index $s = [s_1, \cdots, s_D]^\top$, $\partial^s f$ denotes $\frac{\partial^{|s|} f}{\partial x_1^{s_1} \cdots \partial x_D^{s_D}}$. $\|f\|_{L^p}$ denote the $L^p$ norm of function $f$. We adopt the convention $0/0 = 0$. Given distributions $p$ and $q$, if $p$ is absolutely continuous with respect to $q$, the Pearson $\chi^2$-divergence is defined as $\chi^2(p, q) := \mathbb{E}_q[(\frac{\mathrm{d}p}{\mathrm{d}q} - 1)^2]$.

## 2 PRELIMINARIES

### 2.1 TIME-INHOMOGENEOUS MARKOV DECISION PROCESS

We consider a finite-horizon time-inhomogeneous Markov Decision Process (MDP) $(\mathcal{S}, \mathcal{A}, \{P_h\}_{h=1}^H, \{R_h\}_{h=1}^H, H, \xi)$, where $\xi$ is the initial state distribution. At time step $h = 1, \cdots, H$, from a state $s$ in the state space $\mathcal{S}$, we may choose action $a$ from the action space $\mathcal{A}$ and transition to a random next state $s' \in \mathcal{S}$ according to the transition probability distribution $s' \sim P_h(\cdot \mid s, a)$. Then, the system generates a random scalar reward $r \sim R_h(s, a)$ with $r \in [0, 1]$. We denote the mean of $R_h(s, a)$ by $r_h(s, a)$.

A policy $\pi = \{\pi_h\}_{h=1}^H$ specifies a set of $H$ distributions $\pi_h(\cdot \mid s)$ for choosing actions at every state $s \in \mathcal{S}$ and time step $h$. Given a policy $\pi$, the state-action value function, also known as the *Q-function*, for $h = 1, 2, \cdots, H$, is defined as

$$Q_h^\pi(s, a) := \mathbb{E}^\pi \bigg[ \sum_{h'=h}^H r_{h'}(s_{h'}, a_{h'}) \,\Big|\, s_h = s, a_h = a \bigg],$$

where $a_{h'} \sim \pi_{h'}(\cdot \mid s_{h'})$ and $s_{h'+1} \sim P_{h'}(\cdot \mid s_{h'}, a_{h'})$. Moreover, let $q_h^\pi$ denote the *state-action visitation distribution* of $\pi$ at step $h$, i.e., $q_h^\pi(s, a) := \mathbb{P}^\pi[s_h = s, a_h = a \mid s_1 \sim \xi]$.

For notational ease, we denote $\mathcal{X} := \mathcal{S} \times \mathcal{A}$. Let $\mathcal{P}_h^\pi : \mathbb{R}^\mathcal{X} \to \mathbb{R}^\mathcal{X}$ denote the *conditional transition operator* at step $h$:

$$\mathcal{P}_h^\pi f(s, a) := \mathbb{E}[f(s', a') \mid s, a], \ \forall f : \mathcal{X} \to \mathbb{R},$$

where $a' \sim \pi_h(\cdot \mid s')$ and $s' \sim P_h(\cdot \mid s, a)$.

Denote the *Bellman operator* at time $h$ under policy $\pi$ as $\mathcal{T}_h^\pi$:

$$\mathcal{T}_h^\pi f(s, a) := r_h(s, a) + \mathcal{P}_h^\pi f(s, a), \ \forall f : \mathcal{X} \to \mathbb{R}.$$

The Bellman equation may be written as $Q_h^\pi = \mathcal{T}_h^\pi Q_{h+1}^\pi$.

## 2.2 RIEMANNIAN MANIFOLD

Let $\mathcal{M}$ be a $d$-dimensional Riemannian manifold isometrically embedded in $\mathbb{R}^D$. A *chart* for $\mathcal{M}$ is a pair $(U, \phi)$ such that $U \subset \mathcal{M}$ is open and $\phi : U \to \mathbb{R}^d$ is a homeomorphism, i.e., $\phi$ is a bijection, its inverse and itself are continuous. Two charts $(U, \phi)$ and $(V, \psi)$ are called $\mathcal{C}^k$ *compatible* if and only if

$$\phi \circ \psi^{-1} : \psi(U \cap V) \to \phi(U \cap V) \quad \text{and} \quad \psi \circ \phi^{-1} : \phi(U \cap V) \to \psi(U \cap V)$$

are both $\mathcal{C}^k$ functions ($k$-th order continuously differentiable). A $\mathcal{C}^k$ atlas of $\mathcal{M}$ is a collection of $\mathcal{C}^k$ compatible charts $\{(U_i, \phi_i)\}$ such that $\bigcup_i U_i = \mathcal{M}$. An atlas of $\mathcal{M}$ contains an open cover of $\mathcal{M}$ and mappings from each open cover to $\mathbb{R}^d$.

**Definition 1** (Smooth manifold). A manifold $\mathcal{M}$ is smooth if it has a $\mathcal{C}^\infty$ atlas.

We introduce the reach [19, 53] of a manifold to characterize the curvature of $\mathcal{M}$.

**Definition 2** (Reach, Definition 2.1 in [1]). The medial axis of $\mathcal{M}$ is defined as $\overline{\mathcal{T}}(\mathcal{M})$, which is the closure of

$$\mathcal{T}(\mathcal{M}) = \{x \in \mathbb{R}^D \mid \exists\, x_1 \neq x_2 \in \mathcal{M}, \text{ such that } \|x - x_1\|_2 = \|x - x_2\|_2 = \inf_{y \in \mathcal{M}} \|x - y\|_2\}.$$

The reach $\omega$ of $\mathcal{M}$ is the minimum distance between $\mathcal{M}$ and $\overline{\mathcal{T}}(\mathcal{M})$, i.e.

$$\omega = \inf_{x \in \overline{\mathcal{T}}(\mathcal{M}), y \in \mathcal{M}} \|x - y\|_2.$$

Roughly speaking, reach measures how fast a manifold "bends". A manifold with a large reach "bends" relatively slowly. On the contrary, a small $\omega$ signifies more complicated local geometric structures, which are possibly hard to fully capture.

## 2.3 BESOV FUNCTIONS ON SMOOTH MANIFOLD

Through the concept of atlas, we are able to define Besov space on a smooth manifold.

**Definition 3** (Modulus of Smoothness [12]). Let $\Omega \subset \mathbb{R}^D$. For a function $f : \mathbb{R}^D \to \mathbb{R}$ be in $L^p(\Omega)$ for $p > 0$, the $r$-th modulus of smoothness of $f$ is defined by

$$w_{r,p}(f, t) = \sup_{\|h\|_2 \leq t} \|\Delta_h^r(f)\|_{L^p}, \text{ where}$$

$$\Delta_h^r(f)(x) = \begin{cases} \sum_{j=0}^r \binom{r}{j}(-1)^{r-j} f(x + jh) & \text{if } x \in \Omega, x + rh \in \Omega, \\ 0 & \text{otherwise.} \end{cases}$$

**Definition 4** (Besov functions on $\mathcal{M}$). Let $\mathcal{M}$ be a compact manifold of dimension $d$ with a finite atlas $\{(U_i, \phi_i)\}$. A function $f : \mathcal{M} \mapsto \mathbb{R}$ belongs to the Besov space $\mathcal{B}_{p,q}^\alpha(\mathcal{M})$, if on any chart $U_i$, it holds

$$\|f\|_{\mathcal{B}_{p,q}^\alpha(U_i)} := \|f\|_{L^p(U_i)} + \|f\|_{\mathcal{B}_{p,q}^\alpha(U_i)} < \infty,$$

where

$$\|f\|_{\mathcal{B}_{p,q}^\alpha(U_i)} := \begin{cases} \left(\int_0^\infty (t^{-\alpha} w_{\lfloor\alpha\rfloor+1,p}(f,t))^q \frac{\mathrm{d}t}{t}\right)^{1/q} & \text{if } q < \infty, \\ \sup_{t>0} t^{-\alpha} w_{\lfloor\alpha\rfloor+1,p}(f,t) & \text{if } q = \infty. \end{cases}$$

Further, the Besov norm of $f$ is defined as $\|f\|_{\mathcal{B}_{p,q}^\alpha(\mathcal{M})} = \sum_i \|f\|_{\mathcal{B}_{p,q}^\alpha(U_i)}$. We occasionally omit $\mathcal{M}$ in the Besov norm when it is clear from the context.

## 2.4 CONVOLUTIONAL NEURAL NETWORK

We consider one-sided stride-one convolutional neural networks (CNNs) with rectified linear unit (ReLU) activation function ($\mathrm{ReLU}(z) = \max(z, 0)$). Specifically, a CNN we consider consists of a padding layer, several convolutional blocks, and finally a fully connected output layer.

Given an input vector $x \in \mathbb{R}^D$, the network first applies a padding operator $P : \mathbb{R}^D \to \mathbb{R}^{D \times C}$ for some integer $C \geq 1$ such that

$$Z = P(x) = [x \quad 0 \quad \cdots \quad 0] \in \mathbb{R}^{D \times C}.$$

Then the matrix $Z$ is passed through $M$ convolutional blocks. We will denote the input matrix to the $m$-th block as $Z_m$ and its output as $Z_{m+1}$ (i.e., $Z_1 = Z$).

Let us define the convolution operation in Equation (2). Let $\mathcal{W} = \{\mathcal{W}_{j,i,l}\} \in \mathbb{R}^{C' \times I \times C}$ be a filter where $C'$ is the output channel size, $I$ is the filter size and $C$ is the input channel size. For $Z \in \mathbb{R}^{D \times C}$, the convolution of $\mathcal{W}$ with $Z$, denoted with $\mathcal{W} * Z$, results in $Y \in \mathbb{R}^{D \times C'}$ with

$$Y_{k,j} = \sum_{i=1}^I \sum_{l=1}^C \mathcal{W}_{j,i,l} Z_{k+i-1,l},$$

where we set $Z_{k+i-1,l} = 0$ for $k + i - 1 > D$.

Figure 2: Convolution of $\mathcal{W} * Z$. $\mathcal{W}_{j,:,:}$ is a $I \times C$ matrix for the $j$-th output channel.

In the $m$-th convolutional block, let $\mathcal{W}_m = \{\mathcal{W}_m^{(1)}, ..., \mathcal{W}_m^{(L_m)}\}$ and $\mathcal{B}_m = \{B_m^{(1)}, ..., B_m^{(L_m)}\}$ be a collection of filters and biases of proper sizes. The $m$-th block maps its input matrix $Z_m \in \mathbb{R}^{D \times C}$ to $Z_{m+1} \in \mathbb{R}^{D \times C}$ by

$$Z_{m+1} = \mathrm{ReLU}\left(\mathcal{W}_m^{(L_m)} * \cdots * \mathrm{ReLU}\left(\mathcal{W}_m^{(1)} * Z_m + B_m^{(1)}\right) \cdots + B_m^{(L_m)}\right) \tag{2}$$

with ReLU applied entrywise. For notational simplicity, we denote this series of operations in the $m$-th block with a single operator from $\mathbb{R}^{D \times C}$ to $\mathbb{R}^{D \times C}$ with $\mathrm{Conv}_{\mathcal{W}_m, \mathcal{B}_m}$, so (2) can be abbreviated as

$$Z_{m+1} = \mathrm{Conv}_{\mathcal{W}_m, \mathcal{B}_m}(Z_m).$$

Overall, we denote the mapping from input $x$ to the output of the $M$-th convolutional block as

$$G(x) = (\mathrm{Conv}_{\mathcal{W}_M, \mathcal{B}_M}) \circ \cdots \circ (\mathrm{Conv}_{\mathcal{W}_1, \mathcal{B}_1}) \circ P(x). \tag{3}$$

Given (3), a CNN applies an additional fully connected layer to $G$ and outputs

$$f(x) = W \otimes G(x) + b,$$

where $W \in \mathbb{R}^{D \times C}$ and $b \in \mathbb{R}$ are a weight matrix and a bias, respectively, and $\otimes$ denotes sum of entrywise product, i.e. $W \otimes G(x) = \sum_{i,j} W_{i,j} [G(x)]_{i,j}$. Thus, we define a class of CNNs of the same architecture as

$$\mathcal{F}(M, L, J, I, \tau_1, \tau_2) =$$

$$\{f \mid f(x) = W \otimes Q(x) + b \text{ with } \|W\|_\infty \vee |b| \leq \tau_2, G(x) \text{ in the form of (3) with } M \text{ blocks.}$$

The number of filters per block is bounded by $L$; filter size is bounded by $I$; the number of

channels is bounded by $J$; $\max_{m,l} \|\mathcal{W}_m^{(l)}\|_\infty \vee \|B_m^{(l)}\|_\infty \leq \tau_1\}. \tag{4}$

Furthermore, $\mathcal{F}(M, L, J, I, \tau_1, \tau_2, V)$ is defined as

$$\mathcal{F}(M, L, J, I, \tau_1, \tau_2, V) = \{ f \in \mathcal{F}(M, L, J, I, \tau_1, \tau_2) \mid \|f\|_{L^\infty} \leq V \}. \quad (5)$$

**Universal Approximation of Neural Networks** There exists rich literature about using neural networks to approximate functions supported on compact domain in Euclidean space, from early asymptotic results [34, 23, 10, 33] to later quantitative results [3, 46, 44, 30, 71]. These results suffer from the curse of dimensionality, in that to approximate a function up to certain error, the network size grows exponentially in the data dimension. Recently, a line of work advances the approximation theory of neural network to functions supported on domains with intrinsic geometric structures [8, 57, 60, 58, 7, 42]. They show that neural networks are adaptive to the intrinsic structures in data, suggesting that to approximate a function up to certain error, it suffices to choose the network size only depending on the intrinsic dimension, which is often much smaller than the representation dimension of the data. In addition to existing results, we prove a novel universal approximation theory of CNN en route to our RL result.

## 3 NEURAL FITTED Q-EVALUATION

We consider the off-policy evaluation (OPE) problem of a finite-horizon time-inhomogeneous MDP. The transition model $\{P_h\}_{h=1}^H$ and reward function $\{R_h\}_{h=1}^H$ are unknown, and we are only given the access to an unknown *behavior policy* $\pi_0$ to generate experience data from the MDP. Our objective is to evaluate the value of $\pi$ from a fixed initial distribution $\xi$ over horizon $H$, given by

$$v^\pi := \mathbb{E}^\pi \left[ \sum_{h=1}^H r_h(s_h, a_h) \,\Big|\, s_1 \sim \xi \right],$$

where $a_h \sim \pi(\cdot \mid s_h)$ and $s_{h+1} \sim P_h(\cdot \mid s_h, a_h)$.

At time-step $h$, we generate data $\mathcal{D}_h := \{(s_{h,k}, a_{h,k}, s'_{h,k}, r_{h,k})\}_{k=1}^K$. Specifically, $\{s_{h,k}\}_{k=1}^K$ are i.i.d. samples from some state distribution. For each $s_{h,k}$, we use the unknown behavior policy $\pi_0$ to generate $a_{h,k} \sim \pi_0(\cdot \mid s_{h,k})$. More generally, we may view $\{(s_{h,k}, a_{h,k})\}_{k=1}^K$ as i.i.d. samples from a sampling distribution $q_h^{\pi_0}$. For each $(s_{h,k}, a_{h,k})$, we can further generate $s'_{h,k} \sim P_h(\cdot \mid s_{h,k}, a_{h,k})$ and $r_{h,k} \sim R_h(s_{h,k}, a_{h,k})$ independently for each $k$.

This assumption on data generation is the time-inhomogeneous analog of a standard data assumption in time-homogeneous OPE, which assumes all data are i.i.d. samples from the same sampling distribution [51, 74, 50, 69]. Moreover, our dataset is similar to one that comprises $K$ independent episodes generated by the behavior policy, as [41] introduce a subroutine whereby one can process an episodic dataset and treat it as an i.i.d.-sampled dataset in any downstream algorithm.

To estimate $v^\pi$, neural FQE estimates $Q_h^\pi$ in a backward, recursive fashion. $\widehat{Q}_h^\pi$, our estimate at step $h$, is taken as $\widehat{\mathcal{T}}_h^\pi \left( \widehat{Q}_{h+1}^\pi \right)$, whose update rule is based on the Bellman equation:

$$\widehat{\mathcal{T}}_h^\pi \left( \widehat{Q}_{h+1}^\pi \right) = \arg\min_{f \in \mathcal{F}} \sum_{k=1}^K \left( f(s_{h,k}, a_{h,k}) - r_{h,k} - \int_{\mathcal{A}} \widehat{Q}_{h+1}^\pi(s'_{h,k}, a) \pi_h(a \mid s'_{h,k}) \, \mathrm{d}a \right)^2, \quad (6)$$

where $\widehat{\mathcal{T}}_h^\pi$ is an intermediate estimate of the Bellman operator $\mathcal{T}_h^\pi$, $\widehat{Q}_{h+1}^\pi$ is an intermediate estimate of $Q_{h+1}^\pi$, and $\mathcal{F}$ denotes a class of convolutional neural networks as specified in (5) with proper hyperparameters. The pseudocode for our algorithm is presented in Algorithm 1.

## 4 MAIN RESULTS

In this section, we prove an upper bound on the estimation error of $v^\pi$ by Algorithm 1. First, let us state two assumptions on our MDP of interest.

**Assumption 1** (Low-dimensional state-action space). *The state-action space $\mathcal{X}$ is a $d$-dimensional compact Riemannian manifold isometrically embedded in $\mathbb{R}^D$. There exists $B > 0$ such that $\|x\|_\infty \leq B$ for any $x \in \mathcal{X}$. The reach of $\mathcal{X}$ is $\omega > 0$.*

Assumption 1 characterizes the low-dimensional structures of the MDP represented in high dimensions. We say that the "*intrinsic dimension*" of $\mathcal{X}$ is $d \ll D$. Such a setting, as mentioned in Section 1, is common in practice, because the representation or feature people have access to are often excessive compared to the latent structures of the problem. For instance, images of a dynamical system

---

**Algorithm 1** Neural Fitted Q-Evaluation (Neural-FQE)

---

**Input:** Initial distribution $\xi$, target policy $\pi$, horizon $H$, effective sample size $K$, function class $\mathcal{F}$.
**Init:** $\widehat{Q}^{\pi}_{H+1} := 0$
**for** $h = H, H-1, \cdots, 1$ **do**
    Sample $\mathcal{D}_h = \{(s_{h,k}, a_{h,k}, s'_{h,k}, r_{h,k})\}^{K}_{k=1}$.
    Update $\widehat{Q}^{\pi}_h \leftarrow \widehat{\mathcal{T}}^{\pi}_h \left( \widehat{Q}^{\pi}_{h+1} \right)$ by (6).
**end for**
**Output:** $\widehat{v}^{\pi} := \int_{\mathcal{X}} \widehat{Q}^{\pi}_1(s,a)\xi(s)\pi(a \mid s) \, \mathrm{d}s \, \mathrm{d}a$.

---

are widely believed to admit such low-dimensional latent structures [26, 32, 55]. People often take the visual display of a computer game as its state representations, which are in pixels, but computer only keeps a small number of parameters internally to represent the state of the game.

**Assumption 2** (Bellman completeness). *Under target policy $\pi$, for any time step $h$ and any $f \in \mathcal{F}$, we have $\mathcal{T}^{\pi}_h f \in \mathcal{B}^{\alpha}_{p,q}(\mathcal{X})$, where $0 < p, q \leq \infty$ and $d/p + 1 \leq \alpha < \infty$. Moreover, there exists a constant $c_0 > 0$ that satisfies $\|\mathcal{T}^{\pi}_h f\|_{\mathcal{B}^{\alpha}_{p,q}(\mathcal{X})} \leq c_0$ for any time step $h$.*

Bellman completeness assumption is about the closure of a function class under the Bellman operator. It has been widely adopted in RL literature [70, 14, 6, 69]. Some classic MDP settings implicitly possess this property, e.g., linear MDP [36]. Note that [66] show the necessity of such an assumption on the Bellman operator to regulate the Bellman residual: without such assumption, even in the simple setting of linear function approximation with realizability, to solve OPE up to a constant error, the lower bound on sample complexity is exponential in horizon.

The Besov family contains a large class of smooth functions, and has been widely adopted in existing nonparametric statistics literature for various problems [24, 62, 63]. For MDP, Assumption 2 holds for most common smooth dynamics, as long as certain regularity conditions on smoothness are satisfied. For instance, [51] show a simple yet sufficient condition, under which for any time step $h$ and $s' \in \mathcal{S}$, the reward function $r_h(s,a)$ and the transition kernel $P_h(s'|s,a)$ are functions in $\mathcal{B}^{\alpha}_{p,q}$. This further implies $Q^{\pi}_0, \cdots, Q^{\pi}_H \in \mathcal{B}^{\alpha}_{p,q}(\mathcal{X})$. In addition, Assumption 2 may be satisfied even when the transition kernel is not smooth, examples of which are provided in [18].

Note that while most existing work on function approximation assumes Bellman completeness with respect to the function approximator, which in our work is deep convolutional neural networks with ReLU activation, we are only concerned with the closure of the Besov class $\mathcal{B}^{\alpha}_{p,q}(\mathcal{X})$ under the Bellman operator. This assumption is weaker than the previous work [75], which considers smooth function approximation (excluding ReLU networks).

Our main result is summarized in Theorem 2, which relies on using CNNs to accurately represent $\mathcal{T}^{\pi}_h f$ for any $f \in \mathcal{F}$. The following theorem provides a novel quantitative analysis on how to properly choose CNN classes for approximating $\mathcal{T}^{\pi}_h f$ depending on the regularity of $\mathcal{T}^{\pi}_h$.

**Theorem 1.** *Suppose Assumption 1 and 2 hold. For any positive integers $I \in [2, D]$ and $\widetilde{M}, \widetilde{J} > 0$, we let*

$$L = O\left(\log(\widetilde{M}\widetilde{J}) + D + \log D\right), \ J = O(D\widetilde{J}), \ \tau_1 = (8ID)^{-1}\widetilde{M}^{-\frac{1}{L}} = O(1),$$

$$\log \tau_2 = O\left(\log^2 \widetilde{M}\widetilde{J} + D\log \widetilde{M}\widetilde{J}\right), \ M = O(\widetilde{M}), \tag{7}$$

*Then CNN class $\mathcal{F}(M, L, J, I, \tau_1, \tau_2)$ in (4) can approximate $\mathcal{T}^{\pi}_h f$ for any $f \in \mathcal{F}(M, L, J, I, \tau_1, \tau_2)$ and $h = 1, \cdots, H$, i.e., there exists $\widehat{f} \in \mathcal{F}(M, L, J, I, \tau_1, \tau_2)$ with*

$$\|\widehat{f} - \mathcal{T}^{\pi}_h f\|_{\infty} \leq (\widetilde{M}\widetilde{J})^{-\frac{\alpha}{d}}. \tag{8}$$

$O(\cdot)$ *hides a constant depending on $d$, $\alpha$, $\frac{2d}{\alpha p - d}$, $p$, $q$, $\|\mathcal{T}^{\pi}_h f\|_{\mathcal{B}^{\alpha}_{p,q}(\mathcal{X})}$, $B$, $\omega$ and surface area of $\mathcal{X}$.*

The proof is provided in Appendix B. As can be seen, the rate of approximation is free of the curse of ambient dimension $D$. We remark that Theorem 1 allows an arbitrary rescaling of $\widetilde{M}$ and $\widetilde{J}$, as only their product is relevant to the approximation error. This is more flexible than conventional

approximation theories [7, 54, 51], where the network width and depth have to maintain a fixed ratio in terms of the desired approximation error. In Theorem 2, we choose a configuration of $\widetilde{M}$ and $\widetilde{J}$ that leads to the optimal statistical rate via a bias-variance tradeoff argument.

**Theorem 2.** *Suppose Assumption 1 and 2 hold. By choosing*

$$L = O(\log K + D + \log D), \ J = O(D), \ \tau_1 = O(1),$$

$$\log \tau_2 = O(\log^2 K + D \log K), \ M = O(K^{\frac{d}{2\alpha+d}}), \ V = H \tag{9}$$

*with any integer $I \in [2, D]$ in Algorithm 1, in which $O(\cdot)$ hides factors depending on $d$, $\alpha$, $\frac{2d}{\alpha p - d}$, $p$, $q$, $c_0$, $B$, $\omega$, and the surface area of $\mathcal{X}$, we have*

$$\mathbb{E} |v^\pi - \widehat{v}^\pi| \leq CH^2 \kappa K^{-\frac{\alpha}{2\alpha+d}} \log^{\frac{5}{2}} K, \tag{10}$$

*in which the expectation is taken over the data, and $C$ is a constant depending on $D^{\frac{3\alpha}{2\alpha+d}}$, $d$, $\alpha$, $\frac{d}{\alpha p - d}$, $p$, $q$, $c_0$, $B$, $\omega$ and the surface area of $\mathcal{X}$. The distributional mismatch is captured by*

$$\kappa = \frac{1}{H} \sum_{h=1}^{H} \sqrt{\chi_{\mathcal{Q}}^2(q_h^\pi, q_h^{\pi_0}) + 1},$$

*in which $q_h^\pi$ and $q_h^{\pi_0}$ are the visitation distributions of $\pi$ and $\pi_0$ at step $h$ respectively and $\mathcal{Q}$ is the Minkowski sum between the CNN function class in (5) and the Besov function class, i.e., $\mathcal{Q} = \{f + g \mid f \in \mathcal{B}_{p,q}^\alpha(\mathcal{X}), g \in \mathcal{F}(M, L, J, I, \tau_1, \tau_2, V)\}$.*

We next compare our Theorem 2 with existing work:

**(I) Tight characterization of distributional mismatch**. The term $\kappa$ depicts the distributional mismatch between the target policy's visitation distribution and data coverage via restricted $\chi^2$-divergence. Note that the restricted $\chi^2$-divergence is always no larger than the commonly-used absolute density ratio [51, 6, 67] and can often be substantially smaller. This is because probability measures $q_h^\pi$ and $q_h^{\pi_0}$ might differ a lot over some small regions in the sample space, while their integrations of a smooth function in $\mathcal{Q}$ over the entire sample space could be close to each other. The absolute density ratio measures the former and restricted $\chi^2$-divergence measures the latter.

More strikingly, when considering function approximation (e.g. state-action space is not countably finite), the restricted $\chi^2$-divergence can still remain small even when absolute density ratio becomes unbounded. For example, we consider two isotropic multivariate Gaussian distributions with different means. [52] has shown that Pearson $\chi^2$-divergence, which is always larger than or equal to restricted $\chi^2$-divergence, has a finite expression:

$$\chi^2 \left( \mathcal{N}(\mu_1, I), \mathcal{N}(\mu_2, I) \right) = e^{\|\mu_1 - \mu_2\|_2^2} - 1,$$

whereas one may find the absolute density ratio unbounded: for any $\mu_1 \neq \mu_2$,

$$\left\| \frac{d\mathcal{N}(\mu_1, I)}{d\mathcal{N}(\mu_2, I)} \right\|_\infty = \sup_x \exp \left( x^\top (\mu_1 - \mu_2) - \frac{1}{2} \|\mu_1\|^2 + \frac{1}{2} \|\mu_2\|^2 \right) = \infty.$$

Such a stark comparison can also be observed in other common distributions that have support with infinite cardinality, e.g. Poisson distribution.

Furthermore, when the state-action space exhibits small intrinsic dimensions, i.e., $d \ll D$, the restricted $\chi^2$-divergence adapts to such low-dimensional structure and characterizes the distributional mismatch with respect to $\mathcal{Q}$, which is a small function class depending on the intrinsic dimension. In contrast, the absolute density ratio in [51] does not take advantage of the low-dimensional structure.

In summary, though the absolute density ratio is a tight in the tabular setting and some other special classes of MDPs, in the general function approximation setting, it could easily become intractably vacuous, and restricted $\chi^2$-divergence is tighter characterization of distributional mismatch.

**(II) Adaptation to intrinsic dimension**. Note that our estimation error is dominated by the intrinsic dimension $d$, rather than the representation dimension $D$. Therefore, it is significantly smaller than the error of methods oblivious to the problem's intrinsic dimension such as [51].

Such a fast convergence owes to the adaptability of neural networks to the manifold structure in the state-action space. With properly chosen width and depth, the neural network automatically captures local geometries on the manifold through the empirical risk minimization in Algorithm 1 for approximating Q-functions.

***Sample Complexity Comparison.*** Given a pre-specified estimation error of policy value $\epsilon$, our algorithm requires a sample complexity of
$$\widetilde{O}(H^{4+2d/\alpha}\kappa^{2+d/\alpha}\epsilon^{-2-d/\alpha}).$$
We next compare our result with [51], which among existing work is the most similar to ours. Specifically, we reprove our result with feed-forward ReLU network so as to be in the same setting as [51] (details in Theorem 3 of Appendix E).

When the experience data are allowed to be reused, they show a sample complexity of
$$\widetilde{O}(H^{2+2D/\alpha}\kappa_3^{2+2D/\alpha}\epsilon^{-2-2D/\alpha}).$$
As can be seen, our result is more efficient than theirs as long as $H^{1-\frac{D-d}{\alpha}} \le \epsilon^{-1-\frac{D-d/2}{\alpha}}$. Such a requirement of the horizon can be satisfied in real applications, as $d \ll D$ and $\alpha$ is moderate. Note that even with no consideration for low-dimensional structures, i.e., $d = D$, our result is still more efficient, as $\kappa$ is often substantially smaller than $\kappa_3$. Moreover, when the experience data are used just for one pass, our method is instantly more efficient, as their sample complexity becomes
$$\widetilde{O}(H^{4+2D/\alpha}\kappa_3^{2+D/\alpha}\epsilon^{-2-D/\alpha}).$$

## 5 EXPERIMENTS

We present numerical experiments for evaluating FQE with CNN function approximation on the classic CartPole environment [4]. The CartPole problem has a $4$-dimensional continuous intrinsic state space. We consider a finite-horizon MDP with horizon $H = 100$ in this environment. In our experiments, we solve the OPE problem with FQE (Algorithm 1). We take the visual display of the environment as states. These images serve as a high-dimensional representation of CartPole's original $4$-dimensional continuous state space. In our algorithm, we use a deep CNN to approximate the Q-functions and solve the regression with SGD (see Appendix F.1 for details).

Table 2: Value estimation $\widehat{v}^\pi$ under high resolution and low resolution. The true $v^\pi \approx 65.2$ is computed via Monte Carlo rollout.

| Sample size $K$ | (A) No distribution shift | | (B) Off-policy | |
| --- | --- | --- | --- | --- |
| | High res | Low res | High res | Low res |
| 5000 | $64.6 \pm 2.0$ | $63.5 \pm 1.9$ | $60.4 \pm 2.8$ | $60.0 \pm 3.3$ |
| 10000 | $66.0 \pm 1.3$ | $66.5 \pm 1.7$ | $67.0 \pm 1.8$ | $68.0 \pm 2.3$ |
| 20000 | $65.1 \pm 1.0$ | $65.1 \pm 1.2$ | $65.0 \pm 1.6$ | $65.1 \pm 2.0$ |

We consider two settings with different visual resolutions (see Appendix F.1 for details): one in high resolution (dimension $3 \times 40 \times 150$) and the other in low resolution (dimension $3 \times 20 \times 75$). We use a policy trained for 200 iterations with REINFORCE as the target policy. We conduct this experiment in two cases: (A) data are generated from the target policy itself; (B) data are generated from a mixture policy of $0.8$ target policy and $0.2$ uniform distribution. (A) aims to verify the performance's dependence on data intrinsic dimension without the influence from distribution shift.

We observe that the performance of FQE on high-resolution and low-resolution data is similar, in both the off-policy case and the easier case with no distribution shift. It shows that the estimation error of FQE takes little influence from the representation dimension of the data but rather from the intrinsic structure of the environment, which is the same regardless of resolution. We also observe that the estimation becomes increasingly accurate as sample size $K$ increases. These empirical results confirm our upper bound in Theorem 2, which is only dominated by data intrinsic dimension.

## 6 CONCLUSION

This paper studies nonparametric off-policy evaluation in MDPs. We use CNNs to approximate Q-functions. Our theory proves that when state-action space exhibits low-dimensional structures, the finite-sample estimation error of FQE converges depending on the intrinsic dimension. In the estimation error, the distribution mismatch between the data distribution and target policy's visitation distribution is quantified by a restricted $\chi^2$-divergence term, which is oftentimes much smaller than the absolute density ratio. Our theory also reassures practitioners of the benignity of overrepresentation in deep RL and provides insights into how to choose network hyperparameters properly in presence of low intrinsic dimension. We support our theory with experiments. For future directions, it would be of interest to adapt this low-dimensional analysis to time-homogeneous MDPs. It is nontrivial to preserve the error rate with sample reuse in the presence of temporal dependency.

ACKNOWLEDGEMENT

Mengdi Wang acknowledges the support by NSF grants DMS-1953686, IIS-2107304, CMMI-1653435, ONR grant 1006977, and C3.AI. Tuo Zhao acknowledges the support by DMS-2012652.

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

## A    PROOF OF THEOREM 2

In this section, we provide a proof for the upper bound on the estimation error in Theorem 2. We can tackle the sequential dependencies by recursively conditioning on the previous-step estimation and the fact that the error in the previous steps accumulates linearly. The estimation error can be decomposed into a sum of statistical error and approximation error. A tradeoff exists about the network size: while a larger network reduces the approximation error, it leads to higher variance in the statistical error. Consequently, we choose the network size and architecture appropriately to balance the two types of error, which in turn minimizes the final estimation error.

*Proof of Theorem 2.* The goal is to bound

$$\mathbb{E} \, |\widehat{v}^{\pi} - v^{\pi}| = \mathbb{E} \left| \int_{\mathcal{X}} \left( Q_1^{\pi} - \widehat{Q}_1^{\pi} \right)(s,a) \, \mathrm{d}q_1^{\pi}(s,a) \right| \leq \mathbb{E} \left[ \int_{\mathcal{X}} \left| Q_1^{\pi} - \widehat{Q}_1^{\pi} \right|(s,a) \, \mathrm{d}q_1^{\pi}(s,a) \right].$$

To get an expression for that, we first expand it recursively. To illustrate the recursive relation, we examine the quantity at step $h$:

$$\mathbb{E} \left[ \int_{\mathcal{X}} \left| Q_h^{\pi} - \widehat{Q}_h^{\pi} \right|(s,a) \, \mathrm{d}q_h^{\pi}(s,a) \right]$$

$$= \mathbb{E} \left[ \int_{\mathcal{X}} \left| \mathcal{T}_h^{\pi} Q_{h+1}^{\pi} - \widehat{\mathcal{T}}_h^{\pi} \left( \widehat{Q}_{h+1}^{\pi} \right) \right|(s,a) \, \mathrm{d}q_h^{\pi}(s,a) \right]$$

$$\leq \mathbb{E} \left[ \int_{\mathcal{X}} \left| \mathcal{T}_h^{\pi} Q_{h+1}^{\pi} - \mathcal{T}_h^{\pi} \widehat{Q}_{h+1}^{\pi} \right|(s,a) \, \mathrm{d}q_h^{\pi}(s,a) \right] + \mathbb{E} \left[ \int_{\mathcal{X}} \left| \mathcal{T}_h^{\pi} \widehat{Q}_{h+1}^{\pi} - \widehat{\mathcal{T}}_h^{\pi} \left( \widehat{Q}_{h+1}^{\pi} \right) \right|(s,a) \, \mathrm{d}q_h^{\pi}(s,a) \right]$$

$$= \mathbb{E} \left[ \int_{\mathcal{X}} \left| Q_{h+1}^{\pi} - \widehat{Q}_{h+1}^{\pi} \right|(s,a) \, \mathrm{d}q_{h+1}^{\pi}(s,a) \right]$$

$$\quad + \mathbb{E} \left[ \mathbb{E} \left[ \int_{\mathcal{X}} \left| \mathcal{T}_h^{\pi} \widehat{Q}_{h+1}^{\pi} - \widehat{\mathcal{T}}_h^{\pi} \left( \widehat{Q}_{h+1}^{\pi} \right) \right|(s,a) \, \mathrm{d}q_h^{\pi}(s,a) \mid \mathcal{D}_{h+1}, \cdots, \mathcal{D}_H \right] \right]$$

$$\overset{(a)}{\leq} \mathbb{E} \left[ \int_{\mathcal{X}} \left| Q_{h+1}^{\pi} - \widehat{Q}_{h+1}^{\pi} \right|(s,a) \, \mathrm{d}q_{h+1}^{\pi}(s,a) \right]$$

$$\quad + \mathbb{E} \left[ \mathbb{E} \left[ \sqrt{\int_{\mathcal{X}} \left( \mathcal{T}_h^{\pi} \widehat{Q}_{h+1}^{\pi} - \widehat{\mathcal{T}}_h^{\pi} \left( \widehat{Q}_{h+1}^{\pi} \right) \right)^2 (s,a) \, \mathrm{d}q_h^{\pi_0}(s,a)} \sqrt{\chi_{\mathcal{Q}}^2(q_h^{\pi}, q_h^{\pi_0}) + 1} \mid \mathcal{D}_{h+1}, \cdots, \mathcal{D}_H \right] \right]$$

$$\overset{(b)}{\leq} \mathbb{E} \left[ \int_{\mathcal{X}} \left| Q_{h+1}^{\pi} - \widehat{Q}_{h+1}^{\pi} \right|(s,a) \, \mathrm{d}q_{h+1}^{\pi}(s,a) \right]$$

$$\quad + \sqrt{\mathbb{E} \left[ \mathbb{E} \left[ \int_{\mathcal{X}} \left( \mathcal{T}_h^{\pi} \widehat{Q}_{h+1}^{\pi} - \widehat{\mathcal{T}}_h^{\pi} \left( \widehat{Q}_{h+1}^{\pi} \right) \right)^2 (s,a) \, \mathrm{d}q_h^{\pi_0}(s,a) \mid \mathcal{D}_{h+1}, \cdots, \mathcal{D}_H \right] \right]} \sqrt{\chi_{\mathcal{Q}}^2(q_h^{\pi}, q_h^{\pi_0}) + 1}$$

$$\overset{(c)}{\leq} \int_{\mathcal{X}} \left| Q_{h+1}^{\pi} - \widehat{Q}_{h+1}^{\pi} \right|(s,a) \, \mathrm{d}q_{h+1}^{\pi}(s,a) + \sqrt{C'(5H^2) K^{-\frac{2\alpha}{2\alpha+d}} \log^5 K} \sqrt{\chi_{\mathcal{Q}}^2(q_h^{\pi}, q_h^{\pi_0}) + 1}$$

$$\leq \int_{\mathcal{X}} \left| Q_{h+1}^{\pi} - \widehat{Q}_{h+1}^{\pi} \right|(s,a) \, \mathrm{d}q_{h+1}^{\pi}(s,a) + CHK^{-\frac{\alpha}{2\alpha+d}} \log^{5/2} K \sqrt{\chi_{\mathcal{Q}}^2(q_h^{\pi}, q_h^{\pi_0}) + 1},$$

where $C$ denotes a (varying) constant depending on $D^{\frac{3\alpha}{2\alpha+d}}$, $d$, $\alpha$, $\frac{d}{\alpha p - d}$, $p$, $q$, $c_0$, $B$, $\omega$ and the surface area of $\mathcal{X}$.

In (a), note $\mathcal{T}_h^{\pi} \widehat{Q}_{h+1}^{\pi} \in \mathcal{B}_{p,q}^{\alpha}(\mathcal{X})$ by Assumption 2 and $-\widehat{\mathcal{T}}_h^{\pi} \left( \widehat{Q}_{h+1}^{\pi} \right) \in \mathcal{F}$ by our algorithm, so $\mathcal{T}_h^{\pi} \widehat{Q}_{h+1}^{\pi} - \widehat{\mathcal{T}}_h^{\pi} \left( \widehat{Q}_{h+1}^{\pi} \right) \in \mathcal{Q}$. Then we employ a change-of-measure argument and obtain this inequality by invoking the following lemma.

**Lemma 1.** Given a function class $\mathcal{Q}$ that contains functions mapping from $\mathcal{X}$ to $\mathbb{R}$ and two probability distributions $q_1$ and $q_2$ supported on $\mathcal{X}$, for any $g \in \mathcal{Q}$,

$$\mathbb{E}_{x \sim q_1}[g(x)] \leq \sqrt{\mathbb{E}_{x \sim q_2}[g^2(x)](1 + \chi_{\mathcal{Q}}^2(q_1, q_2))}.$$

*Proof of Lemma 1.*

$$\mathbb{E}_{x \sim q_1}[g(x)] = \sqrt{\mathbb{E}_{x \sim q_2}[g^2(x)] \frac{\mathbb{E}_{x \sim q_1}[g(x)]^2}{\mathbb{E}_{x \sim q_2}[g^2(x)]}}$$

$$\leq \sqrt{\mathbb{E}_{x \sim q_2}[g^2(x)] \sup_{f \in \mathcal{Q}} \frac{\mathbb{E}_{x \sim q_1}[f(x)]^2}{\mathbb{E}_{x \sim q_2}[f^2(x)]}}$$

$$= \sqrt{\mathbb{E}_{x \sim q_2}[g^2(x)](1 + \chi^2_{\mathcal{Q}}(q_1, q_2))},$$

where the last step is by the definition of $\chi^2_{\mathcal{Q}}(q_1, q_2) := \sup_{f \in \mathcal{Q}} \frac{\mathbb{E}_{q_1}[f]^2}{\mathbb{E}_{q_2}[f^2]} - 1$. $\qquad\square$

In (b), we use Jensen's inequality and the fact that square root is concave.

To obtain (c), we invoke Lemma 10, which provides an upper bound on the error of nonparametric regression at each step of the FQE algorithm.

Specifically, we will invoke Lemma 10 when conditioning on $\mathcal{D}_{h+1}, \cdots, \mathcal{D}_H$, i.e. the data from time step $h + 1$ to time step $H$. Note that after conditioning, $\mathcal{T}^\pi_h \widehat{Q}^\pi_{h+1}$ becomes measurable and deterministic with respect to $\mathcal{D}_{h+1}, \cdots, \mathcal{D}_H$. Also, $\mathcal{D}_{h+1}, \cdots, \mathcal{D}_H$ are independent from $\mathcal{D}_h$, which we use in the regression at step $h$.

To justify our use of this theorem, we need to cast our problem into a regression problem described in the theorem. Since $\{(s_{h,k}, a_{h,k})\}_{k=1}^K$ are i.i.d. from $q^{\pi_0}_h$, we can view them as the samples $x_i$'s in the lemma. We can view $\mathcal{T}^\pi_h \widehat{Q}^\pi_{h+1}$, which is measurable under our conditioning, as $f_0$ in the lemma. Furthermore, we let

$$\zeta_{h,k} := r_{h,k} + \int_{\mathcal{A}} \widehat{Q}^\pi_{h+1}(s'_{h,k}, a) \pi(a \mid s'_{h,k}) \, \mathrm{d}a - \mathcal{T}^\pi_h \widehat{Q}^\pi_{h+1}(s_{h,k}, a_{h,k}).$$

In order to invoke Lemma 10 under the conditioning on $\mathcal{D}_{h+1}, \cdots, \mathcal{D}_H$, we need to verify whether three conditions are satisfied (conditioning on $\mathcal{D}_{h+1}, \cdots, \mathcal{D}_H$):

1. Sample $\{(s_{h,k}, a_{h,k})\}_{k=1}^K$ are i.i.d;

2. Sample $\{(s_{h,k}, a_{h,k})\}_{k=1}^K$ and noise $\{\zeta_{h,k}\}_{k=1}^K$ are uncorrelated;

3. Noise $\{\zeta_{h,k}\}_{k=1}^K$ are independent, zero-mean, subgaussian random variables.

In our setting, $\{(s_{h,k}, a_{h,k})\}_{k=1}^K$ are i.i.d. from $q^{\pi_0}_h$. Due to the time-inhomogeneous setting, they are independent from $\mathcal{D}_{h+1}, \cdots, \mathcal{D}_H$, so $\{(s_{h,k}, a_{h,k})\}_{k=1}^K$ are still i.i.d. under our conditioning. Thus, Condition 1 is clearly satisfied.

We may observe that under our conditioning, the transition from $(s_{h,k}, a_{h,k})$ to $s'_{h,k}$ is the only source of randomness in $\zeta_{h,k}$, besides $(s_{h,k}, a_{h,k})$ itself. The distribution of $(s_{h,k}, a_{h,k}, s'_{h,k})$ is actually the product distribution between $P_h(\cdot | s_{h,k}, a_{h,k})$ and $q^{\pi_0}_h$, so a function of $s'_{h,k}$, generated from the transition distribution $P_h(\cdot | s_{h,k}, a_{h,k})$, is uncorrelated with $(s_{h,k}, a_{h,k})$. Thus, $(s_{h,k}, a_{h,k})$'s are uncorrelated with $\zeta_{h,k}$'s under our conditioning, and Condition 2 is satisfied.

Condition 3 can also be easily verified. Under our conditioning, the randomness in $\zeta_{h,k}$ only comes from $(s_{h,k}, a_{h,k}, s'_{h,k}, r_{h,k})$, which are independent from $(s_{h,k'}, a_{h,k'}, s'_{h,k'}, r_{h,k'})$ for any $k' \neq k$, so $\zeta_{h,k}$'s are independent from each other. As for the mean of $\zeta_{h,k}$,

$$\mathbb{E}\left[\zeta_{h,k} \mid \mathcal{D}_{h+1}, \cdots, \mathcal{D}_H\right]$$

$$= \mathbb{E}\left[r_{h,k} + \int_{\mathcal{A}} \widehat{Q}^\pi_{h+1}(s'_{h,k}, a) \pi(a \mid s'_{h,k}) \, \mathrm{d}a - r_h(s_{h,k}, a_{h,k}) - \mathcal{P}^\pi_h \widehat{Q}^\pi_{h+1}(s_{h,k}, a_{h,k}) \mid \mathcal{D}_{h+1}, \cdots, \mathcal{D}_H\right]$$

$$= \mathbb{E}\Bigg[r_{h,k} - r_h(s_{h,k}, a_{h,k}) + \int_{\mathcal{A}} \widehat{Q}^\pi_{h+1}(s'_{h,k}, a) \pi(a \mid s'_{h,k}) \, \mathrm{d}a$$

$$- \mathbb{E}_{s' \sim P_h(\cdot | s_{h,k}, a_{h,k})}\left[\int_{\mathcal{A}} \widehat{Q}^\pi_{h+1}(s', a) \pi(a \mid s') \, \mathrm{d}a \mid s_{h,k}, a_{h,k}, \mathcal{D}_{h+1}, \cdots, \mathcal{D}_H\right] \mid \mathcal{D}_{h+1}, \cdots, \mathcal{D}_H\Bigg]$$

$$= 0 + 0 = 0.$$

On the other hand, $\left\| \widehat{Q}_{h+1}^{\pi} \right\|_{\infty} \leq H$ almost surely, because it is a function in our CNN class $\mathcal{F}$. Thus, $\zeta_{h,k}$ is a bounded random variable with $\zeta_{h,k} \in [-2H, 2H]$ almost surely, so its variance is bounded by $4H^2$. Its boundedness also implies it is a subgaussian random variable. Thus, Condition 3 is also satisfied.

Hence, Lemma 10 proves, for step $h$ in our algorithm,

$$\mathbb{E}\left[ \int_{\mathcal{X}} \left( \mathcal{T}_h^{\pi} \widehat{Q}_{h+1}^{\pi} - \widehat{\mathcal{T}}_h^{\pi} \left( \widehat{Q}_{h+1}^{\pi} \right) \right)^2 (s, a) \, \mathrm{d}q_h^{\pi_0}(s, a) \mid \mathcal{D}_{h+1}, \cdots, \mathcal{D}_H \right]$$
$$\leq C'(H^2 + 4H^2) K^{-\frac{2\alpha}{2\alpha+d}} \log^5 K,$$

where $C'$ depends on $D^{\frac{6\alpha}{2\alpha+d}}$, $d$, $\alpha$, $\frac{2d}{\alpha p - d}$, $p$, $q$, $c_0$, $B$, $\omega$ and the surface area of $\mathcal{X}$.

Note that this upper bound holds for any $\widehat{Q}_{h+1}^{\pi}$ or $\mathcal{D}_{h+1}, \cdots, \mathcal{D}_H$. The sole purpose of our conditioning is that we could view $\widehat{Q}_{h+1}^{\pi}$ as a measurable or deterministic function under the conditioning and then apply Lemma 10. Therefore,

$$\mathbb{E}\left[ \mathbb{E}\left[ \int_{\mathcal{X}} \left( \mathcal{T}_h^{\pi} \widehat{Q}_{h+1}^{\pi} - \widehat{\mathcal{T}}_h^{\pi} \left( \widehat{Q}_{h+1}^{\pi} \right) \right)^2 (s, a) \, \mathrm{d}q_h^{\pi_0}(s, a) \mid \mathcal{D}_{h+1}, \cdots, \mathcal{D}_H \right] \right]$$
$$\leq C'(H^2 + 4H^2) K^{-\frac{2\alpha}{2\alpha+d}} \log^5 K.$$

Finally, we carry out the recursion from time step $1$ to time step $H$, and the final result is

$$\mathbb{E}\left| v^{\pi} - \widehat{v}^{\pi} \right| \leq C H^2 K^{-\frac{\alpha}{2\alpha+d}} \log^{5/2} K \left( \frac{1}{H} \sum_{h=1}^{H} \sqrt{\chi_{\mathcal{Q}}^2(q_h^{\pi}, q_h^{\pi_0}) + 1} \right).$$

$\square$

# B    PROOF OF THEOREM 1

For simplicity, let us denote $f_0 := \mathcal{T}_h^{\pi} f$ in the theorem statement. Note that $f_0 \in \mathcal{B}_{p,q}^{\alpha}(\mathcal{X})$. Moreover, let us define a class of single-block CNNs in the form of

$$f(x) = W \cdot \mathrm{Conv}_{\mathcal{W},\mathcal{B}}(x)$$

as

$$\mathcal{F}^{\mathrm{SCNN}}(L, J, I, \tau_1, \tau_2) = \big\{ f \mid f(x) \text{ in the form of } (3) \text{ with } L \text{ layers. The number of filters per block}$$
$$\text{is bounded by } L; \text{ filter size is bounded by } I; \text{ the number of channels}$$
$$\text{is bounded by } J; \max_{m,l} \|\mathcal{W}_m^{(l)}\|_{\infty} \vee \|B_m^{(l)}\|_{\infty} \leq \tau_1, \|W\|_{\infty} \leq \tau_2 \big\}.$$
$$(11)$$

We will refer to CNNs in this form as "single-block CNNs" and use them as building blocks of our final CNN approximation for the ground truth Besov function.

## B.1    PROOF OVERVIEW OF THEOREM 1

Theorem 1 serves as a building block for Theorem 2, which establishes the relation between network architecture and approximation error. For simplicity, denote $c_0 := \|f_0\|_{\mathcal{B}_{p,q}^{\alpha}(\mathcal{X})}$. Theorem 1 is proven in the following steps:

### STEP 1: DECOMPOSE $f$ AS SUM OF LOCALLY SUPPORTED FUNCTIONS OVER MANIFOLD

Since manifold $\mathcal{X}$ is assumed compact (Assumption 1), we can cover it with a finite set of $D$-dimensional open Euclidean balls $\{B_{\beta}(\mathbf{c}_i)\}_{i=1}^{C_{\mathcal{X}}}$, where $\mathbf{c}_i$ denotes the center of the $i$-th ball and $\beta$ is its radius. We choose $\beta < \omega/2$, and define $U_i = B_{\beta}(\mathbf{c}_i) \cap \mathcal{X}$. Note that each $U_i$ is diffeomorphic

to an open subset of $\mathbb{R}^d$ (Lemma 5.4 in Niyogi et al. [53]); moreover, $\{U_i\}_{i=1}^{C_{\mathcal{X}}}$ forms an open cover for $\mathcal{X}$. There exists a carefully designed open cover with cardinality $C_{\mathcal{X}} \leq \lceil \frac{A(\mathcal{X})}{\beta^d} T_d \rceil$, where $A(\mathcal{X})$ denotes the surface area of $\mathcal{X}$ and $T_d$ denotes the thickness of $U_i$'s, i.e., the average number of $U_i$'s that contain a given point on $\mathcal{X}$. $T_d$ is $O(d \log d)$ (Conway et al. [9]).

Moreover, for each $U_i$, we can define a linear transformation

$$\phi_i(x) = a_i V_i^\top (x - \mathbf{c}_i) + b_i,$$

where $a_i \in \mathbb{R}$ is the scaling factor and $b_i \in \mathbb{R}^d$ is the translation vector, both of which are chosen to ensure $\phi(U_i) \subset [0,1]^d$, and the columns of $V_i \in \mathbb{R}^{D \times d}$ form an orthonormal basis for the tangent space $T_{\mathbf{c}_i}(\mathcal{X})$. Overall, the atlas $\{(\phi_i, U_i)\}_{i=1}^{C_{\mathcal{X}}}$ transforms each local neighborhood on the manifold to a $d$-dimensional cube.

Thus, we can decompose $f_0$ using this atlas as

$$f_0 = \sum_{i=1}^{C_{\mathcal{X}}} f_i \quad \text{with} \quad f_i = f\rho_i, \tag{12}$$

because there exists such a $C^\infty$ partition of unity $\{\rho_i\}_{i=1}^{C_{\mathcal{X}}}$ with $\mathrm{supp}(\phi_i) \subset U_i$ (Proposition 1 in Liu et al. [42]). Since each $f_i$ is only supported on $U_i$, we can further write

$$f_0 = \sum_{i=1}^{C_{\mathcal{X}}} \left( f_i \circ \phi_i^{-1} \right) \circ \phi_i \times \mathbb{1}_{U_i} \quad \text{with} \quad f_i = f\rho_i, \tag{13}$$

where $\mathbb{1}_{U_i}$ is the indicator for membership in $U_i$.

Lastly, we extend $f_i \circ \phi_i^{-1}$ to entire $[0,1]^d$ with 0, which is a function in $\mathcal{B}_{p,q}^\alpha([0,1]^d)$ with $\mathcal{B}_{p,q}^\alpha([0,1]^d)$ Besov norm at most $Cc_0$ (Lemma 4 in Liu et al. [42]), where $C$ is a constant depending on $\alpha$, $p$, $q$ and $d$. This extended function is to be approximated with cardinal B-splines in the next step.

STEP 2: APPROXIMATE EACH LOCAL FUNCTION WITH CARDINAL B-SPLINES

With most things connected with the intrinsic dimension $d$ in the last step, we proceed an approximation of $f_0$ on the low-dimensional manifold. With $\alpha \geq d/p + 1$ assumed in Assumption 2, we can invoke a classic result of using cardinal B-splines to approximate Besov functions (Lemma 5), by setting $r = +\infty$ and $m = \lceil \alpha \rceil + 1$ in the lemma. It states that there exists a weighted sum of cardinal B-splines $\widetilde{f}_i$ in the form

$$\widetilde{f}_i \equiv \sum_{j=1}^{N} \widetilde{f}_{i,j} \approx f_i \circ \phi_i^{-1} \quad \text{with} \quad \widetilde{f}_{i,j} = c_{k,\mathbf{j}}^{(i)} \widetilde{g}_{k,\mathbf{j},m}^d \tag{14}$$

such that

$$\left\| \widetilde{f}_i - f_i \circ \phi_i^{-1} \right\|_{L^\infty} \leq Cc_0 N^{-\alpha/d}. \tag{15}$$

In (14), $c_{k,\mathbf{j}}^{(i)} \in \mathbb{R}$ is coefficient and $\widetilde{g}_{k,\mathbf{j},m}^d : [0,1]^d \to \mathbb{R}$ denotes a cardinal B-spline with index $k, m \in \mathbb{N}^+, \mathbf{j} \in \mathbb{R}^d$. $k$ is a scaling factor, $\mathbf{j}$ is a shifting vector, $m$ is the degree of the B-spline.

By (13) and (14), we now have a sum of cardinal B-splines

$$\widetilde{f} \equiv \sum_{i=1}^{C_{\mathcal{X}}} \widetilde{f}_i \circ \phi_i \times \mathbb{1}_{U_i} = \sum_{i=1}^{C_{\mathcal{X}}} \sum_{j=1}^{N} \widetilde{f}_{i,j} \circ \phi_i \times \mathbb{1}_{U_i}. \tag{16}$$

which can approximate our target Besov function $f_0$ with error

$$\left\| \widetilde{f} - f_0 \right\|_{L^\infty} \leq CC_{\mathcal{X}} c_0 N^{-\alpha/d}. \tag{17}$$

STEP 3: APPROXIMATE EACH CARDINAL B-SPLINE WITH A COMPOSITION OF CNNS

Each summand in (16) is a composition of functions, each of which we can implement with a CNN. Specifically, we do so with a special class of CNNs defined in (11), which we refer to as "single-block CNNs".

The multiplication operation $\times$ can be approximated by a single-block CNN $\widehat{\times}$ with at most $\eta$ error in the $L^\infty$ sense (Proposition 1). $\widehat{\times}$ needs $O(\log \frac{1}{\eta})$ layers and 6 channels. All weight parameters are bounded by $(c_0^2 \vee 1)$.

We consider each $\widetilde{f}_i \circ \phi_i$ together, which we can approximate with a sum of $N$ CNNs $\widehat{f}_{i,j}^{\mathrm{SCNN}} \circ \widehat{\phi}_i$ up to $\delta$ error, namely,

$$\left\| \sum_{j=1}^N \widehat{f}_{i,j}^{\mathrm{SCNN}} - \widetilde{f}_i \circ \phi_i^{-1} \right\|_{L^\infty} \le \delta.$$

In particular, we can use a single-block CNN $\widehat{f}_{i,j}^{\mathrm{SCNN}}$ to approximate the B-spline $\widetilde{f}_{i,j}$ up to $\delta/N$ error. Moreover, since $\phi_i$ is linear, it can be expressed with a single-layer perceptron $\widehat{\phi}_i$. The architecture and size of $\widehat{f}_{i,j}^{\mathrm{SCNN}}$ and $\widehat{\phi}_i$ are characterized in Proposition 2 as functions of $\delta$.

$\mathbb{1}_{U_i}$ is an indicator for membership in $U_i$, so we need $\mathbb{1}_{U_i}(x) = 1$ if $d_i^2(x) = \|x - \mathbf{c}_i\|_2^2 \le \beta^2$ and $\mathbb{1}_{U_i}(x) = 0$ otherwise. By this definition, we can write $\mathbb{1}_{U_i}$ as a composition of a univariate indicator $\mathbb{1}_{[0,\beta^2]}$ and the distance function $d_i^2$:

$$\mathbb{1}_{U_i}(x) = \mathbb{1}_{[0,\beta^2]} \circ d_i^2(x) \quad \text{for} \quad x \in \mathcal{X}. \tag{18}$$

Given $\theta \in (0,1)$ and $\Delta \ge 8DB^2\theta$, it turns out that $\mathbb{1}_{[0,\beta^2]}$ and $d_i^2$ can be approximated with two single-block CNNs $\widehat{\mathbb{1}}_\Delta$ and $\widehat{d}_i^2$ respectively (Proposition 3) such that

$$\left\| \widehat{d}_i^2 - d_i^2 \right\|_{L^\infty} \le 4B^2 D\theta \tag{19}$$

and

$$\widehat{\mathbb{1}}_\Delta \circ \widehat{d}_i^2(x) = \begin{cases} 1, & \text{if } x \in U_i, d_i^2(x) \le \beta^2 - \Delta, \\ 0, & \text{if } x \notin U_i, \\ \text{some value between 0 and 1,} & \text{otherwise.} \end{cases} \tag{20}$$

The architecture and size of $\widehat{\mathbb{1}}_\Delta$ and $\widehat{d}_i^2$ are characterized in Proposition 3 as functions of $\theta$ and $\Delta$.

The above three approximations rely on the classic result of using CNN to approximate cardinal B-splines (Lemma 10 in Liu et al. [42]; Lemma 1 in Suzuki [60]). Putting the above together, we can develop a composition of single-block CNNs

$$\bar{f}_{i,j} \equiv \widehat{\times} \left( \widehat{f}_{i,j}^{\mathrm{SCNN}} \circ \widehat{\phi}_i, \widehat{\mathbb{1}}_\Delta \circ \widehat{d}_i^2 \right) \tag{21}$$

as an approximation for $\widetilde{f}_{i,j} \circ \phi_i \times \mathbb{1}_{U_i}$. The overall approximation error of $\bar{f}_{i,j}$ can be written as a sum of the three types of approximation error above. Details are provided in Appendix B.2. Moreover, by Lemma 6, there exists a single-block CNN $\widehat{f}_{i,j}$ that can express $\bar{f}_{i,j}$.

STEP 4: EXPRESS THE SUM OF CNN COMPOSITIONS WITH A CNN

Finally, we can assemble everything into $\widehat{f}$

$$\widehat{f} \equiv \sum_{i=1}^{C_{\mathcal{X}}} \sum_{j=1}^N \widehat{f}_{i,j}, \tag{22}$$

which serves as an approximation for $f_0$. By choosing the appropriate network size in Lemma 2, which the tradeoff between the approximation error of $\widehat{f}_{i,j}$ and its size, we can ensure that

$$\left\| \widehat{f} - f_0 \right\|_{L^\infty} \le N^{-\alpha/d}. \tag{23}$$

By Lemma 7, for $\widetilde{M}, \widetilde{J} > 0$, we can write this sum of $N \cdot C_{\mathcal{X}}$ single-block CNNs as a sum of $\widetilde{M}$ single-block CNNs with the same architecture, whose channel number upper bound $J$ depends on $\widetilde{J}$. This allows Theorem 1 to be more flexible with network architecture. By Lemma 4, this sum of $\widetilde{M}$ CNNs can be further expressed as one CNN in the CNN class (5). Finally, $N$ will be chosen appropriately as a function of network architecture parameters, and the approximation theory of CNN is proven.

When Theorem 1 is applied in our problem setting, we will take the target function $f_0$ above to be $\mathcal{T}_h^\pi \widehat{Q}_{h+1}^\pi$ at each time step $h$, which is the ground truth of the regression at each step of Algorithm 1. For more details about this, please refer to the proof of Theorem 2 in Appendix A.

## B.2  PROOF OF THEOREM 1

In the following, we provide the proof details for Theorem 1, which quantifies the tradeoff between a CNN in the class of 11 and its approximation error for Besov functions on a low-dimensional manifold. We start from the decomposition of the approximation error of $\widehat{f}$, which is based on the decomposition of the approximation error of $\bar{f}_{i,j}$ in (21), and will proceed to the end of this proof.

**Lemma 2.** *Let $\eta$ be the approximation error of the multiplication operator $\widehat{\times}(\cdot, \cdot)$ as defined in Step 3 of Appendix B.1 and Proposition 1, $\delta$ be defined as in Step 3 of Appendix B.1 and Proposition 2, $\Delta$ and $\theta$ be defined as in Step 3 of Appendix B.1 and Proposition 3. Assume $N$ is chosen according to Proposition 2. For any $i = 1, ..., C_{\mathcal{X}}$, we have $\left\| \widehat{f} - f_0 \right\|_{L^\infty} \leq \sum_{i=1}^{C_{\mathcal{X}}} (A_{i,1} + A_{i,2} + A_{i,3})$ with*

$$A_{i,1} = \sum_{j=1}^{N} \left\| \widehat{\times}(\widehat{f}_{i,j}^{\text{SCNN}} \circ \widehat{\phi}_i, \widehat{\mathbb{1}}_\Delta \circ \widehat{d}_i^2) - \widehat{f}_{i,j}^{\text{SCNN}} \circ \widehat{\phi}_i \times (\widehat{\mathbb{1}}_\Delta \circ \widehat{d}_i^2) \right\|_{L^\infty} \leq C'' \delta^{-d/\alpha} \eta,$$

$$A_{i,2} = \left\| \left( \sum_{j=1}^{N} \left( \widehat{f}_{i,j}^{\text{SCNN}} \circ \widehat{\phi}_i \right) \right) \times (\widehat{\mathbb{1}}_\Delta \circ \widehat{d}_i^2) - f_i \times (\widehat{\mathbb{1}}_\Delta \circ \widehat{d}_i^2) \right\|_{L^\infty} \leq \delta,$$

$$A_{i,3} = \left\| f_i \times (\widehat{\mathbb{1}}_\Delta \circ \widehat{d}_i^2) - f_i \times \mathbb{1}_{U_i} \right\|_{L^\infty} \leq \frac{c(\pi+1)}{\beta(1-\beta/\omega)} \Delta$$

*for some constant $C''$ depending on $d, \alpha, p, q$ and some constant $c$. Furthermore, for any $\varepsilon \in (0, 1)$, setting*

$$\delta = \frac{N^{-\alpha/d}}{3C_{\mathcal{X}}}, \ \eta = \frac{1}{C''} \frac{N^{-1-\alpha/d}}{(3C_{\mathcal{X}})^{d/\alpha}}, \Delta = \frac{\beta(1-\beta/\omega)N^{-\alpha/d}}{3c(\pi+1)C_{\mathcal{X}}}, \ \theta = \frac{\Delta}{16B^2D} \tag{24}$$

*gives rise to*

$$\left\| \widehat{f} - f_0 \right\|_{L^\infty} \leq N^{-\frac{\alpha}{d}}.$$

*The choice in (24) satisfies the condition $\Delta > 8B^2D\theta$ in Proposition 3.*

*Proof of Lemma 2.* As in Proposition 1, $A_{i,1}$ measures the error from $\widehat{\times}$:

$$A_{i,1} = \sum_{j=1}^{N} \left\| \widehat{\times}(\widehat{f}_{i,j}^{\text{SCNN}} \circ \widehat{\phi}_i, \widehat{\mathbb{1}}_\Delta \circ \widehat{d}_i^2) - \widehat{f}_{i,j}^{\text{SCNN}} \circ \widehat{\phi}_i \times (\widehat{\mathbb{1}}_\Delta \circ \widehat{d}_i^2) \right\|_{L^\infty} \leq N\eta \leq C'' \delta^{-d/\alpha} \eta,$$

for some constant $C''$ depending on $d, \alpha, p, q$. The last inequality is due to the choice of $N$ in Proposition 2.

$A_{i,2}$ measures the error from CNN approximation of Besov functions. As in Proposition 2, $A_{i,2} \leq \delta$.

$A_{i,3}$ measures the error from CNN approximation of the chart determination function. The bound of $A_{i,3}$ can be derived using the proof of Lemma 4.5 in Chen et al. [7], since $f_i \circ \phi_i^{-1}$ is a Lipschitz function and its domain is in $[0, 1]^d$. $\qquad\square$

In order to attain the error desired in Lemma 2, we need each network in $\bar{f}_{i,j}$ with appropriate size. The network size of the components in $\bar{f}_{i,j}$ can be analyzed as follows:

- $\widehat{\mathbb{1}}_i$: The chart determination network $\widehat{\mathbb{1}}_i = \widehat{d_i^2} \circ \widehat{\mathbb{1}}_\Delta$ is the composition of $\widehat{d_i^2}$ and $\widehat{\mathbb{1}}_\Delta$. By Proposition 3, $\widehat{d_i^2}$ is a single-block CNN with $O(\log \frac{1}{\theta}) = O(\frac{\alpha}{d}\log N + D + \log D)$ layers and width $6D$; $\widehat{\mathbb{1}}_\Delta$ is a single-block CNN with $O(\log(\beta^2/\Delta)) = O(\frac{\alpha}{d}\log N)$ layers and width 2. In both subnetworks, all parameters are of $O(1)$. By Lemma 6, the chart determination network $\widehat{\mathbb{1}}_i$ is a single-block CNN with $O(\frac{\alpha}{d}\log N + D + \log D)$ layers, width $6D + 2$ and all weight parameters are of $O(1)$.

- $\widehat{\times}$: By Proposition 1, the multiplication network is a single-block CNN with $O(\log \frac{1}{\eta}) = O((1 + \frac{\alpha}{d})\log N)$ layers and $O(1)$ width. All weight parameters are bounded by $(c_0^2 \vee 1)$.

- $\widehat{\phi}_i$: The projection $\phi_i$ is a linear one, so it can be expressed with a single-layer perceptron. By Lemma 8 in Liu et al. [42], this single-layer perceptron can be expressed with a single-block CNN with $2 + D$ layers and width $d$. All parameters are of $O(1)$.

- $\widehat{f}_{i,j}^{\mathrm{SCNN}}$: by Proposition 2, each $\widehat{f}_{i,j}^{\mathrm{SCNN}}$ is a single-block CNN with $O(\log \frac{1}{\delta}) = O(\frac{\alpha}{d}\log N)$ layers and $\lceil 24d(\alpha+1)(\alpha+3) + 8d\rceil$ channels. All weight parameters are in the order of $O\left(\delta^{-(\log 2)(\frac{2d}{\alpha p - d} + c_1 d^{-1})}\right) = O\left(N^{(\log 2)\frac{\alpha}{d}(\frac{2d}{\alpha p - d} + c_1 d^{-1})}\right)$.

Next, we want to show $\bar{f}_{i,j}$, a composition of the aforementioned single-block CNNs, can be simply expressed as a single-block CNN.

By Lemma 6, there exists a single-block CNN $g_{i,j}$ with $O(\log N + D)$ layers and $\lceil 24d(\alpha + 1)(\alpha + 3) + 9d\rceil$ width realizing $\widehat{f}_{i,j}^{\mathrm{SCNN}} \circ \widehat{\phi}_i$. All weight parameters in $g_{i,j}$ are in the order of $O\left(N^{(\log 2)\frac{\alpha}{d}(\frac{2d}{\alpha p - d} + c_1 d^{-1})}\right)$. Moreover, recall that the chart determination network $\widehat{\mathbb{1}}_i$ is a single-block CNN with $O(\log N + D + \log D)$ layers and width $6D + 2$, whose weight parameters are of $O(1)$. By Lemma 14 in Liu et al. [42], one can construct a convolutional block, denoted by $\bar{g}_{i,j}$, such that

$$\bar{g}_{i,j}(x) = \begin{bmatrix} (g_{i,j}(x))_+ & (g_{i,j}(x))_- & (\widehat{\mathbb{1}}_i(x))_+ & (\widehat{\mathbb{1}}_i(x))_- \\ \star & \star & \star & \star \end{bmatrix}. \tag{25}$$

Here $\bar{g}_{i,j}$ has $\lceil 24d(\alpha+1)(\alpha+3) + 9d\rceil + 6D + 2$ channels.

Since the input of $\widehat{\times}$ is $\begin{bmatrix} g_{i,j} \\ \widehat{\mathbb{1}}_i \end{bmatrix}$, by Lemma 15 in Liu et al. [42], there exists a CNN $\mathring{g}_{i,j}$ which takes (25) as the input and outputs $\widehat{\times}(g_{i,j}, \widehat{\mathbb{1}}_i)$.

Note that $\bar{g}_{i,j}$ only contains convolutional layers. The composition $\mathring{g}_{i,j} \circ \bar{g}_{i,j}$, denoted by $\widehat{g}_{i,j}^{\mathrm{SCNN}}$, is a CNN and for any $x \in \mathcal{X}$, $\widehat{g}_{i,j}^{\mathrm{SCNN}}(x) = \bar{f}_{i,j}(x)$. We have $\widehat{g}_{i,j}^{\mathrm{SCNN}} \in \mathcal{F}^{\mathrm{SCNN}}(L, J, I, \tau, \tau)$ with

$$L = O\left(\log N + D + \log D\right), \quad J = \lceil 48d(\alpha+1)(\alpha+3) + 18d\rceil + 12D + O(1),$$
$$\tau = O\left(N^{(\log 2)\frac{d}{\alpha}(\frac{2d}{\alpha p - d} + c_1 d^{-1})}\right), \tag{26}$$

and $I$ can be any integer in $[2, D]$.

Therefore, we have shown that $\widehat{g}_{i,j}^{\mathrm{SCNN}}$ is a single-block CNN that expresses $\bar{f}_{i,j}$, as we desired.

Furthermore, recall that $\widetilde{f}$ can be written as a sum of $C_{\mathcal{X}} N$ such SCNNs. By Lemma 7, for any $\widetilde{M}, \widetilde{J}$ satisfying $\widetilde{M}\widetilde{J} = O(N)$, there exists a CNN architecture $\mathcal{F}^{\mathrm{SCNN}}(L, J, I, \tau, \tau)$ that gives rise to a set of single-block CNNs $\{\widehat{g}_i\}_{i=1}^{\widetilde{M}} \in \mathcal{F}^{\mathrm{SCNN}}(L, J, I, \tau, \tau)$ with

$$\widehat{f} = \sum_{i=1}^{\widetilde{M}} \widehat{g}_i \tag{27}$$

and

$$L = O\left(\log N + D + \log D\right), \quad J = O(D\widetilde{J}), \quad \tau = O\left(N^{(\log 2)\frac{d}{\alpha}(\frac{2d}{\alpha p - d} + c_1 d^{-1})}\right). \tag{28}$$

By Lemma 3 below, we slightly adjust the CNN architecture by re-balancing the weight parameter boundary of the convolutional blocks and that of the final fully connected layer. In particular, we

rescale all parameters in convolutional layers of $\widehat{g}_i$ to be no larger than 1. While this procedure does not change the approximation power of the CNN, it can make the CNN have a smaller covering number, which is conducive to a smaller variance.

**Lemma 3** (Lemma 16 in Liu et al. [42]). *Let $\gamma \geq 1$. For any $g \in \mathcal{F}^{\mathrm{SCNN}}(L, J, I, \tau_1, \tau_2)$, there exists $f \in \mathcal{F}^{\mathrm{SCNN}}(L, J, I, \gamma^{-1}\tau, \gamma^L\tau)$ such that $g(x) = f(x)$.*

In this case, we set $\gamma = c' N^{(\log 2)\frac{d}{\alpha}(\frac{2d}{\alpha p - d} + c_1 d^{-1})}(8ID)\widetilde{M}^{\frac{1}{L}}$, where $c'$ is a constant such that $\tau \leq c' N^{(\log 2)\frac{d}{\alpha}(\frac{2d}{\alpha p - d} + c_1 d^{-1})}$. With this $\gamma$, we have $\widehat{f}_i \in \mathcal{F}^{\mathrm{SCNN}}(L, J, I, \tau_1, \tau_2)$ with

$$L = O(\log N + D + \log D), \ J = O(D), \ \tau_1 = (8ID)^{-1}\widetilde{M}^{-\frac{1}{L}} = O(1),$$

$$\log \tau_2 = O\left(\log \widetilde{M} + \log^2 N + D \log N\right).$$

Finally, we prove that it suffices to use one CNN to realize the sum of single-block CNNs in (27).

**Lemma 4.** *Let $\mathcal{F}^{\mathrm{SCNN}}(L, J, I, \tau_1, \tau_2)$ be any CNN architecture from $\mathbb{R}^D$ to $\mathbb{R}$. Assume the weight matrix in the fully connected layer of $\mathcal{F}^{\mathrm{SCNN}}(L, J, I, \tau_1, \tau_2)$ has nonzero entries only in the first row. For any positive integer $M$, there exists a CNN architecture $\mathcal{F}(M, L, J, I, \tau_1, \tau_2(1 \vee \tau_1^{-1}))$ such that for any $\{\widehat{f}_i(x)\}_{i=1}^M \subset \mathcal{F}^{\mathrm{SCNN}}(L, J, I, \tau_1, \tau_2)$, there exists $\widehat{f} \in \mathcal{F}(M, L, 4 + J, I, \tau_1, \tau_2(1 \vee \tau_1^{-1}))$ with*

$$\widehat{f}(x) = \sum_{m=1}^M \widehat{f}_m(x).$$

Consequently, by Lemma 4, there exists a CNN that can express our sum of $\widetilde{M}$ single-block CNNs with architecture $\mathcal{F}(M, L, J, I, \tau_1, \tau_2)$ with

$$L = O(\log N + D + \log D), \ J = O(D\widetilde{J}), \ \tau_1 = (8ID)^{-1}\widetilde{M}^{-\frac{1}{L}} = O(1),$$

$$\log \tau_2 = O\left(\log \widetilde{M} + \log^2 N + D \log N\right), \ M = O(\widetilde{M}). \tag{29}$$

and $\widetilde{J}, \widetilde{M}$ satisfying

$$\widetilde{M}\widetilde{J} = O(N), \tag{30}$$

which is a requirement inherited from Lemma 7. This CNN is our final approximation for $f_0$.

Applying this relation $N = O(\widetilde{M}\widetilde{J})$ to (29) gives

$$\left\|\widehat{f} - f_0\right\|_{L^\infty} \leq (\widetilde{M}\widetilde{J})^{-\frac{\alpha}{d}} \tag{31}$$

and the network size

$$L = O\left(\log(\widetilde{M}\widetilde{J}) + D + \log D\right), \ J = O(D\widetilde{J}), \ \tau_1 = (8ID)^{-1}\widetilde{M}^{-\frac{1}{L}} = O(1),$$

$$\log \tau_2 = O\left(\log^2 \widetilde{M}\widetilde{J} + D \log \widetilde{M}\widetilde{J}\right), \ M = O(\widetilde{M}).$$

### B.3 Proof of Lemma 4

Denote the architecture of $\widehat{f}_m$ with

$$\widehat{f}_m(x) = W_m \cdot \mathrm{Conv}_{\mathcal{W}_m, \mathcal{B}_m}(x),$$

where $\mathcal{W}_m = \left\{\mathcal{W}_m^{(l)}\right\}_{l=1}^L, \mathcal{B}_m = \left\{B_m^{(l)}\right\}_{l=1}^L$. Furthermore, denote the weight matrix and bias in the fully connected layer of $\widehat{f}$ with $\widehat{W}, \widehat{b}$ and the set of filters and biases in the $m$-th block of $\widehat{f}$ with $\widehat{\mathcal{W}}_m$ and $\widehat{\mathcal{B}}_m$, respectively. The padding layer $\widehat{P}$ in $\widehat{f}$ pads the input $x$ from $\mathbb{R}^D$ to $\mathbb{R}^{D\times 4}$ with zeros. Each column denotes a channel.

Let us first show that for each $m$, there exists some $\mathrm{Conv}_{\widehat{\mathcal{W}}_m, \widehat{\mathcal{B}}_m} : \mathbb{R}^{D\times 4} \to \mathbb{R}^{D\times 4}$ such that for any $Z \in \mathbb{R}^{D\times 4}$ with the form

$$Z = \begin{bmatrix} (x)_+ & (x)_- & \star & \star \end{bmatrix}, \tag{32}$$

where $(x)_+$ means applying $(\cdot \vee 0)$ to every entry of $x$ and $(x)_-$ means applying $-(\cdot \wedge 0)$ to every entry of $x$, so all entries in $Z$ are non-negative. We have

$$\mathrm{Conv}_{\widehat{\mathcal{W}}_m, \widehat{\mathcal{B}}_m}(Z) = \begin{bmatrix} \mathbf{0} & \mathbf{0} & \frac{\tau_1}{\tau_2}(f_m(\boldsymbol{x}) \vee 0) & -\frac{\tau_1}{\tau_2}(f_m(\boldsymbol{x}) \wedge 0) \\ & & \star & \star \\ & & \vdots & \vdots \\ & & \star & \star \end{bmatrix} + Z \qquad (33)$$

where $\star$'s denotes entries that do not affect this result and may take any different value.

For any $m$, the first layer of $f_m$ takes input in $\mathbb{R}^D$. Thus, the filters in $\mathcal{W}_m^{(1)}$ are in $\mathbb{R}^D$. Again, we pad these filters with zeros to get filters in $\mathbb{R}^{D \times 4}$ and construct $\widehat{\mathcal{W}}_m^{(1)}$ such that

$$(\widehat{\mathcal{W}}_m^{(1)})_{1,:,:} = [\mathbf{e}_1 \quad \mathbf{0} \quad \mathbf{0} \quad \mathbf{0}],$$
$$(\widehat{\mathcal{W}}_m^{(1)})_{2,:,:} = [\mathbf{0} \quad \mathbf{e}_1 \quad \mathbf{0} \quad \mathbf{0}],$$
$$(\widehat{\mathcal{W}}_m^{(1)})_{3,:,:} = [\mathbf{0} \quad \mathbf{0} \quad \mathbf{e}_1 \quad \mathbf{0}],$$
$$(\widehat{\mathcal{W}}_m^{(1)})_{4,:,:} = [\mathbf{0} \quad \mathbf{0} \quad \mathbf{0} \quad \mathbf{e}_1],$$
$$(\widehat{\mathcal{W}}_m^{(1)})_{4+j,:,:} = \left[(\mathcal{W}_m^{(1)})_{j,:,:} \quad (-\mathcal{W}_m^{(1)})_{j,:,:} \quad \mathbf{0} \quad \mathbf{0}\right],$$

where we use the fact that $\mathcal{W}_m^{(1)} * (x)_+ - \mathcal{W}_m^{(1)} * (x)_- = \mathcal{W}_m^{(1)} * x$. The first four output channels at the end of this first layer is a copy of $Z$. For the filters in later layers of $\widehat{f}_m$ and all biases, we simply set

$$(\widehat{\mathcal{W}}_m^{(l)})_{1,:,:} = [\mathbf{e}_1 \quad \mathbf{0} \quad \mathbf{0} \quad \mathbf{0} \quad \cdots \quad \mathbf{0}] \qquad \text{for } l = 2, \dots, L,$$
$$(\widehat{\mathcal{W}}_m^{(l)})_{2,:,:} = [\mathbf{0} \quad \mathbf{e}_1 \quad \mathbf{0} \quad \mathbf{0} \quad \cdots \quad \mathbf{0}] \qquad \text{for } l = 2, \dots, L,$$
$$(\widehat{\mathcal{W}}_m^{(l)})_{3,:,:} = [\mathbf{0} \quad \mathbf{0} \quad \mathbf{e}_1 \quad \mathbf{0} \quad \cdots \quad \mathbf{0}] \qquad \text{for } l = 2, \dots, L-1,$$
$$(\widehat{\mathcal{W}}_m^{(l)})_{4,:,:} = [\mathbf{0} \quad \mathbf{0} \quad \mathbf{0} \quad \mathbf{e}_1 \quad \cdots \quad \mathbf{0}] \qquad \text{for } l = 2, \dots, L-1,$$
$$(\widehat{\mathcal{W}}_m^{(l)})_{4+j,:,:} = \left[\mathbf{0} \quad \mathbf{0} \quad \mathbf{0} \quad \mathbf{0} \quad (\mathcal{W}_m^{(l)})_{j,:,:}\right] \qquad \text{for } l = 2, \dots, L-1,$$
$$(\widehat{\mathcal{B}}_m^{(l)})_{j,:,:} = \left[\mathbf{0} \quad \mathbf{0} \quad \mathbf{0} \quad \mathbf{0} \quad (\mathcal{B}_m^{(l)})_{j,:,:}\right] \qquad \text{for } l = 1, \dots, L-1.$$

In $\mathrm{Conv}_{\widehat{\mathcal{W}}_m, \widehat{\mathcal{B}}_m}$, an additional convolutional layer is constructed to realize the fully connected layer in $\widehat{f}_m$. By our assumption, only the first row of $W_m$ is nonzero. Furthermore, we set $\widehat{\mathcal{B}}_m^{(L)} = \mathbf{0}$ and $\widehat{\mathcal{W}}_m^L$ as size-one filters with three output channels in the form of

$$(\widehat{\mathcal{W}}_m^{(L)})_{3,:,:} = \left[\mathbf{0} \quad \mathbf{0} \quad \mathbf{e}_1 \quad \mathbf{0} \quad \frac{\tau_1}{\tau_2}(W_m)_{1,:}\right],$$
$$(\widehat{\mathcal{W}}_m^{(L)})_{4,:,:} = \left[\mathbf{0} \quad \mathbf{0} \quad \mathbf{0} \quad \mathbf{e}_1 \quad -\frac{\tau_1}{\tau_2}(W_m)_{1,:}\right].$$

Under such choices, (33) is proved and all parameters in $\widehat{\mathcal{W}}_m, \widehat{\mathcal{B}}_m$ are bounded by $\tau_1$.

By composing all convolutional blocks, we have

$$(\mathrm{Conv}_{\widehat{\mathcal{W}}_M, \widehat{\mathcal{B}}_M}) \circ \cdots \circ (\mathrm{Conv}_{\widehat{\mathcal{W}}_1, \widehat{\mathcal{B}}_1}) \circ P(x) = \begin{bmatrix} (x)_+ & (x)_- & \frac{\tau_1}{\tau_2} \sum_{m=1}^M (\widehat{f}_m \vee 0) & -\frac{\tau_1}{\tau_2} \sum_{m=1}^M (\widehat{f}_m \wedge 0) \\ & & \star & \star \\ & & \vdots & \vdots \\ & & \star & \star \end{bmatrix}.$$

Lastly, the fully connect layer can be set as

$$\widetilde{W} = \begin{bmatrix} 0 & 0 & \frac{\tau_2}{\tau_1} & -\frac{\tau_2}{\tau_1} \\ 0 & 0 & \mathbf{0} & \mathbf{0} \end{bmatrix}, \quad \widetilde{b} = 0.$$

Note that the weights in the fully connected layer are bounded by $\tau_2(1 \vee \tau_1^{-1})$.

The above construction gives

$$\widehat{f}(x) = \sum_{m=1}^M (\widehat{f}_m(x) \vee 0) + \sum_{m=1}^M (\widehat{f}_m(x) \wedge 0) = \sum_{m=1}^M \widehat{f}_m(x).$$

### B.4 Supporting Lemmae for Theorem 1

Before stating Lemma 5, we provide a brief definition of cardinal B-splines.

**Definition 5** (Cardinal B-spline). *Let $\psi(x) = \mathbb{1}_{[0,1]}(x)$ be the indicator function for membership in $[0,1]$. The cardinal B-spline of order $m$ is defined by taking $m+1$-times convolution of $\psi$:*

$$\psi_m(x) = (\underbrace{\psi * \psi * \cdots * \psi}_{m+1 \text{ times}})(x)$$

*where $f * g(x) \equiv \int f(x-t)g(t)dt$.*

Note that $\psi_m$ is a piecewise polynomial with degree $m$ and support $[0, m+1]$. It can be expressed as [45]

$$\psi_m(x) = \frac{1}{m!} \sum_{j=0}^{m+1} (-1)^j \binom{m+1}{j} (x-j)_+^m.$$

For any $k, j \in \mathbb{N}$, let $\widetilde{g}_{k,j,m}(x) = \psi_m(2^k x - j)$, which is the rescaled and shifted cardinal B-spline with resolution $2^{-k}$ and support $2^{-k}[j, j + (m+1)]$. For $\mathbf{k} = (k_1, \ldots, k_d) \in \mathbb{N}^d$ and $\mathbf{j} = (j_1, \ldots, j_d) \in \mathbb{N}^d$, we define the $d$ dimensional cardinal B-spline as $\widetilde{g}_{\mathbf{k},\mathbf{j},m}^d(x) = \prod_{i=1}^d \psi_m(2^{k_i} x_i - j_i)$. When $k_1 = \ldots = k_d = k \in \mathbb{N}$, we denote $\widetilde{g}_{k,\mathbf{j},m}^d(x) = \prod_{i=1}^d \psi_m(2^k x_i - j_i)$.

#### B.4.1 Approximating Besov functions with Cardinal B-Splines

For any $m \in \mathbb{N}$, let $J(k) = \{-m, -m+1, \ldots, 2^k - 1, 2^k\}^d$ and the quasi-norm of the coefficient $\{c_{k,j}\}$ for $k \in \mathbb{N}, \mathbf{j} \in J(k)$ be

$$\|\{c_{k,\mathbf{j}}\}\|_{b_{p,q}^\alpha} = \left( \sum_{k \in \mathbb{N}} \left[ 2^{k(\alpha - d/p)} \left( \sum_{\mathbf{j} \in J(k)} |c_{k,\mathbf{j}}|^p \right)^{1/p} \right]^q \right)^{1/q}. \tag{34}$$

We can state the following lemma, from DeVore & Popov [13], Dung [17], which provides an upper bound on the error of using cardinal B-splines to approximate functions in $\mathcal{B}_{p,q}^\alpha([0,1]^d)$.

**Lemma 5** (Lemma 2 in Suzuki [60]; DeVore & Popov [13], Dung [17]). *Assume that $0 < p, q, r \leq \infty$ and $0 < \alpha < \infty$ satisfying $\alpha > d(1/p - 1/r)_+$. Let $m \in \mathbb{N}$ be the order of the cardinal B-spline basis such that $0 < \alpha < \min(m, m-1 + 1/p)$. For any $f \in \mathcal{B}_{p,q}^\alpha([0,1]^d)$, there exists $f_N$ satisfying*

$$\|f - f_N\|_{L^r([0,1]^d)} \leq CN^{-\alpha/d} \|f\|_{\mathcal{B}_{p,q}^\alpha([0,1]^d)}$$

*for some constant $C$ with $N \gg 1$. $f$ is in the form of*

$$f_N(x) = \sum_{k=0}^{H} \sum_{\mathbf{j} \in J(k)} c_{k,\mathbf{j}} \widetilde{g}_{k,\mathbf{j},m}^d(x) + \sum_{k=K+1}^{H^*} \sum_{i=1}^{n_k} c_{k,\mathbf{j}_i} \widetilde{g}_{k,\mathbf{j}_i,m}^d(x), \tag{35}$$

*where $\{\mathbf{j}_i\}_{i=1}^{n_k} \subset J(k), H = \lceil c_1 \log(N)/d \rceil, H^* = \lceil \nu^{-1} \log(\lambda N) \rceil + H + 1, n_k = \lceil \lambda N 2^{-\nu(k-H)} \rceil$ for $k = H+1, \ldots, H^*, u = d(1/p - 1/r)_+$ and $\nu = (\alpha - u)/(2u)$. The real numbers $c_1 > 0$ and $\lambda > 0$ are two absolute constants chosen to satisfy $\sum_{k=1}^H (2^k + m)^d + \sum_{k=H+1}^{H^*} n_k \leq N$, which are to $N$. Moreover, we can choose the coefficients $\{c_{k,\mathbf{j}}\}$ such that*

$$\|\{c_{k,\mathbf{j}}\}\|_{b_{p,q}^\alpha} \leq C_1 \|f\|_{\mathcal{B}_{p,q}^\alpha([0,1]^d)}$$

*for some constant $C_1$.*

#### B.4.2 Approximating Cardinal B-Splines and Others with Single-Block CNNs

The following Proposition 1 quantifies the tradeoff between the size of a single-block CNN and its approximation error for the multiplication operator.

**Proposition 1.** *Let $\times$ be defined as in (13). For any $\eta \in (0,1)$, there exists a single-block CNN $\widehat{\times}(\cdot, \cdot)$ such that*

$$\left\| a \times b - \widehat{\times}(a,b) \right\|_{L^\infty} \leq \eta,$$

*where $a, b$ are functions uniformly bounded by $c_0$.*

$\widehat{\times}$ *is a single-block CNN approximation of $\times$ and is in $\mathcal{F}^{\text{SCNN}}(L, J, I, \tau, \tau)$ with $L = O(\log 1/\eta) + D$ layers, $J = 24$ channels and any $2 \leq I \leq D$. All parameters are bounded by $\tau = (c_0^2 \vee 1)$. Furthermore, the weight matrix in the fully connected layer of $\widehat{\times}$ has nonzero entries only in the first row.*

*Proof of Proposition 1.* First, let us define a particular class of feed-forward ReLU networks of the form

$$f(x) = W_L \cdot \text{ReLU}(W_{L-1} \cdots \text{ReLU}(W_1 x + b_1) \cdots + b_{L-1}) + b_L, \tag{36}$$

as

$$\mathcal{F}(L, J, \tau) = \{f \mid f(x) \text{ in the form (36) with } L \text{ layers and width at most } J,$$
$$\|W_i\|_{\infty,\infty} \leq \tau, \ \|b_i\|_\infty \leq \tau \text{ for } i = 1, \cdots, L\}. \tag{37}$$

By Proposition 3 in Yarotsky, there exists a feed-forward ReLU network that can approximate the multiplication operation between values with magnitude bounded by $c_0$, with $\eta$ error. Such feed-forward network has $O(\log 1/\eta)$ layers, whose width is all bounded by 6, and all its parameters are bounded by $c_0^2$. Therefore, such a feed-forward network is sufficient to approximate $\times$ with $\eta$ error in $L^\infty$-norm, because the arguments of $\times$ are uniformly bounded $c_0$ by Assumption 2.

Furthermore, by Lemma 8 in Liu et al. [42], we can express the aforementioned feed-forward network with a single-block CNN in $\mathcal{F}^{\text{SCNN}}(L, J, I, \tau, \tau)$, where $L, J, I, \tau$ are as specified in the statement of the proposition. $\square$

Proposition 2 quantifies the tradeoff between the size of a single-block CNN and its approximation error for the cardinal B-spline $f_i \circ \phi_i^{-1}$.

**Proposition 2** (Proposition 3 in Liu et al. [42]). *Let $f_i \circ \phi_i^{-1}$ be defined as in (13). For any $\delta \in (0, 1)$, set $N = C_1 \delta^{-d/\alpha}$. For any $2 \leq I \leq D$, there exists a set of single-block CNNs $\left\{ \widehat{f}^{\text{SCNN}} \right\}_{j=1}^N$ such that*

$$\left\| \sum_{j=1}^N \widehat{f}_{i,j}^{\text{SCNN}} - f_i \circ \phi_i^{-1} \right\|_{L^\infty} \leq \delta,$$

*where $C_1$ is a constant depending on $\alpha, p, q$ and $d$.*

$\widehat{f}_{i,j}^{\text{SCNN}}$ *is a single-block CNN approximation of $\widetilde{f}_{i,j}$ (defined in (14)) in $\mathcal{F}^{\text{SCNN}}(L, J, I, \tau, \tau)$ with*

$$L = O\left(\log(1/\delta)\right), J = \lceil 24d(\alpha+1)(\alpha+3) + 8d \rceil, \tau = O\left(\delta^{-(\log 2)(\frac{2d}{\alpha p - d} + c_1 d^{-1})}\right).$$

*The constant hidden in $O(\cdot)$ depends on $d, \alpha, \frac{2d}{\alpha p - d}, p, q, c_0$.*

Proposition 3 quantifies the tradeoff between the size of the sub-networks for the chart determination network and its approximation error for the chart determination indicators and the distance function $d_i^2$.

**Proposition 3** (Lemma 9 in Liu et al. [42]). *Let $d_i^2$ and $\mathbb{1}_{[0,\beta^2]}$ be defined as in (18). For any $\theta \in (0, 1)$ and $\Delta \geq 8B^2 D\theta$, there exists a single-block CNN $\widehat{d_i^2}$ approximating $d_i^2$ such that*

$$\|\widehat{d_i^2} - d_i^2\|_{L^\infty} \leq 4B^2 D\theta,$$

*and a CNN $\widehat{\mathbb{1}}_\Delta$ approximating $\mathbb{1}_{[0,\beta^2]}$ with*

$$\widehat{\mathbb{1}}_\Delta(x) = \begin{cases} 1, & \text{if } a \leq (1 - 2^{-k})(\beta^2 - 4B^2 D\theta), \\ 0, & \text{if } a \geq \beta^2 - 4B^2 D\theta, \\ 2^k((\beta^2 - 4B^2 D\theta)^{-1}a - 1), & \text{otherwise}. \end{cases}$$

*for $x \in \mathcal{X}$. The single-block CNN for $\widehat{d_i^2}$ has $O(\log(1/\theta))$ layers, $6D$ channels and all weights parameters are bounded by $4B^2$. The single-block CNN for $\widehat{\mathbb{1}}_\Delta$ has $\lceil \log(\beta^2/\Delta) \rceil$ layers, 2 channels. All weight parameters are bounded by $\max(2, |\beta^2 - 4B^2 D\theta|)$.*

*As a result, for any $x \in \mathcal{X}$, $\widehat{\mathbb{1}}_{\Delta} \circ \widehat{d}_i^2(x)$ gives an approximation of $\mathbb{1}_{U_i}$ satisfying*

$$\widehat{\mathbb{1}}_{\Delta} \circ \widehat{d}_i^2(x) = \begin{cases} 1, & \text{if } x \in U_i \text{ and } d_i^2(x) \leq \beta^2 - \Delta; \\ 0, & \text{if } x \notin U_i; \\ \text{between 0 and 1}, & \text{otherwise.} \end{cases}$$

### B.4.3 LEMMAE ABOUT SUMMATION AND COMPOSITION OF CNN

Lemma 6 states that the composition of two single-block CNNs can be expressed as one single-block CNN with augmented architecture.

**Lemma 6.** *Let $\mathcal{F}_1^{\text{SCNN}}(L_1, J_1, I_1, \tau_1, \tau_1)$ be a CNN architecture from $\mathbb{R}^D \rightarrow \mathbb{R}$ and $\mathcal{F}_2^{\text{SCNN}}(L_2, J_2, I_2, \tau_2, \tau_2)$ be a CNN architecture from $\mathbb{R} \rightarrow \mathbb{R}$. Assume the weight matrix in the fully connected layer of $\mathcal{F}_1^{\text{SCNN}}(L_1, J_1, I_1, \tau_1, \tau_1)$ and $\mathcal{F}_2^{\text{SCNN}}(L_2, J_2, I_2, \tau_2, \tau_2)$ has nonzero entries only in the first row. Then there exists a CNN architecture $\mathcal{F}^{\text{SCNN}}(L, J, I, \tau, \tau)$ from $\mathbb{R}^D \rightarrow \mathbb{R}$ with*

$$L = L_1 + L_2, \ J = \max(J_1, J_2), \ I = \max(I_1, I_2), \tau = \max(\tau_1, \tau_2)$$

*such that for any $f_1 \in \mathcal{F}^{\text{SCNN}}(L_1, J_1, I_1, \tau_1, \tau_1)$ and $f_2 \in \mathcal{F}^{\text{SCNN}}(L_2, J_2, I_2, \tau_2, \tau_2)$, there exists $f \in \mathcal{F}^{\text{SCNN}}(L, J, I, \tau, \tau)$ such that $f(x) = f_2 \circ f_1(x)$. Furthermore, the weight matrix in the fully connected layer of $\mathcal{F}^{\text{SCNN}}(L, J, I, \tau, \tau)$ has nonzero entries only in the first row.*

Lemma 7 states that the sum of $n_0$ single-block CNNs with the same architecture can be expressed as the sum of $n_1$ single-block CNNs with modified width.

**Lemma 7** (Lemma 7 in Liu et al. [43]). *Let $\{f_i\}_{i=1}^{n_0}$ be a set of single-block CNNs with architecture $\mathcal{F}^{\text{SCNN}}(L_0, J_0, I_0, \tau_0, \tau_0)$. For any integers $1 \leq n \leq n_0$ and $\widetilde{J}$ satisfying $n\widetilde{J} = O(n_0 J_0)$ and $\widetilde{J} \geq J_0$, there exists an architecture $\mathcal{F}^{\text{SCNN}}(L, J, I, \tau, \tau)$ that gives a set of single-block CNNs $\{g_i\}_{i=1}^{n}$ such that*

$$\sum_{i=1}^{n} g_i(x) = \sum_{i=1}^{n_0} f_i(x).$$

*Such an architecture has*

$$L = O(L_0), J = O(\widetilde{J}), I = I_0, \tau = \tau_0.$$

*Furthermore, the fully connected layer of $f$ has nonzero elements only in the first row.*

## C PROOF OF CNN CLASS COVERING NUMBER

In this section, we prove a bound on the covering number of the convolutional neural network class used in Algorithm 1.

**Lemma 8.** *Given $\delta > 0$, the $\delta$-covering number of the neural network class $\mathcal{F}(M, L, J, I, \tau_1, \tau_2, V)$ satisfies*

$$\mathcal{N}(\delta, \mathcal{F}(M, L, J, I, \tau_1, \tau_2, V), \|\cdot\|_{\infty}) \leq \left(2(\tau_1 \vee \tau_2)\Lambda_1\delta^{-1}\right)^{\Lambda_2}, \tag{38}$$

*where*

$$\Lambda_1 = (M+3)JD(1 \vee \tau_2)(1 \vee \tau_1)\widetilde{\rho}\rho^+, \ \Lambda_2 = ML(J^2I + J) + JD + 1$$

*with $\widetilde{\rho} = \rho^M, \widetilde{\rho}^+ = 1 + ML\rho^+, \rho = (JI\tau_1)^L$ and $\rho^+ = (1 \vee JI\tau_1)^L$.*

*With a network architecture as stated in Theorem 1, we have*

$$\log \mathcal{N}(\delta, \mathcal{F}(M, L, J, I, \tau_1, \tau_2, V)) = O\left(\widetilde{M}\widetilde{J}^2 D^3 \log^5(\widetilde{M}\widetilde{J}) \log \frac{1}{\delta}\right),$$

where $O(\cdot)$ hides constant depending on $d$, $\alpha$, $\frac{2d}{\alpha p - d}$, $p$, $q$, $c_0$, $B$, $\omega$ and the surface area of $\mathcal{X}$.

## C.1 SUPPORTING LEMMAE AND PROOFS

Proposition 4 below provides an upper bound on the $L_\infty$-norm of a series of convolutional neural network blocks in terms of its architecture parameters, e.g. number of layers, number of channels, etc.

Let $J_m^{(i)}$ be the number of channels in $i$-th layer of the $m$-th block, and let $I_m^{(i)}$ be the filter size of $i$-th layer in the $m$-th block. $Q_{[i,j]}$ is defined as

$$Q_{[i,j]}(x) = \left(\text{Conv}_{\mathcal{W}_j,\mathcal{B}_j}\right) \circ \cdots \circ \left(\text{Conv}_{\mathcal{W}_i,\mathcal{B}_i}\right)(x).$$

**Proposition 4.** *For $m = 1, 2, \cdots, M$ and $x \in [-1,1]^D$, we have*

$$\left\|Q_{[1,m]}(x)\right\|_\infty \leq (1 \vee \tau_1) \left(\prod_{j=1}^{m}\prod_{i=1}^{L_j} J_j^{(i-1)} I_j^{(i)} \tau_1\right)\left(1 + \sum_{k=1}^{m} L_k \prod_{i=1}^{L_k}(1 \vee J_k^{(i-1)} I_k^{(i)} \tau_1)\right).$$

*Proof.*

$$\left\|Q_{[1,m]}(x)\right\|_\infty$$
$$= \left\|\text{Conv}_{\mathcal{W}_m,\mathcal{B}_m}(Q_{[1,m-1]}(x))\right\|_\infty$$
$$\leq \prod_{i=1}^{L_m} J_m^{(i-1)} I_m^{(i)} \tau_1 \left\|Q_{[1,m-1]}(x)\right\|_\infty + \tau_1 L_m \prod_{i=1}^{L_m}(1 \vee J_m^{(i-1)} I_m^{(i)} \tau_1)$$
$$\leq \|P(x)\|_\infty \prod_{j=1}^{m}\prod_{i=1}^{L_j} J_j^{(i-1)} I_j^{(i)} \tau_1 + \tau_1 \sum_{k=1}^{m} L_k \prod_{i=1}^{L_k}(1 \vee J_k^{(i-1)} I_k^{(i)} \tau_1) \prod_{l=j+1}^{m}\prod_{i=1}^{L_l} J_l^{(i-1)} I_l^{(i)} \tau_1$$
$$\leq \|x\|_\infty \prod_{j=1}^{m}\prod_{i=1}^{L_j} J_j^{(i-1)} I_j^{(i)} \tau_1 + \tau_1 \sum_{k=1}^{m} L_k \prod_{i=1}^{L_k}(1 \vee J_k^{(i-1)} I_k^{(i)} \tau_1) \prod_{l=j+1}^{m}\prod_{i=1}^{L_l} J_l^{(i-1)} I_l^{(i)} \tau_1$$
$$\leq (1 \vee \tau_1) \left(\prod_{j=1}^{m}\prod_{i=1}^{L_j} J_j^{(i-1)} I_j^{(i)} \tau_1\right)\left(1 + \sum_{k=1}^{m} L_k \prod_{i=1}^{L_k}(1 \vee J_k^{(i-1)} I_k^{(i)} \tau_1)\right),$$

where the first two inequalities are obtained by applying Proposition 9 from Oono & Suzuki [54] recursively. □

Lemma 9 quantifies the sensitivity of a CNN with respect to small changes in its weight parameters. This will be used to create a discrete covering for the CNN class.

**Lemma 9.** *For $f, f' \in \mathcal{F}(M, L, J, I, \tau_1, \tau_2, V)$ such that for $\epsilon > 0$, $\|W - W'\|_\infty \leq \epsilon$, $\|b - b'\|_\infty \leq \epsilon$, $\left\|\mathcal{W}_m^{(l)} - \mathcal{W}_m^{(l)'}\right\|_\infty \leq \epsilon$ and $\left\|\mathcal{B}_m^{(l)} - \mathcal{B}_m^{(l)'}\right\|_\infty \leq \epsilon$ for all $m$ and $l$, where $(W, b, \{\{(\mathcal{W}_m^{(l)}, \mathcal{B}_m^{(l)})\}_{l=1}^{L_m}\}_{m=1}^M)$ and $(W', b', \{\{(\mathcal{W}_m^{(l)'}, \mathcal{B}_m^{(l)'})\}_{l=1}^{L_m}\}_{m=1}^M)$ are the parameters of $f$ and $f'$ respectively, we have*

$$\|f - f'\|_\infty \leq \Lambda_1 \epsilon,$$

*where $\Lambda_1$ is defined in Lemma 8.*

*Proof.* For any $x \in [-1,1]^D$,

$$|f(x) - f'(x)|$$
$$= |W \otimes Q(x) + b - W' \otimes Q'(x) - b'|$$
$$= |(W - W') \otimes Q(x) + b - b' + W' \otimes (Q(x) - Q'(x))|$$
$$= |(W - W') \otimes Q(x) + b - b' + W' \otimes (Q(x) - \text{Conv}_{\mathcal{W}_M,\mathcal{B}_M}(Q'(x)) + \text{Conv}_{\mathcal{W}_M,\mathcal{B}_M}(Q'(x)) - Q'(x))|$$
$$= \left|(W - W') \otimes Q(x) + b - b' + \sum_{m=1}^{M} W' \otimes Q_{[m+1,M]} \circ \left(\text{Conv}_{\mathcal{W}_m,\mathcal{B}_m} - \text{Conv}_{\mathcal{W}_m',\mathcal{B}_m'}\right) \circ Q'_{[0,m-1]}\right|$$

$$\leq \left|(W - W') \otimes Q(x;\theta) + b - b'\right| + \sum_{m=1}^{M} \left|W' \otimes Q_{[m+1,M]} \circ \left(\mathrm{Conv}_{\mathcal{W}_m,\mathcal{B}_m} - \mathrm{Conv}_{\mathcal{W}'_m,\mathcal{B}'_m}\right) \circ Q'_{[0,m-1]}\right|$$

$$\overset{(a)}{\leq} (3+M)JD(1 \vee \tau_1)(1 \vee \tau_2)\left(\prod_{j=1}^{M}\prod_{i=1}^{L_j} J_j^{(i-1)}I_j^{(i)}\tau_1\right)\left(1 + \sum_{k=1}^{M} L_k \prod_{i=1}^{L_k}(1 \vee J_k^{(i-1)}I_k^{(i)}\tau_1)\right)\epsilon,$$

where (a) is obtained through the following reasoning.

The first term in (a) can be bounded as

$$|(W - W') \otimes Q(x) + b - b'|$$
$$\leq (\|W\|_0 + \|W'\|_0)\|W - W'\|_\infty \|Q(x)\|_\infty + \|b - b'\|_\infty$$
$$\leq 2JD\epsilon \|Q(x)\|_\infty + \epsilon$$
$$\leq 3JD\epsilon \|Q(x)\|_\infty$$
$$\leq 3JD \max\{1, \tau_1\}\left(\prod_{j=1}^{M}\prod_{i=1}^{L_j} J_j^{(i-1)}I_j^{(i)}\tau_1\right)\left(1 + \sum_{k=1}^{M} L_k \prod_{i=1}^{L_k}(1 \vee J_k^{(i-1)}I_k^{(i)}\tau_1)\right)\epsilon,$$

where the first inequality uses Proposition 8 from Oono & Suzuki [54] and the last inequality is obtained by invoking Proposition 4.

For the second term in (a), it is true that for any $m = 1, \cdots, M$, we have

$$\left|W' \otimes Q_{[m+1,M]} \circ \left(\mathrm{Conv}_{\mathcal{W}_m,\mathcal{B}_m} - \mathrm{Conv}_{\mathcal{W}'_m,\mathcal{B}'_m}\right) \circ Q'_{[1,m-1]}\right|$$

$$\overset{(b)}{\leq} \|W'\|_0 \tau_2 \left\|Q_{[m+1,M]} \circ \left(\mathrm{Conv}_{\mathcal{W}_m,\mathcal{B}_m} - \mathrm{Conv}_{\mathcal{W}'_m,\mathcal{B}'_m}\right) \circ Q'_{[1,m-1]}\right\|_\infty$$

$$\overset{(c)}{\leq} JD\tau_2 \left(\prod_{j=m+1}^{M}\prod_{i=1}^{L_j} J_j^{(i-1)}I_j^{(i)}\tau_1\right)\left\|\left(\mathrm{Conv}_{\mathcal{W}_m,\mathcal{B}_m} - \mathrm{Conv}_{\mathcal{W}'_m,\mathcal{B}'_m}\right) \circ Q'_{[1,m-1]}\right\|_\infty$$

$$\overset{(d)}{\leq} JD\tau_2 \left(\prod_{j=m+1}^{M}\prod_{i=1}^{L_j} J_j^{(i-1)}I_j^{(i)}\tau_1\right)\left(\prod_{i=1}^{L_m} J_m^{(i-1)}I_m^{(i)}\tau_1 \left\|Q'_{[1,m-1]}\right\|_\infty \epsilon\right)$$

$$\overset{(e)}{\leq} JD\tau_2 \left(\prod_{j=m+1}^{M}\prod_{i=1}^{L_j} J_j^{(i-1)}I_j^{(i)}\tau_1\right)\left(\prod_{i=1}^{L_m} J_m^{(i-1)}I_m^{(i)}\tau_1\right)$$
$$(1 \vee \tau_1)\left(\prod_{j=1}^{m}\prod_{i=1}^{L_j} J_j^{(i-1)}I_j^{(i)}\tau_1\right)\left(1 + \sum_{k=1}^{m} L_k \prod_{i=1}^{L_k}(1 \vee J_k^{(i-1)}I_k^{(i)}\tau_1)\right)\epsilon$$

$$\leq JD\tau_2 \left(\prod_{j=1}^{M}\prod_{i=1}^{L_j} J_j^{(i-1)}I_j^{(i)}\tau_1\right)(1 \vee \tau_1)\left(1 + \sum_{k=1}^{M} L_k \prod_{i=1}^{L_k}(1 \vee J_k^{(i-1)}I_k^{(i)}\tau_1)\right)\epsilon,$$

where (b) is by Proposition 7 from Oono & Suzuki [54], (c) is by Proposition 2 and 4 from Oono & Suzuki [54], (d) is by Proposition 2 and 5 from Oono & Suzuki [54], and (e) is obtained by invoking Proposition 4. □

## C.2 PROOF OF LEMMA 8

*Proof of Lemma 8.* We grid the range of each parameter into subsets with width $\Lambda_1^{-1}\delta$, so there are at most $2(\tau_1 \vee \tau_2)\Lambda_1\delta^{-1}$ different subsets for each parameter. In total, there are $\left(2(\tau_1 \vee \tau_2)\Lambda_1\delta^{-1}\right)^{\Lambda_2}$ bins in the grid. For any $f, f' \in \mathcal{F}(M, L, J, I, \tau_1, \tau_2, V)$ within the same grid, by Lemma 9, we have $\|f - f'\|_\infty \leq \delta$. We can construct the $\epsilon$-covering with cardinality $\left(2(\tau_1 \vee \tau_2)\Lambda_1\delta^{-1}\right)^{\Lambda_2}$ by selecting one neural network from each bin in the grid.

Taking log and plugging in the network architecture parameters in Lemma 1, we have
$$\log \mathcal{N}(\delta, \mathcal{F}(M, L, J, I, \tau_1, \tau_2, V), \|\cdot\|_\infty) = O\left(\Lambda_2 \log\left((\tau_1 \vee \tau_2)\Lambda_1\delta^{-1}\right)\right)$$

$$\leq O\left(\widetilde{M}DD^2\widetilde{J}^2\log(\widetilde{M}\widetilde{J})\log^2(\widetilde{M}\widetilde{J})\log^2(\widetilde{M}\widetilde{J})\log\frac{1}{\delta}\right)$$

$$= O\left(\widetilde{M}\widetilde{J}^2D^3\log^5(\widetilde{M}\widetilde{J})\log\frac{1}{\delta}\right),$$

where the inequality is due to $\Lambda_2 = O(\widetilde{M}DD^2\widetilde{J}^2\log(\widetilde{M}\widetilde{J}))$. By plugging in the choice of $\tau_1$, $\rho = (1/2)^L M^{-1} \leq M^{-1}$, so $\widetilde{\rho} = (1+M^{-1})^M \leq e$. Moreover, $\widetilde{\rho}^+ = 1 + ML$. □

## D    STATISTICAL RESULT OF CNN-BESOV APPROXIMATION (LEMMA 10)

In this section, we derive the statistical estimation error for using a CNN empirical MSE minimizer to estimate a Besov ground truth function over an i.i.d. dataset. We need to choose the appropriate CNN architecture and size in order to balance the approximation error from Theorem 1 and variance. Thsi statistical estimation error can be decomposed into the error of using CNN to approximate Besov function (Theorem 1), terms that grow with the covering number of our CNN class, and the error of using the discrete covering to approximate our CNN class.

In Theorem 2, we expand the estimation error $\widehat{v}^\pi - v^\pi$ over time steps and upper-bound the amount of estimation error in each time step with Lemma 10. Details of Theorem 2 are in Appendix A.

**Lemma 10.** *Let $\mathcal{X}$ be a $d$-dimensional compact Riemannian manifold that satisfies Assumption 1. We are given a function $f_0 \in \mathcal{B}^\alpha_{p,q}(\mathcal{X})$, where $s,p,q$ satisfies Assumption 2. We are also given samples $S_n = \{(x_i, y_i)\}_{i=1}^n$, where $x_i$ are i.i.d. sampled from a distribution $\mathcal{P}_x$ on $\mathcal{X}$ and $y_i = f_0(x_i) + \zeta_i$. $\zeta_i$'s are i.i.d. sub-Gaussian random noise with variance $\sigma^2$, uncorrelated with $x_i$'s. If we compute an estimator*

$$\widehat{f}_n = \arg\min_{f \in \mathcal{F}} \frac{1}{n}\sum_{i=1}^n \left(f(x_i) - y_i\right)^2,$$

*with the neural network class $\mathcal{F} = \mathcal{F}(M, L, J, I, \tau_1, \tau_2, V)$ such that*

$$L = O(\log n + D + \log D),\ J = O(D),\ \tau_1 = O(1),\ \log\tau_2 = O(\log^2 n + D\log n),$$

$$M = O(n^{\frac{d}{2\alpha+d}}),\ V = \|f_0\|_\infty, \tag{39}$$

*with any integer $I \in [2, D]$ and $\widetilde{M}, \widetilde{J} > 0$ satisfying $\widetilde{M}\widetilde{J} = O(n^{\frac{d}{2\alpha+2d}})$, then we have*

$$\mathbb{E}\left[\int_{\mathcal{X}}\left(\widehat{f}_n(x) - f_0(x)\right)^2 \mathrm{d}\mathcal{P}_x(x)\right] \leq c\left(V_{\mathcal{F}}^2 + \sigma^2\right)n^{-\frac{2\alpha}{2\alpha+d}}\log^5 n, \tag{40}$$

*where $V_{\mathcal{F}} = \|f_0\|_\infty$ and the expectation is taken over the training sample $S_n$, and $c$ is a constant depending on $D^{\frac{6\alpha}{2\alpha+2d}}$, $d$, $\alpha$, $\frac{2d}{\alpha p-d}$, $p$, $q$, $c_0$, $B$, $\omega$ and the surface area of $\mathcal{X}$. $O(\cdot)$ hides constant depending on $d$, $\alpha$, $\frac{2d}{\alpha p-d}$, $p$, $q$, $c_0$, $B$, $\omega$ and the surface area of $\mathcal{X}$.*

First, note that the nonparametric regression error can be decomposed into two terms:

$$\mathbb{E}\left[\int_{\mathcal{X}}\left(\widehat{f}_n(x) - f_0(x)\right)^2 d\mathcal{D}_x(x)\right] = \underbrace{2\mathbb{E}\left[\frac{1}{n}\sum_{i=1}^n(\widehat{f}_n(x_i) - f_0(x_i))^2\right]}_{T_1}$$

$$+ \underbrace{\mathbb{E}\left[\int_{\mathcal{X}}\left(\widehat{f}_n(x) - f_0(x)\right)^2 d\mathcal{D}_x(x)\right] - 2\mathbb{E}\left[\frac{1}{n}\sum_{i=1}^n(\widehat{f}_n(x_i) - f_0(x_i))^2\right]}_{T_2},$$

where $T_1$ reflects the squared bias of using neural networks to approximate ground truth $f_0$, which is related to Theorem 1, and $T_2$ is the variance term.

### D.1    SUPPORTING LEMMAE

**Lemma 11** (Lemma 5 in Chen et al. [7]). Fix the neural network class $\mathcal{F}(M, L, J, I, \tau_1, \tau_2, V)$. For any constant $\delta \in (0, 2V)$, we have

$$T_1 \leq 4\inf_{f \in \mathcal{F}(M,L,J,I,\tau_1,\tau_2,V)}\int_{\mathcal{X}}(f(x) - f_0(x))^2 d\mathcal{P}_x(x)$$

$$+ 48\sigma^2 \frac{\log \mathcal{N}(\delta, \mathcal{F}(M, L, J, I, \tau_1, \tau_2, V), \|\cdot\|_\infty) + 2}{n}$$

$$+ (8\sqrt{6}\sqrt{\frac{\log \mathcal{N}(\delta, \mathcal{F}(M, L, J, I, \tau_1, \tau_2, V), \|\cdot\|_\infty) + 2}{n}} + 8)\sigma\delta,$$

where $\mathcal{N}(\delta, \mathcal{F}(M, L, J, I, \tau_1, \tau_2, V), \|\cdot\|_\infty)$ denotes the $\delta$-covering number of $\mathcal{F}(M, L, J, I, \tau_1, \tau_2, V)$ with respect to the $\ell_\infty$ norm, i.e., there exists a discretization of $\mathcal{F}(M, L, J, I, \tau_1, \tau_2, V)$ into $\mathcal{N}(\delta, \mathcal{F}(M, L, J, I, \tau_1, \tau_2, V), \|\cdot\|_\infty)$ distinct elements, such that for any $f \in \mathcal{F}$, there is $\bar{f}$ in the discretization satisfying $\|\bar{f} - f\|_\infty \leq \epsilon$.

**Lemma 12** (Lemma 6 in Chen et al. [7]). *For any constant $\delta \in (0, 2R)$, $T_2$ satisfies*

$$T_2 \leq \frac{104V^2}{3n} \log \mathcal{N}(\delta/4V, \mathcal{F}(M, L, J, I, \tau_1, \tau_2, V), \|\cdot\|_\infty) + \left(4 + \frac{1}{2V}\right)\delta.$$

### D.2 Proof of Lemma 10

*Proof of Lemma 10.* Recall that the bias and variance decomposition of $\mathbb{E}\left[\int_{\mathcal{X}} \left(\widehat{f}_n(x) - f_0(x)\right)^2 d\mathcal{P}_x(x)\right]$ as

$$\mathbb{E}\left[\int_{\mathcal{X}} \left(\widehat{f}_n(x) - f_0(x)\right)^2 d\mathcal{P}_x(x)\right] = \underbrace{\mathbb{E}\left[\frac{2}{n}\sum_{i=1}^{n}(\widehat{f}_n(x_i) - f_0(x_i))^2\right]}_{T_1}$$

$$+ \underbrace{\mathbb{E}\left[\int_{\mathcal{X}} \left(\widehat{f}_n(x) - f_0(x)\right)^2 d\mathcal{P}_x(x)\right] - \mathbb{E}\left[\frac{2}{n}\sum_{i=1}^{n}(\widehat{f}_n(x_i) - f_0(x_i))^2\right]}_{T_2}.$$

Applying the upper bounds of $T_1$ and $T_2$ in Lemmas 11 and 12 respectively, we can derive

$$\mathbb{E}\left[\int_{\mathcal{X}} \left(\widehat{f}_n(x) - f_0(x)\right)^2 d\mathcal{P}_x(x)\right] \leq 4\inf_{f \in \mathcal{F}(M, L, J, I, \tau_1, \tau_2, V)} \int_{\mathcal{X}} (f(x) - f_0(x))^2 d\mathcal{P}_x(x)$$

$$+ 48\sigma^2 \frac{\log \mathcal{N}(\delta, \mathcal{F}(M, L, J, I, \tau_1, \tau_2, V), \|\cdot\|_\infty) + 2}{n}$$

$$+ 8\sqrt{6}\sqrt{\frac{\log \mathcal{N}(\delta, \mathcal{F}(M, L, J, I, \tau_1, \tau_2, V), \|\cdot\|_\infty) + 2}{n}}\sigma\delta$$

$$+ \frac{104V_{\mathcal{F}}^2}{3n} \log \mathcal{N}(\delta/4V, \mathcal{F}(M, L, J, I, \tau_1, \tau_2, V), \|\cdot\|_\infty)$$

$$+ \left(4 + \frac{1}{2V_{\mathcal{F}}} + 8\sigma\right)\delta.$$

We need there to exist a network in $\mathcal{F}(M, L, J, I, \tau_1, \tau_2, V)$ which can yield a function $f$ satisfying $\|f - f_0\|_\infty \leq \epsilon$ for $\epsilon \in (0, 1)$. $\epsilon$ will be chosen later to balance the bias-variance tradeoff. In order to achieve such $\epsilon$-error, we set $\widetilde{M}\widetilde{J} = \epsilon^{-d/\alpha}$, so we now have our network architecture as specified in Theorem 1 in terms of $\epsilon$. Then, we can use the parameters in this architecture to invoke the upper bound of the covering number in Lemma 8:

$$\log \mathcal{N}(\delta, \mathcal{F}(M, L, J, I, \tau_1, \tau_2, V), \|\cdot\|_\infty) = O\left(\Lambda_2 \log\left((\tau_1 \vee \tau_2)\Lambda_1\delta^{-1}\right)\right)$$

$$\leq O\left(\widetilde{M}\widetilde{J}^2 D^3 \log^5(\widetilde{M}\widetilde{J}) \log\frac{1}{\delta}\right)$$

$$= O\left(\epsilon^{-d/\alpha} D^3 \log^5 \epsilon \log\frac{1}{\delta}\right),$$

where $O(\cdot)$ hides constant depending on $\log D, d, \alpha, \frac{2d}{\alpha p - d}, p, q, c_0, B, \omega$ and the surface area of $\mathcal{X}$.

Plugging it in, we have

$$\mathbb{E}\left[\int_{\mathcal{X}} \left(\widehat{f}_n(x) - f_0(x)\right)^2 d\mathcal{D}_x(x)\right] \leq 4\epsilon^2 + \frac{48\sigma^2}{n}\left(c''\epsilon^{-d/\alpha} D^3 \log^5 \epsilon \log\frac{1}{\delta} + 2\right)$$

$$+ 8\sqrt{6c''}\sqrt{\frac{\epsilon^{-d/\alpha}D^3 \log^5 \epsilon \log \frac{1}{\delta}}{n}}\sigma\delta$$

$$+ \frac{104V^2}{3n}\epsilon^{-d/\alpha}D^3 \log^5 \epsilon \log \frac{1}{\delta}$$

$$+ \left(4 + \frac{1}{2V_\mathcal{F}} + 8\sigma\right)\delta$$

$$= \widetilde{O}\left(\epsilon^2 + \frac{V_\mathcal{F}^2 + \sigma^2}{n}\epsilon^{-\frac{d}{\alpha}}D^3 \log^5 \epsilon \log \frac{1}{\delta}\right.$$

$$\left. + \sigma\delta\sqrt{\frac{\epsilon^{-\frac{d}{\alpha}}D^3 \log^5 \epsilon \log \frac{1}{\delta}}{n}} + \sigma\delta + \frac{\sigma^2}{n}\right). \qquad (41)$$

Finally we choose $\epsilon$ to satisfy $\epsilon^2 = \frac{1}{n}D^3\epsilon^{-\frac{d}{\alpha}}$, which gives $\epsilon = D^{\frac{3\alpha}{2\alpha+d}}n^{-\frac{\alpha}{2\alpha+d}}$. It suffices to pick $\delta = \frac{1}{n}$. Substituting both $\epsilon$ and $\delta$ into (41), we deduce the desired estimation error bound

$$\mathbb{E}\left[\int_\mathcal{X}\left(\widehat{f}_n(x) - f_0(x)\right)^2 d\mathcal{D}_x(x)\right] \leq c(V_\mathcal{F}^2 + \sigma^2)n^{-\frac{2\alpha}{2\alpha+d}}\log^5 n,$$

where constant $c$ depends on $D^{\frac{6\alpha}{2\alpha+d}}$, $d$, $\alpha$, $\frac{2d}{\alpha p - d}$, $p$, $q$, $c_0$, $B$, $\omega$ and the surface area of $\mathcal{X}$. $\qquad\square$

# E  A RESULT FOR FEED-FORWARD RELU NEURAL NETWORK

## E.1  FEED-FORWARD RELU NEURAL NETWORK

We consider multi-layer ReLU (Rectified Linear Unit) neural networks [25]. ReLU activation is popular in computer vision, natural language processing, etc. because the vanishing gradient issue is less severe with it, which is nonetheless common with its counterparts like sigmoid or hyperbolic tangent activation [25, 27]. An $L$-layer ReLU neural network can be expressed as

$$f(x) = W_L \cdot \text{ReLU}(W_{L-1} \cdots \text{ReLU}(W_1 x + b_1) \cdots + b_{L-1}) + b_L, \qquad (42)$$

in which $W_1, \cdots, W_L$ and $b_1, \cdots, b_L$ are weight matrices and vectors and $\text{ReLU}(\cdot)$ is the entrywise rectified linear unit, i.e. $\text{ReLU}(a) = \max\{0, a\}$. The width of a neural network is defined as the number of neurons in its widest layer. For notational simplicity, we define a class of neural networks

$$\mathcal{F}(L, p, I, \tau, V) = \{f \mid f(x) \text{ in the form (42) with } L \text{ layers and width at most } p,$$

$$\|f\|_\infty \leq V, \ \sum_{i=1}^{L}\|W_i\|_0 + \|b_i\|_0 \leq I, \ \|W_i\|_{\infty,\infty} \leq \tau, \ \|b_i\|_\infty \leq \tau \text{ for } i = 1, \cdots, L\}.$$

$$(43)$$

## E.2  POLICY EVALUATION ERROR AND ITS PROOF

From this point, we denote the function class $\mathcal{F}(L, p, I, \tau, V)$, whose parameters $L, p, I, \tau, V$ are chosen according to Theorem 3, with the shorthand $\mathcal{F}$. In this section, this $\mathcal{F}$ is used in Algorithm 1, instead of the CNN class in (11).

**Theorem 3.** Suppose Assumption 1 and 2 hold. By choosing

$$L = O\left(\log K\right), \quad p = O\left(K^{\frac{d}{2\alpha+d}}\right), \quad I = O\left(K^{\frac{d}{2\alpha+d}}\log K\right),$$

$$\tau = \max\{B, H, \sqrt{d}, \omega^2\}, \quad V = H$$

$$(44)$$

in Algorithm 1, in which $O(\cdot)$ hides factors depending on $\alpha$, $d$ and $\log D$, we have

$$\mathbb{E}\left|v^\pi - \widehat{v}^\pi\right| \leq CH^2\kappa\left(K^{-\frac{\alpha}{2\alpha+d}} + \sqrt{D/K}\right)\log^{\frac{3}{2}} K, \qquad (45)$$

in which the expectation is taken over the data, and $C$ is a constant depending on $\log D$, $\alpha$, $B$, $d$, $\omega$, the surface area of $\mathcal{X}$ and $c_0$. The distributional mismatch is captured by

$$\kappa = \frac{1}{H}\sum_{h=1}^{H}\sqrt{\chi_\mathcal{Q}^2(q_h^\pi, q_h^{\pi_0}) + 1},$$

in which $\mathcal{Q}$ is the Minkowski sum between the ReLU function class and the Besov function class, i.e., $\mathcal{Q} = \{f + g \mid f \in \mathcal{B}_{p,q}^\alpha(\mathcal{X}), g \in \mathcal{F}\}$.

*Proof of Theorem 3.* The goal is to bound

$$\mathbb{E}\left|\widehat{v}^{\pi} - v^{\pi}\right| = \mathbb{E}\left|\int_{\mathcal{X}}\left(Q_1^{\pi} - \widehat{Q}_1^{\pi}\right)(s, a)\,\mathrm{d}q_1^{\pi}(s, a)\right| \leq \mathbb{E}\left[\int_{\mathcal{X}}\left|Q_1^{\pi} - \widehat{Q}_1^{\pi}\right|(s, a)\,\mathrm{d}q_1^{\pi}(s, a)\right].$$

To get an expression for that, we first expand it recursively. To illustrate the recursive relation, we examine the quantity at step $h$:

$$\mathbb{E}\left[\int_{\mathcal{X}}\left|Q_h^{\pi} - \widehat{Q}_h^{\pi}\right|(s, a)\,\mathrm{d}q_h^{\pi}(s, a)\right]$$

$$= \mathbb{E}\left[\int_{\mathcal{X}}\left|\mathcal{T}_h^{\pi}Q_{h+1}^{\pi} - \widehat{\mathcal{T}}_h^{\pi}\left(\widehat{Q}_{h+1}^{\pi}\right)\right|(s, a)\,\mathrm{d}q_h^{\pi}(s, a)\right]$$

$$\leq \mathbb{E}\left[\int_{\mathcal{X}}\left|\mathcal{T}_h^{\pi}Q_{h+1}^{\pi} - \mathcal{T}_h^{\pi}\widehat{Q}_{h+1}^{\pi}\right|(s, a)\,\mathrm{d}q_h^{\pi}(s, a)\right] + \mathbb{E}\left[\int_{\mathcal{X}}\left|\mathcal{T}_h^{\pi}\widehat{Q}_{h+1}^{\pi} - \widehat{\mathcal{T}}_h^{\pi}\left(\widehat{Q}_{h+1}^{\pi}\right)\right|(s, a)\,\mathrm{d}q_h^{\pi}(s, a)\right]$$

$$= \mathbb{E}\left[\int_{\mathcal{X}}\left|Q_{h+1}^{\pi} - \widehat{Q}_{h+1}^{\pi}\right|(s, a)\,\mathrm{d}q_{h+1}^{\pi}(s, a)\right]$$

$$+ \mathbb{E}\left[\mathbb{E}\left[\int_{\mathcal{X}}\left|\mathcal{T}_h^{\pi}\widehat{Q}_{h+1}^{\pi} - \widehat{\mathcal{T}}_h^{\pi}\left(\widehat{Q}_{h+1}^{\pi}\right)\right|(s, a)\,\mathrm{d}q_h^{\pi}(s, a) \mid \mathcal{D}_{h+1}, \cdots, \mathcal{D}_H\right]\right]$$

$$\overset{(a)}{\leq} \mathbb{E}\left[\int_{\mathcal{X}}\left|Q_{h+1}^{\pi} - \widehat{Q}_{h+1}^{\pi}\right|(s, a)\,\mathrm{d}q_{h+1}^{\pi}(s, a)\right]$$

$$+ \mathbb{E}\left[\mathbb{E}\left[\sqrt{\int_{\mathcal{X}}\left(\mathcal{T}_h^{\pi}\widehat{Q}_{h+1}^{\pi} - \widehat{\mathcal{T}}_h^{\pi}\left(\widehat{Q}_{h+1}^{\pi}\right)\right)^2(s, a)\,\mathrm{d}q_h^{\pi_0}(s, a)}\sqrt{\chi_{\mathcal{Q}}^2(q_h^{\pi}, q_h^{\pi_0}) + 1} \mid \mathcal{D}_{h+1}, \cdots, \mathcal{D}_H\right]\right]$$

$$\overset{(b)}{\leq} \mathbb{E}\left[\int_{\mathcal{X}}\left|Q_{h+1}^{\pi} - \widehat{Q}_{h+1}^{\pi}\right|(s, a)\,\mathrm{d}q_{h+1}^{\pi}(s, a)\right]$$

$$+ \sqrt{\mathbb{E}\left[\mathbb{E}\left[\int_{\mathcal{X}}\left(\mathcal{T}_h^{\pi}\widehat{Q}_{h+1}^{\pi} - \widehat{\mathcal{T}}_h^{\pi}\left(\widehat{Q}_{h+1}^{\pi}\right)\right)^2(s, a)\,\mathrm{d}q_h^{\pi_0}(s, a) \mid \mathcal{D}_{h+1}, \cdots, \mathcal{D}_H\right]\right]}\sqrt{\chi_{\mathcal{Q}}^2(q_h^{\pi}, q_h^{\pi_0}) + 1}$$

$$\overset{(c)}{\leq} \int_{\mathcal{X}}\left|Q_{h+1}^{\pi} - \widehat{Q}_{h+1}^{\pi}\right|(s, a)\,\mathrm{d}q_{h+1}^{\pi}(s, a) + \sqrt{c(5H^2)\left(K^{-\frac{2\alpha}{2\alpha+d}} + \frac{D}{K}\right)\log^3 K}\sqrt{\chi_{\mathcal{Q}}^2(q_h^{\pi}, q_h^{\pi_0}) + 1}$$

$$\leq \int_{\mathcal{X}}\left|Q_{h+1}^{\pi} - \widehat{Q}_{h+1}^{\pi}\right|(s, a)\,\mathrm{d}q_{h+1}^{\pi}(s, a) + CH\left(K^{-\frac{\alpha}{2\alpha+d}} + \sqrt{\frac{D}{K}}\right)\log^{3/2} K\sqrt{\chi_{\mathcal{Q}}^2(q_h^{\pi}, q_h^{\pi_0}) + 1},$$

where $C$ denotes a (varying) constant depending on $\log D$, $\alpha$, $B$, $d$, $\omega$, the surface area of $\mathcal{X}$ and $c_0$.

In (a), note $\mathcal{T}_h^{\pi}\widehat{Q}_{h+1}^{\pi} \in \mathcal{B}_{p,q}^{\alpha}(\mathcal{X})$ by Assumption 2 and $-\widehat{\mathcal{T}}_h^{\pi}\left(\widehat{Q}_{h+1}^{\pi}\right) \in \mathcal{F}$ by our algorithm, so $\mathcal{T}_h^{\pi}\widehat{Q}_{h+1}^{\pi} - \widehat{\mathcal{T}}_h^{\pi}\left(\widehat{Q}_{h+1}^{\pi}\right) \in \mathcal{Q}$. Then we obtain this inequality by invoking the following lemma.

In (b), we use Jensen's inequality and the fact that square root is concave.

To obtain (c), we invoke the following lemma, which provides an upper bound on the regression error.

Specifically, we will use Lemma 13 when conditioning on $\mathcal{D}_{h+1}, \cdots, \mathcal{D}_H$, i.e. the data from time step $h + 1$ to time step $H$. Note that after conditioning, $\mathcal{T}_h^{\pi}\widehat{Q}_{h+1}^{\pi}$ becomes measurable and deterministic with respect to $\mathcal{D}_{h+1}, \cdots, \mathcal{D}_H$. Also, $\mathcal{D}_{h+1}, \cdots, \mathcal{D}_H$ are independent from $\mathcal{D}_h$, which we use in the regression at step $h$.

To justify our use of Lemma 13, we need to cast our problem into a regression problem described in the lemma. Since $\{(s_{h,k}, a_{h,k})\}_{k=1}^K$ are i.i.d. from $q_h^{\pi_0}$, we can view them as the samples $x_i$'s in the lemma. We can view $\mathcal{T}_h^{\pi}\widehat{Q}_{h+1}^{\pi}$, which is measurable under our conditioning, as $f_0$ in the lemma. Furthermore, we let

$$\zeta_{h,k} := r_{h,k} + \int_{\mathcal{A}}\widehat{Q}_{h+1}^{\pi}(s_{h,k}', a)\pi(a \mid s_{h,k}')\,\mathrm{d}a - \mathcal{T}_h^{\pi}\widehat{Q}_{h+1}^{\pi}(s_{h,k}, a_{h,k}).$$

In order to invoke Lemma 13 under the conditioning on $\mathcal{D}_{h+1}, \cdots, \mathcal{D}_H$, we need to verify whether three conditions are satisfied (conditioning on $\mathcal{D}_{h+1}, \cdots, \mathcal{D}_H$):

1. Sample $\{(s_{h,k}, a_{h,k})\}_{k=1}^K$ are i.i.d;

2. Sample $\{(s_{h,k}, a_{h,k})\}_{k=1}^K$ and noise $\{\zeta_{h,k}\}_{k=1}^K$ are uncorrelated;

3. Noise $\{\zeta_{h,k}\}_{k=1}^K$ are independent, zero-mean, subgaussian random variables.

In our setting, $\{(s_{h,k}, a_{h,k})\}_{k=1}^K$ are i.i.d. from $q_h^{\pi_0}$. Due to the time-inhomogeneous setting, they are independent from $\mathcal{D}_{h+1}, \cdots, \mathcal{D}_H$, so $\{(s_{h,k}, a_{h,k})\}_{k=1}^K$ are still i.i.d. under our conditioning. Thus, Condition 1 is clearly satisfied.

We may observe that under our conditioning, the transition from $(s_{h,k}, a_{h,k})$ to $s'_{h,k}$ is the only source of randomness in $\zeta_{h,k}$, besides $(s_{h,k}, a_{h,k})$ itself. The distribution of $(s_{h,k}, a_{h,k}, s'_{h,k})$ is actually the product distribution between $P_h(\cdot|s_{h,k}, a_{h,k})$ and $q_h^{\pi_0}$, so a function of $s'_{h,k}$, generated from the transition distribution $P_h(\cdot|s_{h,k}, a_{h,k})$, is uncorrelated with $(s_{h,k}, a_{h,k})$. Thus, $(s_{h,k}, a_{h,k})$'s are uncorrelated with $\zeta_{h,k}$'s under our conditioning, and Condition 2 is satisfied.

Condition 3 can also be easily verified. Under our conditioning, the randomness in $\zeta_{h,k}$ only comes from $(s_{h,k}, a_{h,k}, s'_{h,k}, r_{h,k})$, which are independent from $(s_{h,k'}, a_{h,k'}, s'_{h,k'}, r_{h,k'})$ for any $k' \neq k$, so $\zeta_{h,k}$'s are independent from each other. As for the mean of $\zeta_{h,k}$,

$$\mathbb{E}[\zeta_{h,k} \mid \mathcal{D}_{h+1}, \cdots, \mathcal{D}_H]$$

$$= \mathbb{E}\left[r_{h,k} + \int_{\mathcal{A}} \widehat{Q}_{h+1}^{\pi}(s'_{h,k}, a)\pi(a \mid s'_{h,k})\,\mathrm{d}a - r_h(s_{h,k}, a_{h,k}) - \mathcal{P}_h^{\pi}\widehat{Q}_{h+1}^{\pi}(s_{h,k}, a_{h,k}) \mid \mathcal{D}_{h+1}, \cdots, \mathcal{D}_H\right]$$

$$= \mathbb{E}\Bigg[r_{h,k} - r_h(s_{h,k}, a_{h,k}) + \int_{\mathcal{A}} \widehat{Q}_{h+1}^{\pi}(s'_{h,k}, a)\pi(a \mid s'_{h,k})\,\mathrm{d}a$$

$$\qquad - \mathbb{E}_{s' \sim P_h(\cdot|s_{h,k}, a_{h,k})}\left[\int_{\mathcal{A}} \widehat{Q}_{h+1}^{\pi}(s', a)\pi(a \mid s')\,\mathrm{d}a \mid s_{h,k}, a_{h,k}, \mathcal{D}_{h+1}, \cdots, \mathcal{D}_H\right] \mid \mathcal{D}_{h+1}, \cdots, \mathcal{D}_H\Bigg]$$

$$= 0 + 0 = 0.$$

On the other hand, $\left\|\widehat{Q}_{h+1}^{\pi}\right\|_{\infty} \leq H$ almost surely, because it is a function in our ReLU network class $\mathcal{F}$. Thus, $\zeta_{h,k}$ is a bounded random variable with $\zeta_{h,k} \in [-2H, 2H]$ almost surely, so its variance is bounded by $4H^2$. Its boundedness also implies it is a subgaussian random variable. Thus, Condition 3 is also satisfied.

Hence, Lemma 13 proves, for step $h$ in our algorithm,

$$\mathbb{E}\left[\int_{\mathcal{X}} \left(\mathcal{T}_h^{\pi}\widehat{Q}_{h+1}^{\pi} - \widehat{\mathcal{T}}_h^{\pi}\left(\widehat{Q}_{h+1}^{\pi}\right)\right)^2 (s, a)\,\mathrm{d}q_h^{\pi_0}(s, a) \mid \mathcal{D}_{h+1}, \cdots, \mathcal{D}_H\right]$$

$$\leq c(H^2 + 4H^2)\left(K^{-\frac{2\alpha}{2\alpha+d}} + \frac{D}{K}\right)\log^3 K.$$

Note that this upper bound holds for any $\widehat{Q}_{h+1}^{\pi}$ or $\mathcal{D}_{h+1}, \cdots, \mathcal{D}_H$. The sole purpose of our conditioning is that we could view $\widehat{Q}_{h+1}^{\pi}$ as a measurable or deterministic function under the conditioning and then apply Lemma 13. Therefore,

$$\mathbb{E}\left[\mathbb{E}\left[\int_{\mathcal{X}} \left(\mathcal{T}_h^{\pi}\widehat{Q}_{h+1}^{\pi} - \widehat{\mathcal{T}}_h^{\pi}\left(\widehat{Q}_{h+1}^{\pi}\right)\right)^2 (s, a)\,\mathrm{d}q_h^{\pi_0}(s, a) \mid \mathcal{D}_{h+1}, \cdots, \mathcal{D}_H\right]\right]$$

$$\leq c(H^2 + 4H^2)\left(K^{-\frac{2\alpha}{2\alpha+d}} + \frac{D}{K}\right)\log^3 K.$$

Finally, we carry out the recursion from time step 1 to time step $H$, and the final result is

$$\mathbb{E}\left|v^{\pi} - \widehat{v}^{\pi}\right| \leq CH^2\left(K^{-\frac{\alpha}{2\alpha+d}} + \sqrt{\frac{D}{K}}\right)\log^{3/2} K\left(\frac{1}{H}\sum_{h=1}^H \sqrt{\chi_{\mathcal{Q}}^2(q_h^{\pi}, q_h^{\pi_0}) + 1}\right).$$

$\square$

### E.3 LEMMA 13 AND ITS PROOF

**Lemma 13.** *Let $\mathcal{X}$ be a $d$-dimensional compact Riemannian manifold isometrically embedded in $\mathbb{R}^D$ with reach $\omega$. There exists a constant $B > 0$ such that for any $x \in \mathcal{X}$, $|x_j| \leq B$ for all $j = 1, \cdots, D$. We are given a function $f_0 \in \mathcal{B}_{p,q}^\alpha(\mathcal{X})$ and samples $S_n = \{(x_i, y_i)\}_{i=1}^n$, where $x_i$ are i.i.d. sampled from a distribution $\mathcal{P}_x$ on $\mathcal{X}$ and $y_i = f_0(x_i) + \zeta_i$. $\zeta_i$'s are i.i.d. sub-Gaussian random noise with variance $\sigma^2$, uncorrelated with $x_i$'s. If we compute an estimator*

$$\widehat{f}_n = \arg\min_{f \in \mathcal{F}} \frac{1}{n} \sum_{i=1}^n (f(x_i) - y_i)^2 \,,$$

*with the neural network class $\mathcal{F} = \mathcal{F}(L, p, I, \tau, V)$ such that*

$$L = O\left(\log n\right), p = O\left(n^{\frac{d}{2\alpha+d}}\right), I = O\left(n^{\frac{d}{2\alpha+d}} \log n\right),$$

$$\tau = \max\{B, V_\mathcal{F}, \sqrt{d}, \omega^2\}, V = V_\mathcal{F}, \tag{46}$$

*then we have*

$$\mathbb{E}\left[\int_\mathcal{X} \left(\widehat{f}_n(x) - f_0(x)\right)^2 \mathrm{d}\mathcal{P}_x(x)\right] \leq c\left(V_\mathcal{F}^2 + \sigma^2\right)\left(n^{-\frac{2\alpha}{2\alpha+d}} + \frac{D}{n}\right)\log^3 n, \tag{47}$$

*where $V_\mathcal{F} = \|f_0\|_\infty$ and the expectation is taken over the training sample $S_n$, and $c$ is a constant depending on $\log D$, $\alpha$, $B$, $d$, $\omega$, the surface area of $\mathcal{X}$ and $c_0$.*

*Proof of Lemma 13.* Recall that the bias and variance decomposition of $\mathbb{E}\left[\int_\mathcal{X}\left(\widehat{f}_n(x) - f_0(x)\right)^2 d\mathcal{P}_x(x)\right]$ as

$$\mathbb{E}\left[\int_\mathcal{X}\left(\widehat{f}_n(x) - f_0(x)\right)^2 d\mathcal{P}_x(x)\right] = \underbrace{\mathbb{E}\left[\frac{2}{n}\sum_{i=1}^n(\widehat{f}_n(x_i) - f_0(x_i))^2\right]}_{T_1}$$

$$+ \underbrace{\mathbb{E}\left[\int_\mathcal{X}\left(\widehat{f}_n(x) - f_0(x)\right)^2 d\mathcal{P}_x(x)\right] - \mathbb{E}\left[\frac{2}{n}\sum_{i=1}^n(\widehat{f}_n(x_i) - f_0(x_i))^2\right]}_{T_2}.$$

Applying the upper bounds of $T_1$ and $T_2$ in Lemmas 11 and 12 respectively, we can derive

$$\mathbb{E}\left[\int_\mathcal{X}\left(\widehat{f}_n(x) - f_0(x)\right)^2 d\mathcal{P}_x(x)\right] \leq 4 \inf_{f \in \mathcal{F}(L,p,I,\tau,V)} \int_\mathcal{X} (f(x) - f_0(x))^2 d\mathcal{P}_x(x)$$

$$+ 48\sigma^2 \frac{\log\mathcal{N}(\delta, \mathcal{F}(L,p,I,\tau,V), \|\cdot\|_\infty) + 2}{n}$$

$$+ 8\sqrt{6}\sqrt{\frac{\log\mathcal{N}(\delta, \mathcal{F}(L,p,I,\tau,V), \|\cdot\|_\infty) + 2}{n}}\sigma\delta$$

$$+ \frac{104V_\mathcal{F}^2}{3n}\log\mathcal{N}(\delta/4V, \mathcal{F}(L,p,I,\tau,V), \|\cdot\|_\infty)$$

$$+ \left(4 + \frac{1}{2V_\mathcal{F}} + 8\sigma\right)\delta.$$

We need there to exist a network in $\mathcal{F}(L, p, I, \tau, V)$ which can yield a function $f$ satisfying $\|f - f_0\|_\infty \leq \epsilon$ for $\epsilon \in (0, 1)$. $\epsilon$ will be chosen later to balance the bias-variance tradeoff. By Lemma 2 of Nguyen-Tang et al. [51], in order to achieve such $\epsilon$-error, we need

$$L = O\left(\log\frac{1}{\epsilon}\right), p = O\left(\epsilon^{-\frac{d}{\alpha}}\right), I = O\left(\epsilon^{-\frac{d}{\alpha}}\log\frac{1}{\epsilon}\right),$$

$$\tau = \max\{B, V_\mathcal{F}, \sqrt{d}, \omega^2\}, V = V_\mathcal{F},$$

where $O(\cdot)$ hides factors of $\log D$, $\alpha$, $d$ and the surface area of $\mathcal{X}$, so we now have our network architecture as specified in Theorem 1 in terms of $\epsilon$. Then, we can use the architecture parameters in (13) to invoke the upper bound of the covering number in Lemma 7 of Chen et al. [7]:

$$\log\mathcal{N}(\delta, \mathcal{F}(L, p, I, \tau, V), \|\cdot\|_\infty) = \log\left(\frac{2L^2(pB+2)\tau^L p^{L+1}}{\delta}\right)^I$$

$$\leq c'' \epsilon^{-\frac{d}{\alpha}} \log^3 \frac{1}{\epsilon} \log \frac{1}{\delta},$$

where $c''$ is a constant depending on $\log B$, $\omega$ and $\log\log n$.

Plugging it in, we have

$$\mathbb{E}\left[\int_{\mathcal{X}} \left(\widehat{f}_n(x) - f_0(x)\right)^2 d\mathcal{D}_x(x)\right] \leq 4\epsilon^2 + \frac{48\sigma^2}{n}\left(c''\epsilon^{-d/\alpha}\log^3\frac{1}{\epsilon}\log\frac{1}{\delta} + 2\right)$$

$$+ 8\sqrt{6c''}\sqrt{\frac{\epsilon^{-d/\alpha}\log^3\frac{1}{\epsilon}\log\frac{1}{\delta}}{n}}\sigma\delta$$

$$+ \frac{104V_{\mathcal{F}}^2}{3n}\epsilon^{-d/\alpha}\log^3\frac{1}{\epsilon}\log\frac{1}{\delta}$$

$$+ \left(4 + \frac{1}{2V_{\mathcal{F}}} + 8\sigma\right)\delta$$

$$= \widetilde{O}\left(\epsilon^2 + \frac{V_{\mathcal{F}}^2 + \sigma^2}{n}\epsilon^{-\frac{d}{\alpha}}\log^3\frac{1}{\epsilon}\log\frac{1}{\delta}\right.$$

$$\left.+ \sigma\delta\sqrt{\frac{\epsilon^{-\frac{d}{\alpha}}\log^3\frac{1}{\epsilon}\log\frac{1}{\delta}}{n}} + \sigma\delta + \frac{\sigma^2}{n}\right). \qquad (48)$$

Finally we choose $\epsilon$ to satisfy $\epsilon^2 = \frac{1}{n}\epsilon^{-\frac{d}{\alpha}}$, which gives $\epsilon = n^{-\frac{\alpha}{2\alpha+d}}$. It suffices to pick $\delta = \frac{1}{n}$. Substituting both $\epsilon$ and $\delta$ into (48), we deduce the desired estimation error bound

$$\mathbb{E}\left[\int_{\mathcal{X}} \left(\widehat{f}_n(x) - f_0(x)\right)^2 d\mathcal{D}_x(x)\right] \leq c(V_{\mathcal{F}}^2 + \sigma^2)\left(n^{-\frac{2\alpha}{2\alpha+d}} + \frac{D}{n}\right)\log^3 n,$$

where constant $c$ depends on $\log D$, $d$, $\alpha$, $\frac{2d}{\alpha p - d}$, $p$, $q$, $c_0$, $B$, $\omega$ and the surface area of $\mathcal{X}$. $\qquad\square$

## F  SUPPLEMENT FOR EXPERIMENTS

### F.1  DETAILS FOR EXPERIMENTS WITH CARTPOLE

We use the CartPole environment from OpenAI gym. We consider it as a time-inhomogeneous finite-horizon MDP by setting a time limit of 100 steps. We turn the terminal states in the original CartPole into absorbing states, so if a trajectory terminates before 100 steps, the agent would keep receiving zero reward in its terminal state until the end. The target policy is a policy trained for 200 iterations using REINFORCE, in which each iteration samples for 100 trajectories with truncation after 150 time steps. The target policy value $v^\pi$ is estimated to be 65.2117, which we obtain by Monte Carlo rollout from the initial state distribution.

For a given behavior policy, to obtain dataset $\mathcal{D}_h$ at time step $h$, we sample for $K$ independent episodes under the behavior policy and only take the $(s, a, s', r)$ tuple from the $h$-th transition in each episode. This is an excessive way to guarantee the independence among these $K$ samples; in practice, we could directly sample from a sampling distribution. We sample for $\mathcal{D}_h$ for each $h = 1, \cdots, 100$.

We use the render function in OpenAI gym for the visual display of CartPole. We downsample images to the desired resolution via cubic interpolation. A high-resolution image (see Figure 3) is represented as a $3 \times 40 \times 150$ RGB array; a low-resolution image (see Figure 4) is represented as a $3 \times 20 \times 75$ RGB array.

For the function approximator in FQE, we use a neural network that comprises 3 convolutional layers each with output channel size 16, 32 and 32 and a final linear layer. These layers are interleaved with ReLU activation and batch norm layers for weight normalization. For high resolution input, we use kernel size 5 and stride 2; for low resolution input, we use kernel size 3 and stride 1. For experiments with high resolution, in each step of FQE, we solve the regression by training the network via stochastic gradient descent with batch size 256 for 20 epochs. In high-resolution experiments, we use 0.01 learning rate; in low-resolution experiments, we use 0.001 learning rate. We compute the average and standard deviation of FQE's result over 5 random seeds.

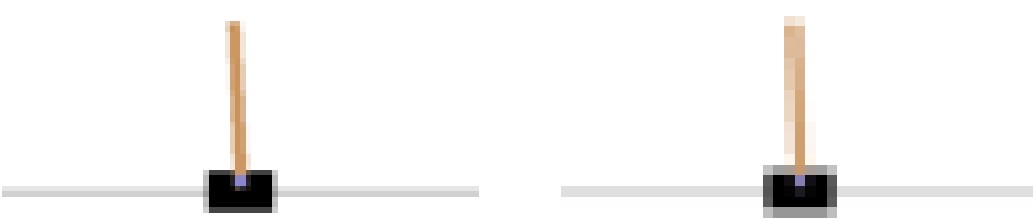



Figure 3: CartPole in high resolution.

Figure 4: CartPole in low resolution.



## F.2 EXPERIMENTS WITH LUNARLANDER

In addition to our CartPole experiments, we present numerical experiments for evaluating FQE with CNN function approximation on the LunarLander environment [5]. The LunarLander problem has a 8-dimensional continuous intrinsic state space. We consider a finite-horizon MDP with horizon $H = 100$ in this environment. In our experiments, we solve the OPE problem with FQE (Algorithm 1). We take the visual display of the environment as states. These images serve as a high-dimensional representation of LunarLander's original 8-dimensional continuous state space. In our algorithm, we use a deep CNN to approximate the Q-functions and solve the regression with SGD. The results are report in Table 3.

Table 3: Value estimation $\widehat{v}^\pi$ under high resolution and low resolution. The true $v^\pi \approx 55.982$ is computed via Monte Carlo rollout.

| Sample size | (A) No distribution shift | | (B) Off-policy | |
| $K$ | High res | Low res | High res | Low res |
|---|---|---|---|---|
| 5000 | $55.9 \pm 3.8$ | $55.5 \pm 3.6$ | $55.5 \pm 5.5$ | $55.0 \pm 5.9$ |
| 10000 | $55.1 \pm 2.7$ | $55.7 \pm 2.9$ | $55.8 \pm 4.0$ | $55.4 \pm 3.9$ |
| 20000 | $55.4 \pm 1.9$ | $55.6 \pm 2.0$ | $56.3 \pm 3.1$ | $56.0 \pm 3.5$ |

The experiment setup is almost the same as our CartPole experiments. We conduct this experiment in two cases: (A) data are generated from the target policy itself; (B) data are generated from a mixture policy of 0.9 target policy and 0.1 uniform distribution. A high-resolution image (see Figure 5) is represented as a $3 \times 40 \times 70$ RGB array; a low-resolution image (see Figure 6) is represented as a $3 \times 20 \times 35$ RGB array.

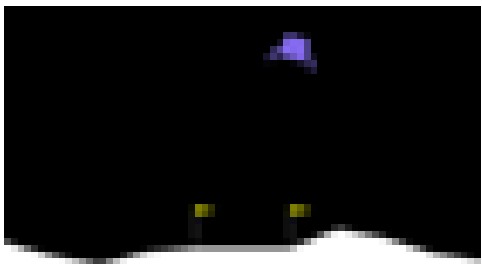



Figure 5: LunarLander in high resolution.

Figure 6: LunarLander in low resolution.



## F.3 ERROR DECAY RATE IN CARTPOLE

We plot the relative error of FQE in our CartPole experiments from Section 5. Figure 7 and 8 show that the estimation error follows an exponential decay. We also plot this relative error in log-log plots (Figure 9 and 10). We can observe that the curves are linear, which confirms the form of our

theoretical bound in Theorem 2. Moreover, we can observe that the slope, which represents the decay rate of the estimation error, is generally the same between high-resolution and low-resolution experiments. This confirms our theory that the decay rate takes little influence from the ambient dimension.

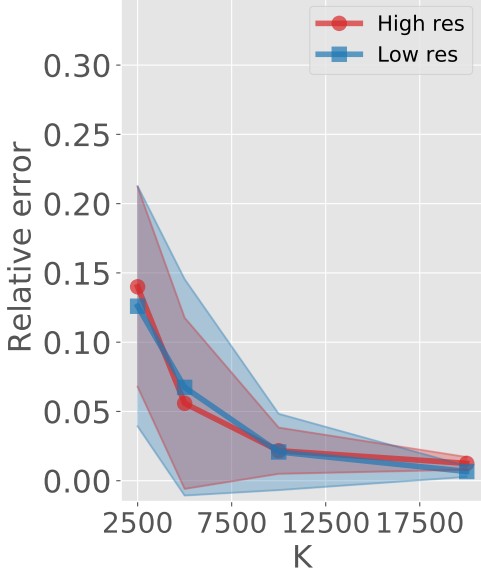

Figure 7: On-policy CartPole.

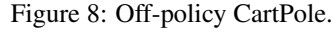

Figure 8: Off-policy CartPole.

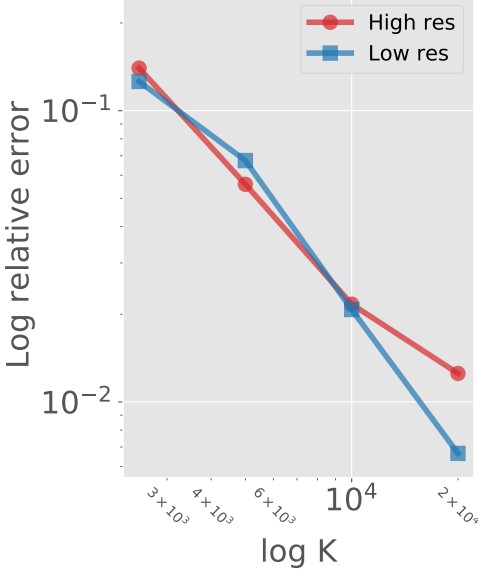

Figure 9: On-policy CartPole (log-log plot).

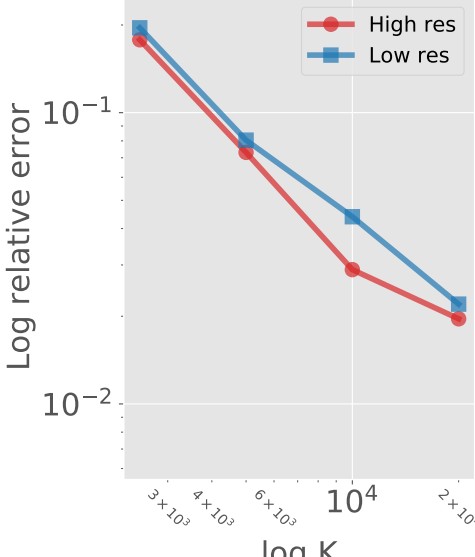

Figure 10: Off-policy CartPole (log-log plot).

