# OpenReview forum: "Sample Complexity of Nonparametric Off-Policy Evaluation on Low-Dimensional Manifolds using Deep Networks"
_ICLR.cc/2023/Conference — ICLR 2023 poster_

### Official Review · Reviewer_sfri · 2022-10-24

**Confidence:** 3
**Correctness:** 4
**Technical Novelty And Significance:** 3
**Empirical Novelty And Significance:** 1
**Recommendation:** 8

**Clarity, Quality, Novelty And Reproducibility:**

**Clarity:**
The manuscript is written very clearly and easy to follow.
One thing that limits the insight provided by the main body of the manuscript is that it hardly sheds light into the proof techniques.
Occasionally, things could be improved, in particular in the Definition of Besov spaces as well as the statement of the main result in Theorem 1 -- see the more detailed comments down below.

**Quality:**
The manuscript is written very well, the theoretical tools are adequate and are applied properly.

**Novelty:**
The study of the estimation error under the assumption of a low dimensional state-action space appears to be novel. I can right now not judge the relation of the theoretical tools to existing results on the generalization error under the assumption of low dimensionality in the context of supervised learning.


**Strength And Weaknesses:**

**Strengths:**
* Sound rigorous analysis; proper application of adequate mathematical tools.

**Weaknesses:**
* Little insight into the main steps of the proof is given in the main body.
* Experimens are weak: I do not believe that a theoretical paper necessarily requires experiments. However, if experiments are provided, they should be well designed and provide substantial insight to the theoretical results. The biggest shortcomings of the experiments is their implications: 1) "performance [i.e., the fitted value function] of FQE on high-resolution data and low-resolution data is similar" 2) "estimation becomes increasingly accurate as sample size K increases". Where these findings don't contradict the Theorem 1 they do not evaluate its main claim, which is a error decay rate, which is independent of the ambient dimension. I think the experiments would be much stronger if tested for this rate, i.e., for a given setup simply increase the number of samples $K$ and then plot the estimation error on a log log plot including the predicted power law decay of $O(K^{-\frac{\alpha}{2\alpha+d}})$.

**Summary Of The Paper:**

The manuscript studies the estimation error of fitted Q-evaluation with convolutional neural networks under the assumption of a low dimensional state-action space.
The main contribution is that the decay rate of the estimation error only depends on the dimension of the state-action space rather than the ambient dimension.
The result builds on an new approximation result for Besov functions on submanifold, which is established in the manuscript.


**Summary Of The Review:**

The manuscript studies an important and timely problem within the theory of reinforcement learning.
It is well written and provides a result on the estimation error, which depends on the intrinsic dimension of the state-action space and offers an analogon to existing result in the context of supervised learning.
Overall, I believe this manuscript provides a valuable contribution to the theoretical RL literature. My main critcism is summarized in the following points:

* *Insights into the proof and tools:* The manuscript would be stronger if it provided more insight into the techniques and main steps in the proof of the main results. In particular, this would is more important in my view compared to the discussion of the $\chi^2$ distance and absolute density ratio, which could be moved to the appendix without big sacrifices, as well as as the paragraph *Sample Complexity Comparison*, which does not offer too much insight compared to Table 1.
* *Experiments:* The experiments show that estimated value function does only weakly depend on the ambient dimension. However, this does not imply anything about the estimation error nor its decay rate for increasing sample size, which is the content of Theorem 1. I think a much more suitable experiment to get experimental evidence for Theorem 1 would be one that evaluates the estimation error and compares the decay to the predicted power law decay $O(K^{-\frac{\alpha}{2\alpha+d}})$. In its current form, I think the space used for the experimental section could be of better use to provide more insight to the proof. However, with experiments confirming the predicted power law decay and possibly its tightness, a presentation of them would be justified in the main body of the manuscript.
* *Definition of Besov space:* Currently, the definition is not very clear to pass. More precisely, $U$ is not introduced in the definition of the Besov norm and it is unclear which atlas is to be taken in the definition of the global Besov norm since its value does depend on the atlas (it could even be infinite). Further, I am suprised to not see any charts appearing in the definition and thus I am wondering what the difference to the notion of the Besov norm induced by the ambient space.
* *Relation to existing results:* The manuscript does give an extensive overview of results on Q-value function estimation. However, the relation to results using Besov functions defined and low dimensional structures in the context of supervised learning is not discussed sufficiently.
* *Presentation of Theorem 1:* The choice $\mathcal F$ as a family induced by a CNN is only implicit and should be made explicit. Further, I believe that Algorithm 1 should be in the main body since in its current form it is required to parse the main result.

---

> ### Author Response · Authors · 2022-11-18
> **Response to Reviewer sfri (1/3)**
>
> We really appreciate your valuable review! We provide responses to your comments and questions in the following. Please kindly let us know if you have further concerns or if there was anything we misconstrued. We also updated our submission in accordance with the comments in the review.
>
> > Little insight into the main steps of the proof is given in the main body.
>
> > *Insights into the proof and tools:* The manuscript would be stronger if it provided more insight into the techniques and main steps in the proof of the main results. In particular, this would is more important in my view compared to the discussion of the chi-sqaure distance and absolute density ratio, which could be moved to the appendix without big sacrifices, as well as as the paragraph *Sample Complexity Comparison*, which does not offer too much insight compared to Table 1.
>
> We provided a proof overview for our main theorem (Theorem 2) in our revision. (We swapped the order of Theorem 1 and 2 in the revision, so our main theorem is now Theorem 2.) It is an excellent point that the “Sample Complexity Comparison” paragraph somewhat overlaps with Table 1, so we moved it to the appendix in the revision. Please also feel free to refer to the proof overview for Theorem 1 in Appendix C.1, which is about the $L_\infty$ approximation theory of CNNs.
>
> > Experiments are weak: I do not believe that a theoretical paper necessarily requires experiments. However, if experiments are provided, they should be well designed and provide substantial insight to the theoretical results. The biggest shortcomings of the experiments is their implications: 1) "performance [i.e., the fitted value function] of FQE on high-resolution data and low-resolution data is similar" 2) "estimation becomes increasingly accurate as sample size K increases". Where these findings don't contradict the Theorem 1 they do not evaluate its main claim, which is a error decay rate, which is independent of the ambient dimension. I think the experiments would be much stronger if tested for this rate, i.e., for a given setup simply increase the number of samples K and then plot the estimation error on a log log plot including the predicted power law decay.
>
> We agree that our original experiments only show the estimation error “depends weakly on the ambient dimension” and does not say much about the error decay rate. We plotted the relative estimation error of our CartPole experiments in Appendix G.3 of our revision. We also made a log-log plot of this error, in which we observe that the estimation error curves are approximately linear, which confirms the form of our theoretical bound. Moreover, we observed that the slope is generally the same between high-resolution and low-resolution experiments, signifying the decay rate has a weak dependence on the ambient dimension, congruous with our theory. However, it is difficult to compare the slope in these figures with the exponent $-\frac{\alpha}{2\alpha + d}$ in our theoretical bound. The reason is twofold: (1) the leading constants in our theoretical bound could play a noticeable role in experiments and occlude the real exponent; (2) more importantly, it is hard to know exact values of $\alpha$ and $d$ in practice. Thus, it would be difficult to have a meaningful comparison for the decay rate between empirical observations and theoretical results.

---

> > ### Author Response · Authors · 2022-11-18
> > **Response to Reviewer sfri (2/3)**
> >
> > > *Definition of Besov space:* Currently, the definition is not very clear to pass. More precisely, U is not introduced in the definition of the Besov norm and it is unclear which atlas is to be taken in the definition of the global Besov norm since its value does depend on the atlas (it could even be infinite). Further, I am suprised to not see any charts appearing in the definition and thus I am wondering what the difference to the notion of the Besov norm induced by the ambient space.
> >
> > $U$ is a local patch (chart) of the manifold $\mathcal{M}$ introduced in Section 2.2. The definition of Besov functions actually does not rely on the choice of atlas, in that, charts are compatible on a smooth manifold (Definition 1). Please feel free to refer to [2,3,4] for more details about Besov space.
> >
> > On the other hand, to define the Besov norm, as the reviewer has noted, it is essential to pre-fix an atlas on the manifold so that the norm is well-defined to prevent arbitrary rescaling, as it is highlighted at the end of our original Definition 4. We revised Definition 4 (in Section 2.3) to avoid any possible ambiguity. As a side remark, we can simply choose a pre-fixed atlas as tangential projections similar to Definition 5 in [1].
> >
> > [1] Minshuo Chen, Haoming Jiang, Wenjing Liao, Tuo Zhao. Nonparametric regression on low-dimensional manifolds using deep ReLU networks: Function approximation and statistical recovery.2022. [2] Geller, D. and Pesenson, I. Z. Band-limited localized parseval frames and Besov spaces on compact homogeneous manifolds. Journal of Geometric Analysis, 21(2):334– 371, 2011. [3] Triebel, H. Theory of Function Spaces. Modern Birkhäuser Classics. Birkhäuser Basel, 1983. [4] Triebel, H. Theory of function spaces II. Monographs in Mathematics. Birkhäuser Basel, 1992.
> >
> > > *Relation to existing results:* The manuscript does give an extensive overview of results on Q-value function estimation. However, the relation to results using Besov functions defined and low dimensional structures in the context of supervised learning is not discussed sufficiently.
> >
> > We added a literature review for related work in supervised learning at the end of Section 2 of our revision. There exists rich literature about using neural networks to approximate functions supported on compact domain in Euclidean space [5,6,7,8,9,10]. These results suffer from the curse of dimensionality, in that to approximate a function up to certain error, the network size grows exponentially in the data dimension. A recent line of work in supervised learning [10,11,12] confirms the ability of neural networks to adapt to data’s intrinsic low-dimensional structures. These results provide an important motivation for our study; however, they cannot be directly applied to off-policy RL problems. The major difference is that off-policy RL problems are sequential in nature, and the difference between target and behavior policies can lead to a substantial distribution shift. We need to develop new analysis to tackle such a challenge. It is also worth mentioning that the most recent, relevant one along this line of work [11] studies convolutional residual networks for classification tasks, which is different from our setting. Our study on CNNs is comparatively new.
> >
> > [5] Bunpei Irie and Sei Miyake. Capabilities of three-layered perceptrons. IEEE 1988 International Conference on Neural Networks, pp. 641–648 vol.1, 1988. [6] Ken-Ichi Funahashi. On the approximate realization of continuous mappings by neural networks. Neural networks, 2(3):183–192, 1989. [7] Boris Hanin. Universal function approximation by deep neural nets with bounded width and relu activations. arXiv preprint arXiv:1708.02691, 2017. [8] Zhou Lu, Hongming Pu, Feicheng Wang, Zhiqiang Hu, and Liwei Wang. The expressive power of neural networks: A view from the width. In Advances in Neural Information Processing Systems, pp. 6231–6239, 2017.[9] Dmitry Yarotsky. Error bounds for approximations with deep relu networks. Neural Networks, 94:103–114, 2017. [10] Minshuo Chen, Haoming Jiang, Wenjing Liao, Tuo Zhao. Efficient approximation of deep relu networks for functions on low dimensional manifolds. NeurIPS 2019. [11] Hao Liu, Minshuo Chen, Tuo Zhao, Wenjing Liao. Besov function approximation and binary classification on low-dimensional manifolds using convolutional residual networks. ICML 2021. [12] Liu, Hao, et al. "Benefits of Overparameterized Convolutional Residual Networks: Function Approximation under Smoothness Constraint." arXiv preprint arXiv:2206.04569(2022).

---

> > > ### Author Response · Authors · 2022-11-18
> > > **Response to Reviewer sfri (3/3)**
> > >
> > > > *Presentation of Theorem 1:* The choice $\mathcal{F}$ as a family induced by a CNN is only implicit and should be made explicit. Further, I believe that Algorithm 1 should be in the main body since in its current form it is required to parse the main result.
> > >
> > > Thank you for your suggestion! In our revision, we moved Algorithm 1 to the end of Section 3 and highlighted what the CNN class $\mathcal{F}$ is in the statement of our main theorem (now Theorem 2).
> > >
> > > Hope we have successfully addressed your questions and concerns! Please let us know if you have any other questions.

---

> ### Author Response · Authors · 2022-12-01
> **Follow-up**
>
> Dear reviewer,
>
> Hope you have seen our response. Aside from our response below, we also revised our submission, for instance, by adding a proof overview for our main theorem at the end of Section 4 as well as the estimation error decay rate plots in Appendix G. (See a full summary of changes in "Response to All Reviewers" above.) We hope our response and revision has successfully addressed your questions and concerns. Please definitely let us know if you have any further questions. Thanks!

---

> ### Author Response · Authors · 2022-12-12
> **Follow-up for Reviewer sfri**
>
> Dear Reviewer sfri,
>
> Hope you have read our response. We hope that our response and revision has successfully addressed your questions and concerns. If you should have any questions, we will try our best to respond before the discussion stage closes. Thank you for your attention!
>
> Finally, as the discussion period is coming to an end, we would like to thank you again for reviewing our submission!

---

> > ### Comment · Reviewer_sfri · 2022-12-12
> > **Reply**
> >
> > Dear authors,
> >
> > thank you for your very extensive replies!
> >
> > I see that you have made some nice changes including an added outline of the proof of the main results. I am recommending to accept the paper to the area chair once more.
> >
> > Regarding the experiments: I appreciate that you added log-log plots in the appendix and agree that $\alpha$ is hard to get a hold on ($d$ could be controlled in a synthetic experiment). Nevertheless, I think a plot with more points would be beneficial, currently, for every curve there are only four points.
> >
> > Regarding the definition of the Besov norm: the definition is not independent of the atlas. Hence, I am still wondering, which one your are taking. Is it an arbitrary but fixed one?
> >
> > Best

---

> > > ### Author Response · Authors · 2022-12-12
> > > **Reply to Reviewer sfri**
> > >
> > > Dear Reviewer sfri,
> > >
> > > Thank you very much for your reply and recommendation! We agree that more points in the log-log plots would be beneficial and will improve this in the next version.
> > >
> > > As for the Besov norm, you’re right that the the definition is not independent of the atlas. For our result, our atlas $\{(\phi_i, U_i)\}$ is specified in Appendix C.1 Step 1. This atlas essentially is a projection onto the tangent space $T_{\textbf{c}_i}(\mathcal{X})$ (plus some rescaling that is without loss of generality). We will make this clearer in the next version, and please feel free to let us know if you have any further questions.

---

### Official Review · Reviewer_Mweq · 2022-10-25

**Confidence:** 3
**Correctness:** 3
**Technical Novelty And Significance:** 2
**Empirical Novelty And Significance:** 2
**Recommendation:** 5

**Clarity, Quality, Novelty And Reproducibility:**

The paper is well written, however, the theoretical results seem to be a combination of existing works.

**Strength And Weaknesses:**

Strengths:
- The paper is well written.
- The regret bound with provided theoretical analysis, and especially using a function class-restricted  $\chi^2$ -divergence for insufficient data coverage are new.

Weaknesses:
- The results seem to be a combination of existing results: the sample complexity in Besov space of off-policy evaluation using deep neural networks from [1] and low-dimensional manifolds in deep networks from [2,3,4,5].

 [1]  Thanh Nguyen-Tang, Sunil Gupta, Hung Tran-The, and Svetha Venkatesh. Sample complexity of ofﬂine reinforcement learning with deep relu networks. arXiv preprint arXiv:2103.06671, 2021.
 [2]  Minshuo Chen, Haoming Jiang, Wenjing Liao, Tuo Zhao. Efficient approximation of deep relu networks for functions on low dimensional manifolds. NeurIPS 2019.
 [3] Minshuo Chen, Haoming Jiang, Wenjing Liao, Tuo Zhao. Nonparametric regression on low-dimensional manifolds using deep ReLU networks: Function approximation and statistical recovery.2022.
 [4] Hao Liu, Minshuo Chen, Tuo Zhao, Wenjing Liao. Besov function approximation and binary classification on low-dimensional manifolds using convolutional residual networks. ICML 2021.
 [5] Minshuo Chen, Hao Liu, Wenjing Liao, Tuo Zhao. Doubly robust off-policy learning on low-dimensional manifolds by deep neural networks. 2020.
- Recently, several results about the sample complexity of offline RL without using function approximations are published.
The authors are missing a discussion about these works.



**Summary Of The Paper:**

This paper considers the off-policy evaluation problem in MDPs using function approximations. To estimate the offline target policy, the fitted Q-evaluation method is used, and to approximate Q-functions, CNNs are used. Their theoretical results show that when state-action space has low-dimensional structures, the finite-sample estimation error of FQE converges, depending on (1) the intrinsic dimension without suffering from the curse of high dimensionality, and (2) a function class-restricted  $\chi^2$ -divergence which measures the distribution shift between the experience data distribution and the target policy. They use an experiment on CartPole to evaluate their theory.


**Summary Of The Review:**

The novelty and originality of this paper are low. I lean toward rejecting this paper.

---

> ### Author Response · Authors · 2022-11-18
> **Response to Reviewer Mweq**
>
> We thank Reviewer Mweq for the review! It seems there are some fundamental questions raised in this review. We answer them in the following in hopes that our responses can clear things up:
>
> > The results seem to be a combination of existing results: the sample complexity in Besov space of off-policy evaluation using deep neural networks from [1] and low-dimensional manifolds in deep networks from [2,3,4,5].
>
> We respectfully disagree with the judgment that this paper is a mere combination of existing results. The reason is as follows:
>
> This paper is indeed inspired by the recent progress in supervised learning research which can account for the low-dimensional structures in data [2,3,4]. However, these results are not directly applicable to our setting. The closest recent paper along this line is [3], which studies the generalization error of using ConvResNet on low-dimensional data for 0-1 classification tasks. Though our $L_\infty$ approximation theory might be similar at a high level (we still have different network architecture, so only the high-level ideas like approximating the function over a set of open covers with Taylor/spline approximation are similar), our policy value estimation problem, with a sequential nature and possibly substantial distribution shift, is inherently different from their supervised classification task and thus uses a different statistical technique. Moreover, note that the problem in [5] is causal inference and not sequential.
>
> On the other hand, the goal of this paper is perpendicular to that of [1], which is about the general case of neural network function approximation for Besov $Q$-functions. Our goal is to explore how the low-dimensional structure of the state-action space, which is common in practice, could affect things in off-policy RL problems (e.g., standard RL assumptions, distribution shift, etc.) and verify if using CNNs as $Q$-function estimators sequentially is still adaptive to such low-dimensional structure in OPE, as mentioned in Paragraph 4 of our introduction. These questions that arise when we introduce low-dimensionality into offline RL are of independent interest and not obvious to answer, even in the presence of [1,2,3,4,5].
>
> Moreover, we’d like to point out that our main theorem (now Theorem 2) does not follow from [1] and uses a different proof technique. Actually, the order of the estimation error in [1] is “suboptimal” (please see Table 1 in our paper). Instead, we take an entirely different route in the statistical part of our analysis in order to get our better rate. This is another difference between our paper and [1], besides our focus on low-dimensionality and CNNs.
>
> [1] Thanh Nguyen-Tang, Sunil Gupta, Hung Tran-The, and Svetha Venkatesh. Sample complexity of ofﬂine reinforcement learning with deep relu networks. arXiv preprint arXiv:2103.06671, 2021. [2] Minshuo Chen, Haoming Jiang, Wenjing Liao, Tuo Zhao. Efficient approximation of deep relu networks for functions on low dimensional manifolds. NeurIPS 2019. [3] Minshuo Chen, Haoming Jiang, Wenjing Liao, Tuo Zhao. Nonparametric regression on low-dimensional manifolds using deep ReLU networks: Function approximation and statistical recovery.2022. [4] Hao Liu, Minshuo Chen, Tuo Zhao, Wenjing Liao. Besov function approximation and binary classification on low-dimensional manifolds using convolutional residual networks. ICML 2021. [5] Minshuo Chen, Hao Liu, Wenjing Liao, Tuo Zhao. Doubly robust off-policy learning on low-dimensional manifolds by deep neural networks. 2020.
>
> > Recently, several results about the sample complexity of offline RL without using function approximations are published. The authors are missing a discussion about these works.
>
> In our revision, we added a few more papers that do not use function approximation in the “Additional Related Work” paragraph at the end of Section 1, in addition to the ones we originally had. Please let us know if we misunderstood or if there is any other literature that we should discuss.
>
> Hope we have clarified some of your doubts on our paper. Please let us know if you still have any questions. Thank you for your reconsideration!

---

> ### Author Response · Authors · 2022-12-01
> **Follow-up**
>
> Dear reviewer,
>
> Hope you have seen our response. We explained more about our contributions in the response and added more literature review in our revision. We hope our response and revision has successfully addressed your questions and concerns. Please definitely let us know if you have any further questions. Thanks!

---

> ### Author Response · Authors · 2022-12-12
> **Follow-up for Reviewer Mweq**
>
> Dear Reviewer Mweq,
>
> Hope you have read our response. If our response and revision has successfully addressed your questions and concerns, would you kindly consider raising the score? And if you should have any questions, we will try our best to respond before the discussion stage closes. Thank you for your attention and consideration!
>
> Finally, as the discussion period is coming to an end, we would like to thank you again for reviewing our submission!

---

### Official Review · Reviewer_cmJs · 2022-10-26

**Confidence:** 3
**Correctness:** 3
**Technical Novelty And Significance:** 3
**Empirical Novelty And Significance:** 2
**Recommendation:** 8

**Clarity, Quality, Novelty And Reproducibility:**

The paper is very clear, novel and reproducible. And I found it to be high quality.


**Strength And Weaknesses:**

Strengths:
- Novel and tighter sample complexity bound for off-policy deep Q-learning with convnets that is characterized in terms of the dimension of the underlying data manifold and the \chi^2 divergence between on- and off-line policies.

Weaknesses:
- the class of convnet functions used in the does not include skip connections which are always used in modern CNN architectures like ResNets. Does incorporating skip connections change the analysis or complexity bounds?
- the experimental results are suggestive but quite cursory. It would be interesting to test networks with different numbers of channels, layers and see if the empirical estimation errors actually increase and decrease in accordance with what is predicted by the new bounds.
- the bounds agree with empirical findings that CNNs seem to have a sample complexity that is lower than the ambient image dimension. But it's not clear if the theory will yield advances in applications.


**Summary Of The Paper:**


This paper provides new sample complexity guarantees for convolutional neural networks applied to off-policy Q-learning of MDPs with low-dimensional manifold structure.

The work is novel in that it provides a new characterization of the sample-complexity in terms of the underlying manifold dimension rather than the the ambient dimension of the data and action space -- which can be extremely large for state spaces defined in terms of image and video data. The bound is also novel in that it is utilizes the \chi^2 divergence between the logging policy and the target policy, rather than the maximum density ratio which can be unbounded for continuous action spaces.

The authors define their bound with respect to a restricted but flexible class of Besov functions which captures many of the essential aspects of modern convolutional neural networks.

Finally, the paper concludes with a limited set of experiments on the CartPole environment which demonstrate that Q-functions fitted on images do learn with fewer examples than the ambient dimension of the data.


**Summary Of The Review:**

This work provides novel and improved bounds that further our understanding of how and why deep Q-learning (with convnets) performs well in very high dimensional state spaces. As such it is of interest to theorists currently working at the intersection of deep learning and RL. I support adding this paper to the 2023 ICLR program.

---

> ### Author Response · Authors · 2022-11-18
> **Response to Reviewer cmJs**
>
> We really appreciate your kind review! We provide responses to your comments and questions in the following. Please kindly let us know if you have further concerns or if there was anything we misconstrued. We also updated our submission in accordance with the comments in the review.
>
> > The class of convnet functions used in the does not include skip connections which are always used in modern CNN architectures like ResNets. Does incorporating skip connections change the analysis or complexity bounds?
>
> From a theoretical perspective, including skip connections in CNNs does not change our analysis or complexity bounds. It would just require a new analysis for the covering number in Lemma 8. We focus on CNNs for consistency, as our experiments also do not use residual networks.
>
> From a computational perspective, including skip connections can help the training (optimization) of neural networks, but this is beyond the scope of our paper.
>
> > The experimental results are suggestive but quite cursory. It would be interesting to test networks with different numbers of channels, layers and see if the empirical estimation errors actually increase and decrease in accordance with what is predicted by the new bounds.
>
> The original goal of our experiments is just to show the estimation error does not really depend on the data representation dimension, but it is a good question. Theoretically, we expect any parameterization deviating too much from ours to have larger estimation error on average because it would break the optimal bias-variance tradeoff, but it is difficult to expect how the estimation error would trend empirically as network architecture varies.
>
> > The bounds agree with empirical findings that CNNs seem to have a sample complexity that is lower than the ambient image dimension. But it's not clear if the theory will yield advances in applications.
>
> The primary goal of this theoretically oriented paper was to provide a theoretical guarantee that the OPE estimation error (likewise the sample complexity to reach certain accuracy) is mostly governed by the intrinsic dimension. However, this result can have some practical implications. For example, as our empirical studies show, the estimation error is indeed mostly uninfluenced by the data representation dimension. This can give practitioners more confidence in using large, overrepresented features for data. Moreover, our theory also tells practitioners that they may avoid using unnecessarily large neural networks when data exhibit low-dimensional structures. This can reduce variance and computational requirements. We revised the conclusion of our submission by adding a short discussion about these practical implications.
>
> Finally, thank you again for your review! Hope we have successfully addressed your questions and concerns! Please let us know if you have any other questions.

---

> > ### Comment · Reviewer_cmJs · 2022-12-13
> > **Reply to authors' reply**
> >
> > > The original goal of our experiments is just to show the estimation error does not really depend on the data representation dimension, but it is a good question. Theoretically, we expect any parameterization deviating too much from ours to have larger estimation error on average because it would break the optimal bias-variance tradeoff, but it is difficult to expect how the estimation error would trend empirically as network architecture varies.
> >
> > Thanks. Then summing up, I take it that you picked your experimental CNN network structure based on what the theorems say is optimal. And the subsequent experimental results included above show that using a network whose architecture differs from this optimal structure actually produces worse results. Is this the correct take-away message from the table added above?

---

> > > ### Author Response · Authors · 2022-12-14
> > > **Reply to Reviewer cmJs**
> > >
> > > Dear Reviewer cmJs,
> > >
> > > Thank you for your reply! Your conclusion is correct, though in reality, one cannot exactly find the optimal network architecture suggested by our theorem. The reason is twofold: (1) the leading constants in our theorem could play a noticeable role in practice; (2) more importantly, it is hard to know exact values of $\alpha$ and $d$ in practice. Therefore, in our original experiments, we just used the architecture that performs best empirically. Our follow-up experiments show that deviating from this architecture increases the error, and this is congruous with our theorem, which says deviating from the optimal architecture would break the bias-variance tradeoff and increase the error.

---

> ### Author Response · Authors · 2022-12-01
> **Follow-up**
>
> Dear reviewer,
>
> We also conducted additional experiments on how modifications in network architecture might affect estimation error. First, we conducted a set of experiments examining the effect of number of convolutional layers. In comparison to our architecture in the submission, we tested a CNN with an additional layer of the same size added to the end and a CNN with the last layer removed (on high-resolution data with $K=10000$).
>
> | Fewer Layers | Original (3 Layers) | More Layers |
> | :---: | :---: | :---: |
> | $59.2 \pm 1.1$ | $67.0\pm 1.8$ | $63.6\pm 5.1$ |
>
> In the same setting, we also examined the effect of number of channels ($J$ in our paper). Keeping the number of layers constant at 3, we tested a CNN with fewer channels (8, 16, 16) and a CNN with more channels (32, 64, 64).
>
> | Fewer Channels | Original (16, 32, 32) | More Channels |
> | :---: | :---: | :---: |
> | $65.0 \pm 1.9$ | $67.0\pm 1.8$ | $63.6\pm 3.6$ |
>
> Please definitely let us know if you have any further questions. Thanks!

---

### Official Review · Reviewer_5uSb · 2022-10-27

**Confidence:** 3
**Correctness:** 4
**Technical Novelty And Significance:** 2
**Empirical Novelty And Significance:** 2
**Recommendation:** 6

**Clarity, Quality, Novelty And Reproducibility:**

The writing of this paper is mostly ok in the sense that it is easy to follow. However, some part essential for understanding the significance of this work is missing in the main context, especially the technical part. Also, the experimental result does not seem to properly address the main claim.

**Strength And Weaknesses:**


The main strength of this paper is that it provides the first regret bound result on off-policy evaluation with deep neural network approximation with practical assumptions. The authors provide a theorem that bounds the sample complexity in terms of the intrinsic dimension and the $\chi^2$-divergence between the sampling policy and the target policy, where both the dimension and the divergence generally improves upon previous results.

However, there are a several vagueness and weakness, listed as follows

- The paper spends lots of effort in introducing the background (at least the CNN part is a little bit distracting, as the CNN structure does not seem to influence the complexity analysis?)
- On the other hand, the authors does not provide much explanation about what leads to the theoretical success of capturing the intrinsic dimension of the dataset. Has similar tools been leveraged in the literature of supervised learning?
- Similarly, what is the intuition behind the proof that removes the data coverage assumption?
- The paper adopted the Besov spaces and the Besov closure property, and the sample complexity is dependent on the Besov parameter $\alpha$. How does this $\alpha$ affect the sample complexity and the scope of hypothesis class?
- Theorem 2 appears abruptly. What is this theorem for? Is Theorem 2 basically a Lemma essential for proving Theorem 1? This part seems incomplete.
- In the experimental section, the authors Illustrate their claim by showing that the performance under high resolution and low resolution are similar. However, that might be due to the simplicity of the CartPole problem, let along the fact that $3\times 40\times 150$ and $3 \times 20 \times 75$ are not significantly different. More empirical validations are needed to truly verify their claim.

Also, there are a few typos:
- In the definition of Reach, the $inf$ has to be taken on $x \in \bar{\mathcal{T}}$ instead of $\mathcal{T}$?
- In the convolutional network part, $\mathcal{W}_{j, :, :}$ is not $D \times C$ (Caption of Figure 2)

**Summary Of The Paper:**

This paper considers the problem of off-policy evaluation using deep convolutional neural network. The main contribution of this paper is a theoretical justification of the ability that a deep neural network captures low-rank structures in high-dimensional datasets in off-policy RL settings. Although preliminary results on off-policy evaluation problem has been obtained in Nguyen-Tang et al. (2021), this work is stronger without the data coverage assumption. Also, it leverages a restricted $\chi^2$-divergence which is generally tighter than the density ratio. Finally, the sample complexity is measured in terms of the intrinsic dimension of the problem, that is significantly smaller than the data dimension.

Overall, the proposed theoretical analysis on the convergence of deep fitted Q-evaluation for off-policy RL and CNN networks achieves superior regret bound result due to the ability to capture low-dimensional structure.


**Summary Of The Review:**

Overall, the paper seems prospective, it provides a sharper sample complexity under a more general assumption. I will consider raising my score if the technical novelty of this paper, Theorem 2, and the experimental results can be better explained.

---

> ### Author Response · Authors · 2022-11-18
> **Response to Reviewer 5uSb (1/2)**
>
> We really appreciate your valuable review and your willingness to raise the score! We provide responses to your comments and questions as follows. Please kindly let us know if you have further concerns or if there was anything we misconstrued. We also updated our submission in accordance with the comments in the review.
>
> > Theorem 2 appears abruptly. What is this theorem for? Is Theorem 2 basically a Lemma essential for proving Theorem 1? This part seems incomplete.
>
> In our original submission, Theorem 2 is essential for proving Theorem 1. In detail, Theorem 2 suggests a proper choice of CNN structures in Theorem 1 for estimating $Q$-functions and characterizes the bias in the estimation. In Section 4 of our revision, we introduce Theorem 2 first for better presentation. (Onward in this response, we refer to the CNN approximation result as Theorem 1 and our main theorem as Theorem 2.)
>
> > The paper spends lots of effort in introducing the background (at least the CNN part is a little bit distracting, as the CNN structure does not seem to influence the complexity analysis?)
>
> Thank you for your suggestion! The preliminary section is meant to provide a necessary brief exposure to core concepts used in later sections, including manifold, Besov functions, and CNNs. The CNN structure is directly relevant to the complexity analysis for proof of Lemma 8, where a perturbation analysis is used based on the network structure. However, we understand that it might come across as distracting. In our revision, we trimmed down the introduction to CNNs and deferred some details to the appendix.
>
> > On the other hand, the authors do not provide much explanation about what leads to the theoretical success of capturing the intrinsic dimension of the dataset. Has similar tools been leveraged in the literature of supervised learning?
>
> We capture intrinsic structures in the state-action space by estimating the $Q$-function with a properly chosen CNN class. Specifically, our CNN class is capable of extracting local geometric structures on the manifold (see Step 1 of Theorem 1’s proof sketch in Appendix C.1): each point on the manifold can be mapped to a point in a low-dimensional Euclidean space. As a consequence, the bias (Theorem 1) and variance (Lemma 8 and 10) in $Q$-function estimation are free of the curse of data ambient dimensionality, which leads to the fast estimation rate in Theorem 2.
> We added a literature review for related work in supervised learning at the end of Section 2 of our revision. A recent line of work in supervised learning [1,2,3] confirms the ability of neural networks to adapt to data’s intrinsic low-dimensional structures. These results provide an important motivation for our study; however, they cannot be directly applied to off-policy RL problems. The major difference is that off-policy RL problems are sequential in nature, and the difference between target and behavior policies can lead to a substantial distribution shift. We need to develop new analysis to tackle such a challenge. It is also worth mentioning that the most recent, relevant one along this line of work [2] studies convolutional residual networks for *classification* tasks, which is different from our setting. Our study on CNNs is comparatively new.
>
> [1] Minshuo Chen, Haoming Jiang, Wenjing Liao, Tuo Zhao. Efficient approximation of deep relu networks for functions on low dimensional manifolds. NeurIPS 2019. [2] Hao Liu, Minshuo Chen, Tuo Zhao, Wenjing Liao. Besov function approximation and binary classification on low-dimensional manifolds using convolutional residual networks. ICML 2021. [3] Liu, Hao, et al. "Benefits of Overparameterized Convolutional Residual Networks: Function Approximation under Smoothness Constraint." arXiv preprint arXiv:2206.04569(2022).

---

> > ### Author Response · Authors · 2022-11-18
> > **Response to Reviewer 5uSb (2/2)**
> >
> > > Similarly, what is the intuition behind the proof that removes the data coverage assumption?
> >
> > We avoid the traditional pointwise data coverage assumption with a careful change of measure argument. Prior papers bound the distribution shift by the single $(s,a)$-pair with the worst density ratio (in the $L_\infty$ sense), which can be prohibitively large and even unbounded when the state-action space becomes very large and the behavior policy does not have a full coverage (see examples in the first item following Theorem 2). Instead, since policy value (our quantity of interest) is defined as the integration of the $Q$-function over a specific distribution, namely the visitation distribution, we conduct a change of measure under the integral and characterize the resulting density ratio with a $f$-divergence that involves the function class that $Q$-functions belong to. This allows the density ratio to stay within an integral in the final result, which can significantly mitigate the effect of any $(s,a)$-pairs that has low probability but has a large density ratio and thus gives a much sharper characterization of the distribution shift. Roughly speaking, we improve the data coverage assumption from the $L_\infty$ sense to the $L^2(P)$ sense. We also added a proof sketch for our main theorem (Theorem 2) in our revision.
> >
> > > The paper adopted the Besov spaces and the Besov closure property, and the sample complexity is dependent on the Besov parameter $\alpha$. How does this $\alpha$ affect the sample complexity and the scope of hypothesis class?
> >
> > $\alpha$ describes the smoothness of the Besov function class—it is a common parameter determining the complexity of nonparametric estimation [4, 5, 6]. A larger $\alpha$ suggests a smoother function class which is easier to estimate. Such an observation matches our theoretical result. In Theorem 2, the obtained convergence rate becomes faster as $\alpha$ increases, approaching a limit of $K^{-1/2}$.
> >
> > Moreover, $\alpha$ is also related to the size of CNN class (hypothesis class). As can be seen from Theorem 1, when $\alpha$ is larger, the size of CNNs becomes smaller, indicating that the target function is easier to estimate.
> >
> > [4] Geller, D. and Pesenson, I. Z. Band-limited localized parseval frames and Besov spaces on compact homogeneous manifolds. Journal of Geometric Analysis, 21(2):334– 371, 2011. [5] Triebel, H. Theory of Function Spaces. Modern Birkhäuser Classics. Birkhäuser Basel, 1983. [6] Triebel, H. Theory of function spaces II. Monographs in Mathematics. Birkhäuser Basel, 1992.
> >
> > > In the experimental section, the authors Illustrate their claim by showing that the performance under high resolution and low resolution are similar. However, that might be due to the simplicity of the CartPole problem, let alone the fact that 3x40x150 and 3x20x75 are not significantly different. More empirical validations are needed to truly verify their claim.
> >
> > It is a good point that more experiments can buttress our claim. We conducted a new set of experiments on LunarLander [7], which is a more complex game environment. This is added in Appendix G.2 of our revision.
> >
> > On the other hand, while it is true that CartPole has simpler dynamics than most other continuous environments, it is still not an easy one to solve in the offline setting [8]. Although the total number of pixels/resolutions between the two experiments just differs by a factor of 4, downsampling by a factor of 4 via cubic interpolation can dramatically alter the “information” in the state images (see Figure 3 and 4 in Appendix G.1; under low resolution, the strong blurring effect makes it impossible to determine the pole’s angle with good accuracy, in contrast to high resolution). To this end, we believe the CartPole experiments in our original submission can still validate our claim to a good extent.
> >
> > [7] https://www.gymlibrary.dev/environments/box2d/lunar_lander/. [8] Agarwal, Anish, et al. "Persim: Data-efficient offline reinforcement learning with heterogeneous agents via personalized simulators." Advances in Neural Information Processing Systems 34 (2021): 18564-18576.
> >
> > Thanks for pointing out the typos! We have corrected them accordingly in the revision.
> >
> > Hope we have successfully addressed your questions and explained the significance of our result! Please let us know if you have any other questions.

---

> ### Author Response · Authors · 2022-12-01
> **Follow-up**
>
> Dear reviewer,
>
> Hope you have seen our response. Please definitely let us know if you have any further questions.
>
> In addition to our revision, we also conducted additional experiments on CartPole with even larger contrast of resolution. We focused on the off-policy setting in Section 5 and compared the estimation error with high-resolution data (dimension 80 $\times 300$) and low-resolution data (dimension 20 $\times 75$). Note that we used grayscale data for faster computation (as opposed to RGB data in Section 5). We observed the same phenomenon as in Section 5, that is, the estimation error of FQE takes little influence from the representation dimension of the data.
>
>
> | $K$ | High res | Low res |
> | :---: | :---: | :---: |
> | $5000$ | $55.2 \pm 7.1$ | $54.8\pm 7.2$ |
> | $10000$ | $65.2 \pm 4.6$ | $64.9\pm 5.0$ |
> | $20000$ | $64.2 \pm 2.8$ | $64.0\pm 3.2$ |
>
>
> If our response and revision has successfully addressed your questions and concerns, would you consider raising the score? Thanks a lot for your consideration!

---

> ### Author Response · Authors · 2022-12-12
> **Follow-up for Reviewer 5uSb**
>
> Dear Reviewer 5uSb,
>
> Hope you have read our previous response and follow-up. If our response and revision has successfully addressed your questions and concerns, would you kindly consider raising the score? And if you should have any questions, we will try our best to respond before the discussion stage closes. Thank you for your attention!
>
> Finally, as the discussion period is coming to an end, we would like to thank you again for reviewing our submission!

---

> ### Author Response · Authors · 2022-12-12
> **Reminder for Reviewer 5uSb**
>
> Dear Reviewer 5uSb,
>
> We’d just like to make a final check on whether you have any further questions about our revised submission, as today is the last day for any discussion between reviewers and authors. If our response and revision has successfully addressed your questions and concerns, would you kindly consider raising the score? Thank you very much for your attention and time!

---

### Author Response · Authors · 2022-11-18
**Response to All Reviewers**

We thank all reviewers for their time and valuable comments! We revised our submission in accordance with the reviews. Major changes are summarized as follows.

- We reorganized parts of our submission to streamline the presentation, including moving some preliminary material for CNN from Section 2.4 to the appendix and condensing the discussion about Theorem 2 in Section 4. We also corrected typos.
- We moved the pseudocode of our algorithm (Algorithm 1) from Appendix to Section 3.
- We exchanged the order of Theorem 1 and Theorem 2 for better presentation and added a short explanation for better introduction to our approximation theory (now Theorem 1).
- We added the citation of a few related works that do not use function approximation in the “Additional Related Work” paragraph of Section 1 as well as a literature review for the related work in supervised learning (“Universal Approximation of Neural Networks” paragraph of Section 2).
- We added a proof sketch for our main theorem (now Theorem 2) in Section 4.
- In Appendix G, we added new experiments in more complex environments as well as plots that show the decay rate of the estimation error in CartPole experiments.
- We revised our conclusion to include a brief discussion about the practical implications of our results.

---

### Decision · Program_Chairs · 2023-01-20

**Decision:**

Accept: poster

**Justification For Why Not Higher Score:**

Primarily of theoretical interest. Moreover similar results appear in other settings; hence not a groundbreaking theoretical development.

**Justification For Why Not Lower Score:**

A solid theoretical contribution for the well established setting of deep fitted q iteration.

**Metareview: Summary, Strengths And Weaknesses:**

The paper offers the first theoretical analysis of deep fitted Q that provides statistical rates that adapt to the intrinsic dimensionality of the state-action space.

Despite similar statements for instrinsic dimension dependence appearing in past work for simpler or different problems like empirical risk minimization, causal inference, static policy learning; this paper offers such an analogous result for deep fitted q evaluation.

For this reason the paper makes a solid theoretical contribution; and this is recognized by most reviewers.

There is some concern that the experimental evaluation is not rich enough; but give that the main contribution is a theoretical guarantee I dont view this as an obstacle for acceptance. Moreover the limited experiments that the authors run do give practical evidence for adaptivity to intrinsic dimension. Moreover, based on reviews the authors did enrich a bit their set of experiments.

**Note From Pc:**

if the above contains the word "oral" or "spotlight" please see: "oral" presentation means -> notable-top-5% and "spotlight" means -> notable-top-25%. As stated in our emails, we are disassociating presentation type from AC recommendations